# Endocrine lineage biases arise in temporally distinct endocrine progenitors during pancreatic morphogenesis

Marissa A. Scavuzzo[1], Matthew C. Hill[1], Jolanta Chmielowiec[2,3,4], Diane Yang[4], Jessica Teaw[2,3,4], Kuanwei Sheng[5,6,7], Yuelin Kong[8], Maria Bettini [8,9], Chenghang Zong[6,7,9], James F. Martin[1,10,11,12] & Malgorzata Borowiak[1,2,3,4,7,9]

Decoding the molecular composition of individual *Ngn3*+ endocrine progenitors (EPs) during pancreatic morphogenesis could provide insight into the mechanisms regulating hormonal cell fate. Here, we identify population markers and extensive cellular diversity including four EP subtypes reflecting EP maturation using high-resolution single-cell RNA-sequencing of the e14.5 and e16.5 mouse pancreas. While e14.5 and e16.5 EPs are constantly born and share select genes, these EPs are overall transcriptionally distinct concomitant with changes in the underlying epithelium. As a consequence, e16.5 EPs are not the same as e14.5 EPs: e16.5 EPs have a higher propensity to form beta cells. Analysis of e14.5 and e16.5 EP chromatin states reveals temporal shifts, with enrichment of beta cell motifs in accessible regions at later stages. Finally, we provide transcriptional maps outlining the route progenitors take as they make cell fate decisions, which can be applied to advance the in vitro generation of beta cells.

[1] Program in Developmental Biology, Baylor College of Medicine, Houston, TX 77030, USA. [2] Center for Cell and Gene Therapy, Texas Children's Hospital, and Houston Methodist Hospital, Baylor College of Medicine, Houston, TX 77030, USA. [3] Stem Cell and Regenerative Medicine Center, Baylor College of Medicine, Houston, TX 77030, USA. [4] Molecular and Cellular Biology Department, Baylor College of Medicine, Houston, TX 77030, USA. [5] Integrative Molecular and Biomedical Sciences Graduate Program, Baylor College of Medicine, Houston, TX 77030, USA. [6] Department of Molecular and Human Genetics, Baylor College of Medicine, Houston, TX 77030, USA. [7] Dan L Duncan Comprehensive Cancer Center, Baylor College of Medicine, Houston, TX 77030, USA. [8] Department of Pediatrics, Section of Diabetes and Endocrinology, Texas Children's Hospital, Baylor College of Medicine, Houston, TX 77030, USA. [9] McNair Medical Institute, Baylor College of Medicine, Houston, TX 77030, USA. [10] Department of Molecular Physiology and Biophysics, Baylor College of Medicine, Houston, TX 77030, USA. [11] The Texas Heart Institute, Houston, TX 77030, USA. [12] Cardiovascular Research Institute, Baylor College of Medicine, Houston, TX 77030, USA. These authors contributed equally: Marissa A. Scavuzzo, Matthew C. Hill. Correspondence and requests for materials should be addressed to J.F.M. (email: jfmartin@bcm.edu) or to M.B. (email: borowiak@bcm.edu)

D erivation of beta cells for regenerative medicine depends upon the robust production of pancreatic progenitors (PPs), yet little is known about the gene expression dynamics of individual progenitors; if heterogeneous progenitors exist that are not recognized by current in vitro approaches, then derived progenitors might have a lower fidelity for the desired fate. Of the progenitors en route to the beta cell fate, endocrine progenitors (EPs), a transient cell type in the embryonic pancreas, are essential to understand as they give rise to all five pancreatic islet cell types (alpha, beta, delta, gamma, and epsilon). Genetic studies have shown that the endocrine master regulator Neurogenin3 + (Ngn3) is necessary for EP formation[1–4]. Ngn3 is also critical for human pancreatic endocrine development: null mutations in *Ngn3* cause neonatal diabetes and block beta cell differentiation from human pluripotent stem cells[5,6]. Thus, all EPs must traverse through a window of Ngn3 expression during embryogenesis, with Ngn3 conserved as a master regulator of the endocrine program across species[7].

During early murine pancreatic development (termed the primary transition), only a few EPs form, mostly giving rise to alpha cells and it is unclear whether they persist into adulthood[2,8]. In later pancreatic development (termed the secondary transition), EP birth is robust and all endocrine cell types are formed[9]. While EPs are able to develop into all islet cell types, individually EPs are thought to be post-mitotic and only give rise to one islet cell[10]. Recent studies have shown that EPs with low *Ngn3* levels retain a higher mitotic index before *Ngn3* expression is upregulated[11,12]. Thus, upon high levels of *Ngn3*, EPs exit the cell cycle and delaminate from the epithelium to differentiate into one endocrine cell type.

*Ngn3* + EPs mostly originate from *Sox9*+/*Spp1*+ bipotent progenitors (BPs) in the trunk epithelium[13]. These BPs can either be induced into EPs and delaminate or remain in the trunk and differentiate into ducts[14]. Tip cells of the epithelium can also give rise to EPs prior to e12.5 when tip cells still retain multipotency[15,16], but their overall contribution to the endocrine pool is not well established. Further, around e15.5 the pancreatic epithelium is remodeled from a web-like plexus to a branched arborized ductal system[17,18]. This reorganization leads to compartmentalization of the branching epithelium, with Ptf1a + and Cpa1/2 + cells forming the pro-acinar tips while EP induction becomes restricted to the trunk[19,20].

How EPs differentiate into endocrine islet cells is intensively studied. This interest has been further enhanced by the immediate need to create a robust source of human beta cells. Multiple studies identified a number of transcription factors involved in the specification of endocrine fate, including alpha cell-specific genes like *Arx*, *Pou3f4*, *Irx1*, and *Irx2*[21,22]. Other studies showed that endocrine differentiation and fate determination is regulated epigenetically[23], with Dnmt3a, Hdac1, and Nkx2-2 repressing the alpha cell gene *Arx* to promote beta cell formation[24]. However, the in vivo chromatin landscapes of EPs are insufficiently characterized, and it is unknown precisely how the epigenomic state influences endocrine cell fate determination.

It is also unknown whether EPs are heterogeneous. Analyzing single Ngn3 + EPs would help to characterize their heterogeneity and further determine if functional EP subtypes exist that may be biased towards one specific endocrine fate over another. Currently EPs are identified mainly by the expression of broad or single markers such as Ngn3, possibly neglecting important distinctions between EPs. Furthermore, lineage tracing experiments have indicated that islet cell fate is determined before hormone expression[10,25]. However, when EPs diverge to differentiate into specific islet cell types is not known, therefore whether this decision occurs before, during, or after *Ngn3* expression remains a prominent question in the field.

Using comprehensive and high-depth approaches, we determine that four *Ngn3* + cell subtypes exist during pancreatic organogenesis. The *Ngn3* + cell diversity reflects developmental maturation rather than EPs primed for specific endocrine cell types and persists from e14.5 to e16.5. We find that e16.5 EPs are transcriptionally different from e14.5 EPs. Further, e16.5 EPs have a higher propensity to form beta cells as the epithelium undergoes remodeling, with changes in EP birth location, as well as in BPs from e14.5 to e16.5 leading to epigenetic and transcriptional shifts in the resulting EPs. Our data suggest that the changes prior to *Ngn3* expression alter the type of EPs that form, with intrinsic shifts in the temporal chromatin accessibility and thus EP potential. Finally, we map out the transcriptional route progenitors take to differentiate into alpha and beta cells, a valuable resource to advance the field of regenerative medicine.

## Results

**Single-cell RNA-seq of the e14.5 pancreas.** The majority of murine pancreatic EPs appear between e13.5 and e17.5, with an abundance of Ngn3-eGFP + EPs arising at e14.5 and e16.5 (Supplementary Fig. 1a-c). We employed a combination of high-throughput and high-depth approaches to gain insight into the molecular signature of EPs and their potential to differentiate into alpha or beta cells (Fig. 1a). Using droplet-based single-cell RNA-seq[26], we transcriptionally profiled 15,228 single cells from 39 e14.5 pancreata, with each cell marked by a STAMP-ID (single-cell transcriptomes attached to microparticles identification; Supplementary Fig. 2a and 2b). To group single cells into respective cell types, we performed graph-based clustering followed by visualization using t-distributed stochastic neighbor embedding (tSNE; Supplementary Fig. 2c), revealing 26 transcriptionally unique subtypes (Fig. 1b, e). We classified the cluster identity using known genes, for instance the expression of *Ngn3* in EPs or *Rbpjl*, *Ptf1a*, *Nr5a2*, and *Cpa1* in tip cells (Supplementary Fig. 2e). We found that a high number of pancreatic cell types and subtypes exist, with heterogeneity in EPs, mesenchyme, and mesothelium. We also captured blood cells along with endothelial cells and neurons. We found equal representation of cells from all three batches in every cluster, with the exception of three mesenchyme clusters and a cluster of hepatocytes composed mostly of batch 1 cells (Mes2 cluster 1; Pr. Mes2 cluster 6; Mes3 cluster 8; Hepato cluster 27), likely due to increased inclusion of surrounding tissue during the first dissection (Supplementary Fig. 2d). We scored each cell in the pancreas on their expression for S-phase, G1, and G2/M transition genes, classifying clusters as proliferating (Pr.) or non-proliferating (Fig. 1c)[27]. While most cells were actively dividing as expected at e14.5, the majority of EPs and alpha and beta cells were found to be in the G1 phase (89.3%), revealing a very limited proliferative capacity of embryonic endocrine cells and progenitors.

The de novo characterization of transcriptionally distinct cell types and subtypes from scRNA-seq data sets relies on robust statistical methodologies. Thus, we used an additional clustering method, Single-Cell Interpretation via Multikernel Learning (SIMLR)[28], which recapitulated the 26 distinct clusters defined through graph-based clustering (Fig. 1d and Supplementary Fig. 2f). However, graph-based clustering more faithfully captured the developmental relationships of cells than kernel-based clustering when visualized by dimensionality reduction, for example the close proximity of EPs to alpha or beta cells (Supplementary Fig. 2g). Thus, we used graph-based clustering for subsequent droplet-based scRNA-seq analyses.

We found two EP subtypes, with EPhi (high levels of *Ngn3*) clustering further from alpha and beta cells while EPlo (low levels of *Ngn3*) were proximal to alpha or beta cells (Fig. 1b). We also

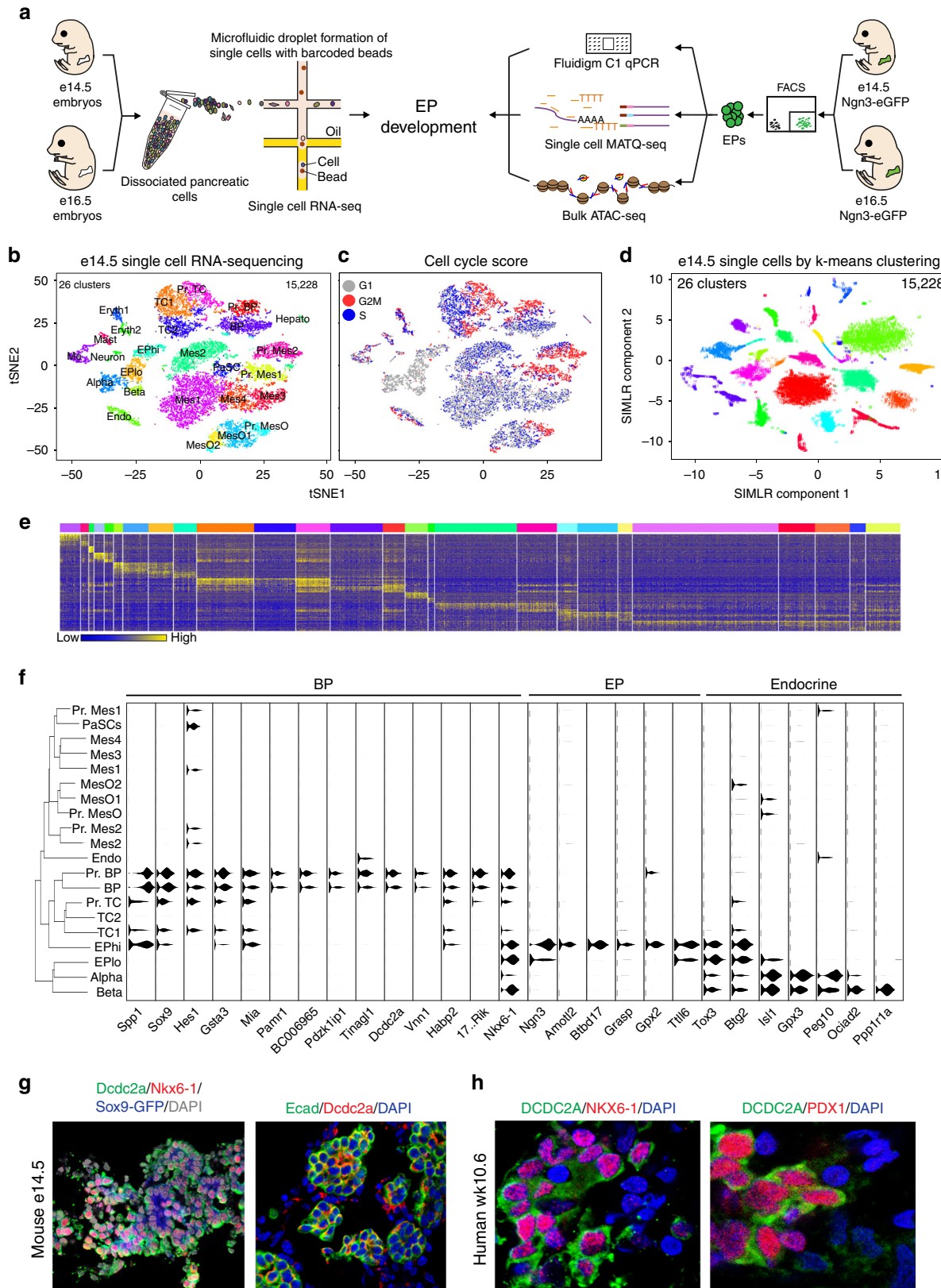

found a higher number of alpha cells (30.8% of endocrine cells) than beta cells (11.3% of endocrine cells; Supplementary Fig. 3a). Hierarchical clustering showed EPlo cells in close association with alpha and beta cells, while EPhi clustered with BPs and tip cells, suggesting that EPhi precede EPlo cells, which then differentiate into alpha or beta cells (Fig. 1f). We validated EPhi and EPlo cells by immunostaining Ngn3-eGFP pancreata for differentially expressed genes (Nkx6-1 and Sox4 in EPhi; Chga, NeuroD1, and E-cadherin in EPlo; Supplementary Fig. 3b and 3c). We also found transcripts expressed exclusively in EPhi, such as *Btbd17*

**Fig. 1** Clustering of distinct cellular populations from the e14.5 mouse pancreas using single-cell RNA-seq. **a** Schematic illustration of the experimental pipeline to investigate EP developmental dynamics. **b** tSNE representation of single-cell transcriptomes grouped using graph-based clustering. 15,228 single-cell transcriptomes were derived from 39 e14.5 pancreata from 3 litters revealing 26 clusters from the top 22 principle components. Cell type annotation from expression of known markers as listed in Supplementary Fig 1. Pr. Mes proliferating mesenchyme, Mes mesenchyme, MesO mesothelium, Pr. BP proliferating bipotent progenitor, TC tip cell, Pr. TC proliferating tip cell, EPhi EPs with high *Ngn3* expression, EPlo EPs with low *Ngn3* expression, Eryth erythrocyte, PaSC pancreatic stellate cells, Mast mast cells, MΦ macrophages, Endo endothelial cells, Hepato hepatocytes. **c** Cell cycle phase analysis of single embryonic pancreatic cells. The approximate cell cycle phase was calculated by scoring individual cells on their expression for S-phase, G1, and G2M transition genes as defined by Kowalczyk et al[31]. **d** Visualization of single-cell transcriptomes using the SIMLR multi-kernel learning algorithm revealed 26 clusters. **e** Each of the 26 graph-based clusters is transcriptionally distinct. Heatmap of the top 100 differentially expressed genes (*y*-axis) from single cells (*x*-axis). Expression ranges from blue (low) to high (yellow). **f** Violin plot of transcripts (shown in *x*-axis) enriched in BP or endocrine clusters (shown on *y*-axis). Dendrogram on left shows cluster relationships. 17...Rik denotes 1700011H14Rik. **g** Newly described BP gene Dcdc2a is expressed in the e14.5 mouse pancreas. On left, immunofluorescent staining of Dcdc2a (green), knock-in Sox9-eGFP (blue), with Nkx6-1 (red) and nuclei (DAPI, gray). On right, immunofluorescent staining of Dcdc2a (red) with epithelial cells marked by E-cadherin (Ecad, green) while nuclei are marked by DAPI in blue. Scale bar = 50 μm. Single channels in Supplementary Fig. 1d. **h** DCDC2a is expressed in the wk10.6 human pancreas. Immunofluorescent staining of DCDC2a (green) with nuclei marked by DAPI in blue. BPs are marked on left, by NKX6-1 (red) and on right, by PDX1. Scale bar = 10 μm. Single channels in Supplementary Fig. 1d. See also Supplementary Figures 1-3

and *Grasp* (Fig. 1f). *Btbd17* has no known function besides epigenetic dysregulation in hepatocellular carcinoma[29], while *Grasp* regulates neuronal intracellular trafficking and metabotropic glutamate receptor organization[30]. In contrast, EPlo uniquely and differentially expressed *Fev* and *St18*, a paralog of Myt1. We also found transcripts expressed in BPs, including enrichment of *Dcdc2a*, which plays a role in inhibition of Wnt signaling in renal tissue[31]. We confirmed Dcdc2a expression by immunostaining in e14.5 BPs and in the human wk10.6 pancreas (Fig. 1g, h, and Supplementary Fig. 1d; 77% ± 4.8 of Sox9 + cells were Dcdc2a + ). By providing transcriptional profiles of every adequately abundant cell type in the e14.5 pancreas, we found new cellular subtypes and candidate markers (Supplementary Data 1).

**Four *Ngn3* + cell subtypes exist in the e14.5 pancreas**. We next sought to delineate the cellular diversity of EPs at e14.5. Of the 15,228 single cells profiled, 9.9% of the total cells in e14.5 pancreas were endocrine cells, with EPs making up ~5.7% of all captured cells (Supplementary Fig. 4a). To account for the low EP percentage and characterize EP heterogeneity, we subclustered cells expressing *Ngn3*, revealing four e14.5 *Ngn3* + cell subtypes termed N14_1, N14_2, N14_3, and N14_4 (Fig. 2a). Tracing of each cell to their original litter and sequencing batch showed that each *Ngn3* + cell subtype was composed of cells evenly mixed from each batch (Supplementary Fig. 4b).

The four *Ngn3* + clusters were transcriptionally distinct (Fig. 2b and Supplementary Data 2), with tip cell markers *Cpa1* and *Cpa2* and trunk cell marker *Spp1* enriched in N14_1 while the top 10 highly expressed genes in N14_4 included endocrine genes such as *Chga*, *Pax6*, *Fev*, and *Pyy*. As these cells were selected by *Ngn3* expression, we sought to determine which pancreatic clusters each cell originated from (Fig. 2c, d). Tracing the STAMP-ID of individual cells revealed that the majority of EPhi cells clustered into N14_1, and N14_3, while EPlo almost exclusively clustered in N14_4, with few cells in N14_3. Interestingly, we found that besides EPhi cells, the N14_1 cluster was also composed of *Ngn3* + BPs and *Ngn3* + tip cells. Thus, N14_1 cells may represent the first-born *Ngn3* + cells. The small N14_2 cluster (6.8% of total *Ngn3* + cells) was overwhelmingly derived from mesenchymal (Mes) clusters, with high expression of epithelial-to-mesenchymal (EMT) markers. N14_3 was composed of 81% EPhi and 10% EPlo, while N14_4 was 1% EPhi and 83% EPlo, with 9% *Ngn3* expressing alpha cells and 7% *Ngn3* expressing beta cells, representing the earliest hormonal cells that still retained *Ngn3* transcript.

To validate expression of select *Ngn3* + cell genes, we used qPCR, RNA in situ hybridization, immunostaining, and flow cytometry. First, we isolated pancreatic Ngn3-eGFP + cells by FACS, with post-sort analysis showing > 95% GFP purity, enrichment of *Ngn3* expression, and no *Ins1/2* or *Gcg* expression (Supplementary Fig. 4c). We further verified that Ngn3-eGFP + cells overlapped in expression with Ngn3 protein by immunostaining, though we found a small number of GFP + anti-Ngn3– cells, likely due to GFP's half-life (Supplementary Fig. 4d). Next, we validated expression of *Ngn3* + cell genes Tox3 and Gpx3 in e14.5 *Ngn3* + cells by qPCR before using in situ hybridization to confirm expression of *Ngn3* + cell genes *Tox3, Gch1, Baiap3*, and *Grin2c* (Supplementary Fig. 4e and 4f). By immunofluorescent staining, we showed the expression of *Ngn3* + cell subtype-specific genes in Ngn3-eGFP + cells in the e14.5 pancreas, including Dcdc2a for N14_1, Vim for N14_1 and N14_2, Dcn and Zeb2 for N14_2, and Rbfox2, Neurod1, and Chga for N14_3 and N14_4 (Fig. 2e, f, and Supplementary Fig. 5a). By quantification, we validated the proportion of Ngn3-eGFP + cells co-expressing these markers, showing strong correlation with scRNA-seq data (Supplementary Fig. 5b). We further validated the different *Ngn3* + subtypes by immunostaining (Supplementary Figs. 6 and 7). For example, we observed N14_1 cells that co-expressed Ngn3-eGFP with E-cadherin but not Chga, as well as Ngn3-eGFP + cells that co-express Chga but not E-cadherin for cluster N14_4 (Supplementary Fig. 6). Flow cytometric analysis further confirmed that *Ngn3* + cell subtypes expressed Rbfox2, Muc1, E-cadherin, Reep5, Nkx6-1, and Dcn at comparable percentages as identified by scRNA-seq (Supplementary Fig. 8a). Together, our data confirm at the transcript and protein level the existence of different *Ngn3* + cell subtypes in the e14.5 pancreas.

***Ngn3* + cell subtypes represent distinct EP maturation stages**. As *Ngn3* + cells in cluster N14_1 express epithelial markers, while N14_4 co-expressed *Ngn3* with alpha and beta cell markers, we hypothesized that each *Ngn3* + cell subtype represented different stages of EP maturation. Using pseudotime analysis[32] as an unsupervised method to determine the temporal order of individual *Ngn3* + cells, we established the developmental trajectory as N14_1, N14_2, followed by N14_3, and finally N14_4 (Fig. 3a; Supplementary Fig. 9a and 9b), supporting our hypothesis that the subtypes reflect EP maturation.

Interestingly, early *Ngn3* + cell subtypes exhibited a high proliferative capacity, defined as % of total cells in G2/M and S-phase (66.7% at N14_1 and 78.9% at N14_2), however with maturation this capacity decreased (33.9% at N14_3 and 12.8% at N14_4; Fig. 3b and Supplementary Fig. 9c). We corroborated and

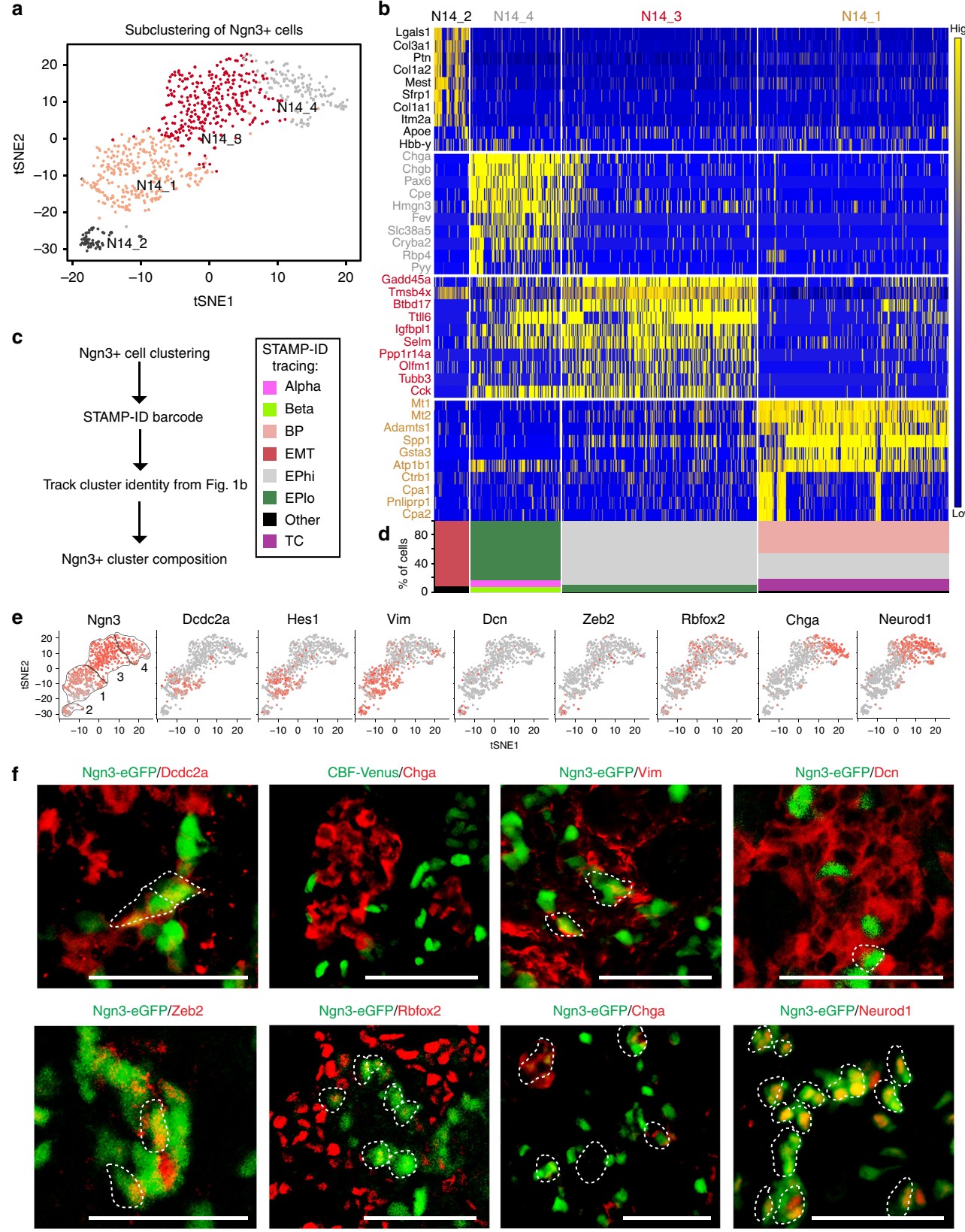

expanded upon previous studies[12], in which EPs expressing low levels of *Ngn3* retained a higher mitotic index than other EPs. Here, we show that N14_1 and N14_2, representing inter-epithelial and delaminating *Ngn3*+ cells, contained a high percentage of cells in the G2/M and S-phase (Fig. 3b). Further,

we found that few embryonic hormonal cells still retained an ability to divide (6.7% of alpha cells and 4.1% of beta cells at e14.5).

Supporting the *Ngn3*+ cell subtype developmental order, we observed enriched expression of known early EP markers in

**Fig. 2** *Ngn3* + cells are heterogeneous with four subtypes validated in the e14.5 mouse pancreas. **a** Four subpopulations of *Ngn3* + cells exist at e14.5. Subclustering of *Ngn3* + cells visualized using tSNE. Colors indicate cluster identity. Clusters were numbered by developmental trajectory (see Fig. 3). **b** Four *Ngn3* + cell subtypes are transcriptionally distinct and represent different developmental stages. Heatmap showing top 10 differentially expressed genes (*x*-axis) of single *Ngn3* + cells (*y*-axis). Expression ranges from blue (low) to high (yellow). **c** Molecular barcode tracing of individual subclustered *Ngn3* + cells to determine the origin of cells as percentage of total subtype in respective Fig. 1b clusters (EMT = Mes clusters). **d** *Ngn3* + cells originate mostly from EPhi and EPlo clusters with some *Ngn3* + cells coming from other clusters. **e** Examples of known and not yet described endocrine markers differentially expressed among *Ngn3* + cells. Feature plots show select transcripts colored by expression level (dark red, high; light red, low; gray, none). **f** Immunofluorescent staining of differentially expressed endocrine progenitor genes, including Dcdc2a, Chga, Vim, Dcn, Zeb2, and Neurod1 in the e14.5 pancreas of Ngn3-eGFP and CBF-Venus (marking Hes gene expression) transgenic mice showing EP heterogeneity. Scale bar = 50 μm. See also Supplementary Figures 4-8

N14_1 (including *Sox9*, *Spp1*, *Onecut1*, and *Hnf1b)*, delamination and migration markers in N14_2 (including *Twist1*, *Snai1*, and *Snai2*), and committed EP markers in N14_3 and N14_4 (including *Chga, Arx, Irx1/2, Neurod1, Pax4,* and *Pax6*; Fig. 3c and Supplementary Fig. 9d). To support that N14_1 cells mature into other *Ngn3* + cell subtypes, we crossed Sox9-Cre[ERT2] mice[33] with the ROSA-mTmG line[34]. As Sox9 is expressed in BPs and is essential for *Ngn3* expression[14], Cre recombination GFP marks BPs as well as their progeny: EPs and endocrine cells. We pulsed these mice with tamoxifen at e13.5 or e15.5 to label N14_1 cells with GFP before collecting tissue 48 h later to analyze markers of *Ngn3* + cell maturation (Supplementary Fig. 10a and 10b). We observed GFP + cells expressing markers of all four stages of *Ngn3* + cells supporting that N14_1 cells expressing Sox9 and E-cadherin develop into N14_2 (Zeb2 + ), N14_3 (Insm1 + , Nkx6-1 + , and Acly + ), and N14_4 (Insm1 + , Nkx6-1 + , Acly + , Chga + , Ghrl + , and Sst + ; Supplementary Fig. 10c).

Endocrine lineage decisions are debated to occur after *Ngn3* expression, with each EP forming a single islet cell[35]. We analyzed transcripts enriched in *Ngn3* + cell subtypes to determine whether alpha or beta cell primed *Ngn3* + cell subtypes exist. While we did not find different subtypes of cells biased towards the alpha or beta lineage within this stage, we did note enriched expression of alpha cell genes *Arx, Irx1, Irx2*, and *Pou3f4* in the N14_4 subtype (Fig. 3c).

Gene ontology analysis of transcripts significantly enriched ($p < 0.01$) in each cluster revealed enrichment of cell-cell adhesion, tissue morphogenesis, and epithelial cell differentiation in N14_1, reflecting early EPs still residing in the epithelium (Fig. 3d)[36]. The EMT functions enriched in N14_2 include cell motility/migration and axon guidance. In the N14_3 subtype, processes associated with epigenetic regulation were enriched, including nucleosome disassembly and chromatin remodeling, as well as secretion and calcium-mediated signaling. N14_4 exhibited enriched genes associated with endocrine pancreatic development, peptide hormone processing, and second-messenger mediated signaling (Fig. 3d).

To analyze transcript dynamics across *Ngn3* + cell maturation, we grouped together genes with similar developmental trajectories using hierarchical clustering (Fig. 3e and Supplementary Fig. 9e). Transcripts associated with the epithelial state including *Spp1* and *Dcdc2a* gradually decreased over time. *Vim1* and *Yap1* peaked and then decreased, inferring the role of the Hippo pathway in this phase of EP maturation. The calcium channel gene *Cacna2d1* and EP gene *Tox3* gradually increased in expression, while genes that peaked in expression in late EP maturation included *Ngn3* and anti-proliferation factor *Btg2* (Supplementary Data 3). Together this analysis allows identification of genes involved in different steps of *Ngn3* + cell maturation.

In order to ascertain the conservation of *Ngn3* + cell transcriptional dynamics, we next compared the transcriptional trends from mouse e14.5 *Ngn3* + cell maturation to human wk9 EP maturation, a comparable developmental stage (Fig. 3f and Supplementary Data 4)[37]. To do this, we first grouped transcripts[38] from human EPs and endocrine cells into corresponding pseudotime clusters from mouse data. Next, we found which genes significantly change between human EPs and endocrine cells (two-sided Student's *t*-test, $p < 0.05$). Finally, we visualized these genes from human endocrine development to assess how the human and mouse expression trends correlate. As in mouse *Ngn3* + cell maturation, we found that genes in cluster 1 and 2 overwhelmingly decreased during human EP maturation. Similar correlation was observed to genes in cluster 3 and 4, with expression increasing during endocrine differentiation. The conserved gene expression dynamics indicates that these genes may play important roles in EP development.

**Ngn3 + cell subtypes are temporally distinct**. To determine whether EP subtypes were transiently present at e14.5 or if they were discerned at a later stage, we performed scRNA-seq of 2006 cells from e16.5 pancreata (Fig. 4a and Supplementary Data 5). We delineated 17 distinct clusters in the e16.5 pancreas equally composed of e16.5 batches (Supplementary Fig. 11a), with two EP subtypes, EPhi and EPlo, also present at this later stage (Supplementary Fig. 11b). After determining the cell cycle phase of each cell, we found widespread proliferation throughout the e16.5 pancreas, with again the exception of EPs and endocrine cells (Fig. 4b).

After subclustering of e16.5 *Ngn3* + cells, we found four transcriptionally distinct *Ngn3* + cell subtypes (Fig. 4c, Supplementary Fig. 11d, and Supplementary Data 6). Tracing each STAMP-ID showed the N16_1 cluster was mostly composed of EPhi cells and *Ngn3* + BPs; with some contribution of *Ngn3* + cells from Mes clusters indicating EMT, and a small percentage of EPlo and *Ngn3* + tip cells (Supplementary Fig. 11d). This suggests that new *Ngn3* + cells co-expressing BP and tip cell markers are constantly forming at e14.5 and e16.5. The N16_2 and N16_3 subtypes were entirely made up of EPhi cells, while N16_4 was mostly EPlo with some EPhi and *Ngn3* + beta cells. No *Ngn3* + alpha cells were captured at e16.5. Thus, it appears that the four *Ngn3* + cell subtypes at e16.5 also reflected EP maturation, with N16_1 coming from epithelial progenitors while N16_4 were beginning to repress *Ngn3* and differentiate.

Concurrent with the notion that N16 subtypes reflect EP maturation, cell cycle analysis showed a gradual decrease in proliferative capacity from N16_1 to N16_2, followed by N16_3, N16_4, with e16.5 beta and alpha cells at the lowest end (Supplementary Fig. 11c). This decrease in the proportion of G2/M and S-phase cells as *Ngn3* + cells commit to the endocrine fate mirrored that of e14.5 *Ngn3* + cell subtypes and endocrine cells. To verify that the e16.5 *Ngn3* + subtypes followed a temporal order, we performed unsupervised pseudotime analysis on e16.5 *Ngn3* + cells. We found that N16_1 cells appeared first, followed by N16_2, then N16_3, and finally N16_4 (Fig. 4d and

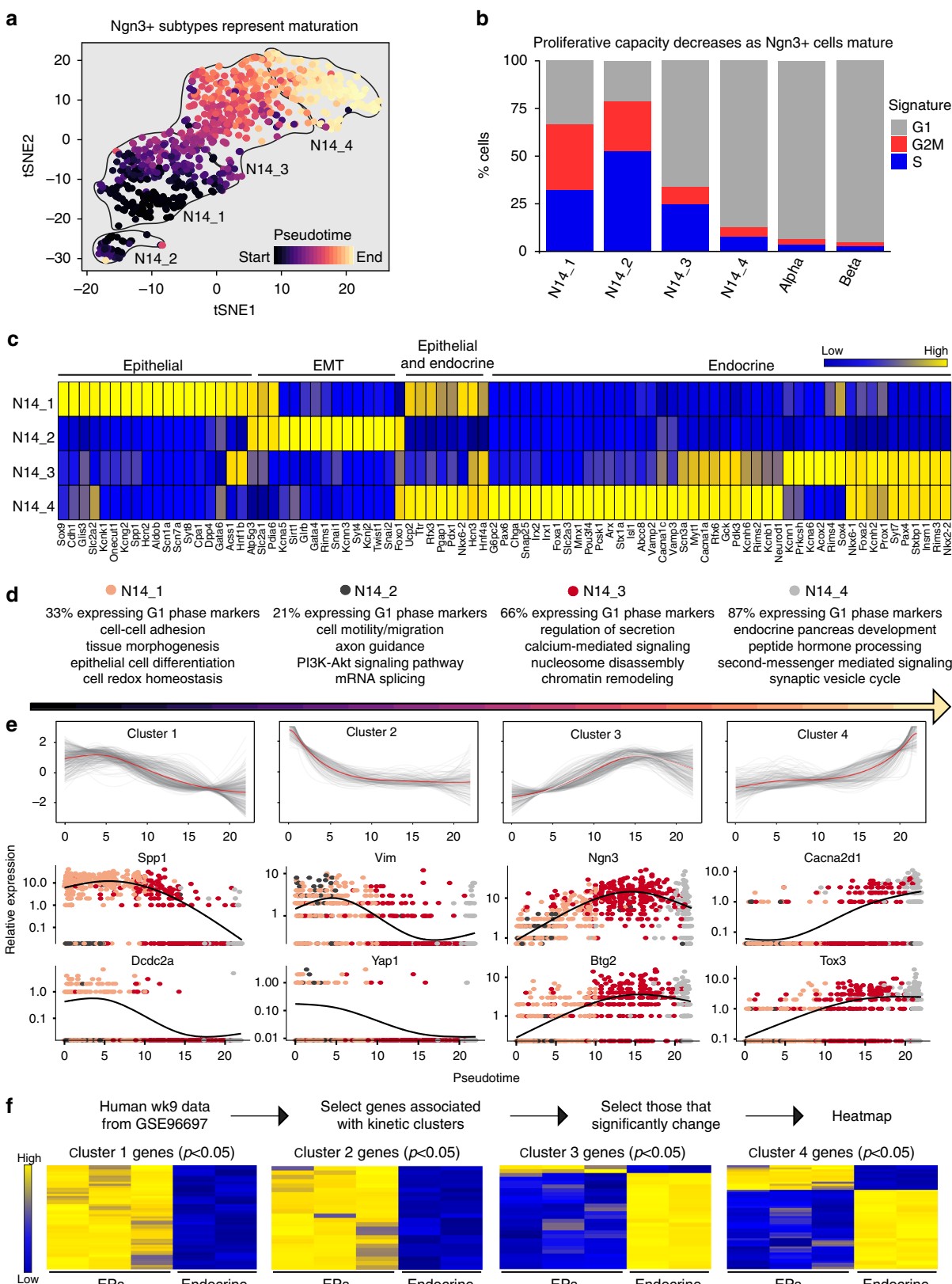

Supplementary Fig. 11e). We further discovered that, similar to e14.5 EPs, N16_1 expressed genes associated with cell adhesion and migration while N16_4 were enriched for exocytosis, glucose homeostasis, and insulin processing (Supplementary Fig. 11f). These results echo the developmental trajectory implied through

STAMP-ID tracing, with N16_1 cells composed of progenitors and N16_4 composed of EPlo and beta cells, and all *Ngn3* + cells gradually exiting the cell cycle.

Next, we validated e16.5 *Ngn3* + cell heterogeneity. Immunos-taining confirmed e16.5 *Ngn3* + cell subtypes, showing

**Fig. 3** *Ngn3* + cell heterogeneity reflects maturation. **a** *Ngn3* + cell subtypes belong to different temporal groups. Pseudotemporal ordering of single cells revealed the developmental trajectory of *Ngn3* + cell subtypes as N14_1, N14_2, N14_3, then N14_4. **b** Cell cycle phase analysis of e14.5 endocrine and *Ngn3* + cell subtypes. The approximate cell cycle phase was calculated by scoring individual cells on their expression for S-phase genes, G1 genes, and G2M transition genes as defined by Kowalczyk et al.[31]. **c** Log2 Z-score normalized heatmap showing *Ngn3* + cell-stage-specific transcripts x-axis of *Ngn3* + cell subtypes, N14_1 through N14_4 (y-axis). Expression ranges from blue (low) to high (yellow). N14_1 highly express epithelial markers, N14_2 express delamination and migration markers, while N14_3 and N14_4 express markers of differentiated endocrine cells. **d** N14 clusters reflect stages of *Ngn3* + cell maturation. Gene ontology analysis of genes significantly enriched ($p < 0.01$) in each cluster reveals potential functions of each *Ngn3* + cell cluster, with colors indicating *Ngn3* + cell cluster identity. **e** Dynamics of gene expression as *Ngn3* + cells mature. Hierarchical clustering of relative gene expression patterns shifting across pseudotime, with examples of genes enriched in each cluster shown below, each dot representing a single cell and color coded by *Ngn3* + cell cluster identity, see also Figs. 2a or 3d. **f** Transcriptional dynamics are conserved during human wk9 EP maturation. Genes from kinetic clusters 1–4 (Fig. 3e) were used to cluster human wk9 transcripts with significant changes between EPs and endocrine cells ($p < 0.05$, two-sided Student's *t*-test). EPs are cells that have downregulated glycoprotein 2, while endocrine maturation is defined by a decrease in E-cadherin expression. Data shown as z-score derived from Ramond et al. (2017); GSE96697. See also Supplementary Figures 9 and 10

heterogeneity of Ngn3-eGFP + cells with Acly, Sox4, Yap1, E-cadherin, Chga, and Nkx6-1 (Supplementary Fig. 12). Flow cytometry showed subsets of Ngn3 + cells co-expressing Mucin1 (enriched in N16_1), Rbfox2 (N16_2 and N16_3), and Nkx6-1 (Supplementary Fig. 8b). We next performed qPCR analysis on FACS enriched Ngn3-eGFP + cells (1.68% Ngn3-eGFP + out of total cells). The Ngn3-eGFP + cells expressed *Ngn3* but no significant levels of *Ins1/2* or *Gcg* (Supplementary Fig. 11g). The newly described pancreatic genes *Gpx3* and *Tox3* were significantly enriched in e16.5 Ngn3-eGFP + EPs compared to Ngn3-eGFP- cells (Supplementary Fig. 11h). Further, beta cell gene *Ociad2* was enriched in e16.5 EPs but not in e14.5 EPs.

We next asked how similar the e16.5 *Ngn3* + cell subtypes were to the e14.5 *Ngn3* + cell subtypes by analyzing select EP markers. Hierarchical clustering of e14.5 *Ngn3* + cell subtypes markers in e16.5 *Ngn3* + cells showed that, based on these key markers, N14_1, N14_2, and N16_1 were related while N14_3, N16_2, and N16_3 clustered together and N14_4 clustered with N16_4 (Fig. 4e). While we found N14_2 as a distinct subtype of Ngn3 + cells highly expressing EMT genes, enrichment of these transcripts were observed in N16_1. This is possibly due to a smaller cell number sampled at e16.5, with this cluster similar to both N14_1 and N16_1. These results demonstrate that e14.5 and e16.5 EPs are heterogeneous reflecting developmental progression, with canonical markers for each subtype maintained between both timepoints.

After merging e14.5 and e16.5 single *Ngn3* + cells (Supplementary Fig. 13a), we found five post-mitotic and two mitotic clusters (6 and 7; Supplementary Fig. 13b) that were predominantly composed of N14_1 cells (Supplementary Fig. 13c). Remarkably, when we traced each cell origin in relation to the e14.5 and e16.5 *Ngn3* + cell subtypes, we found that the e16.5 *Ngn3* + cells clustered separately from the e14.5 *Ngn3* + cells (Fig. 4f). The four e16.5 *Ngn3* + cell subtypes grouped into cluster 4, while the e14.5 *Ngn3* + cells clustered into six groups with N14_1 splitting into three closely related clusters composed of those derived from tip cells, BPs, and EPs (Supplementary Fig. 13c and 13e). To ensure that the differences between e14.5 and e16.5 *Ngn3* + cells was not an artifact of batch effects, we subclustered *Ngn3* + cells with e14.5 beta cells, alpha cells, and BPs, and in all cases founded the e16.5 *Ngn3* + cells clustered separately from e14.5 *Ngn3* + cells (Supplementary Fig. 13d). We found that each cluster was transcriptionally distinct, with the e16.5 cluster 4 expressing beta cell genes such as *Acly, Miat*, and *Insrr* while the N14_1 split apart into *Ngn3* + tip cells in cluster 7 expressing *Cpa1* and *Cpa2* and BPs in cluster 1 expressing *Spp1* (Supplementary Fig. 13f). Hence, while specific markers were maintained between both stages, *Ngn3* + cells are overall changing dramatically over these timepoints.

The divergence of *Ngn3* + cells from e14.5 to e16.5 was surprisingly stark. Therefore, we sought to compare the transcriptomes of single e14.5 and e16.5 *Ngn3* + cells using two additional independent methods: Fluidigm C1 and multiple annealing and dC-tailing-based quantitative scRNA-seq (MATQ-seq)[39]. After isolating e14.5 and e16.5 Ngn3-eGFP + cells, we observed differential expression of established EP genes in single cells over time using Fluidigm C1 analysis, with clustering analysis showing one branch composed of 92.9% e16.5 single EPs while the other was 81.6% e14.5 single EPs (Supplementary Fig. 14a). This supports that *Ngn3* + cells are different between e14.5 and e16.5. To observe changes in a single *Ngn3* + cell subtype over time we next isolated N14_4 and N16_4 by FACS sorting Ngn3-eGFP + cells (Supplementary Fig. 14b). We then applied these cells for MATQ-seq and clustered using the SIMLR algorithm (Fig. 4g), that was shown to be suited for small scRNA-seq data sets (<3000 cells)[40]. MATQ-seq of single *Ngn3* + cells substantiated that EPs change over time, with 799 genes significantly enriched in e14.5 EPs compared to 1854 genes in e16.5 EPs and separation e14.5 EPs and e16.5 EPs ($p < 0.05$, Fisher's exact test).

We next applied an additional unsupervised method to determine Ngn3 + subtype relationships. *Ngn3* + cells ordered from early e14.5, with N14_1 appearing before the branch point, diverging into either later e14.5 *Ngn3* + cell subtypes such as N14_4 clustering last or e16.5 *Ngn3* + cells (Fig. 4h and Supplementary Fig. 13g). We combined this analysis with branched expression analysis modeling (BEAM)[32] to detect statistically significant (*q*-value < 0.01) changes in transcripts between clusters. We found that genes enriched in e16.5 *Ngn3* + cells grouped together (cluster 1), as well as genes from early e14.5 *Ngn3* + cells (cluster 2), late e14.5 *Ngn3* + cell genes (cluster 4), and shared genes (cluster 3, Fig. 4i and Supplementary Data 7). We found that early e14.5 *Ngn3* + cells were enriched for tip cell genes, including *Cpa1, Cpa2, Rbpjl*, and *Prss2*, while late e14.5 *Ngn3* + cells exhibited transcripts associated with alpha cells such as *Arx, Gcg*, and *Pax4*. In contrast, beta cell genes were enriched in the e16.5 *Ngn3* + cells (cluster 1). Further, gene ontology analysis of enriched transcripts in e16.5 *Ngn3* + cells compared to e14.5 *Ngn3* + cells showed functions associated with epigenetic regulation and beta cell development (Fig. 4j). Direct comparison of e14.5 and e16.5 *Ngn3* + cell subtypes revealed differentially expressed genes, with the top 50 most significant changes illustrated in Supplementary Fig. 14c. Taken together, this shows that while e14.5 and e16.5 *Ngn3* + cell subtypes exist, they change over time, with differences in transcript profiles including alpha cell genes in late e14.5 *Ngn3* + cells and beta cell genes and epigenetic regulators enriched in late e16.5 *Ngn3* + cells.

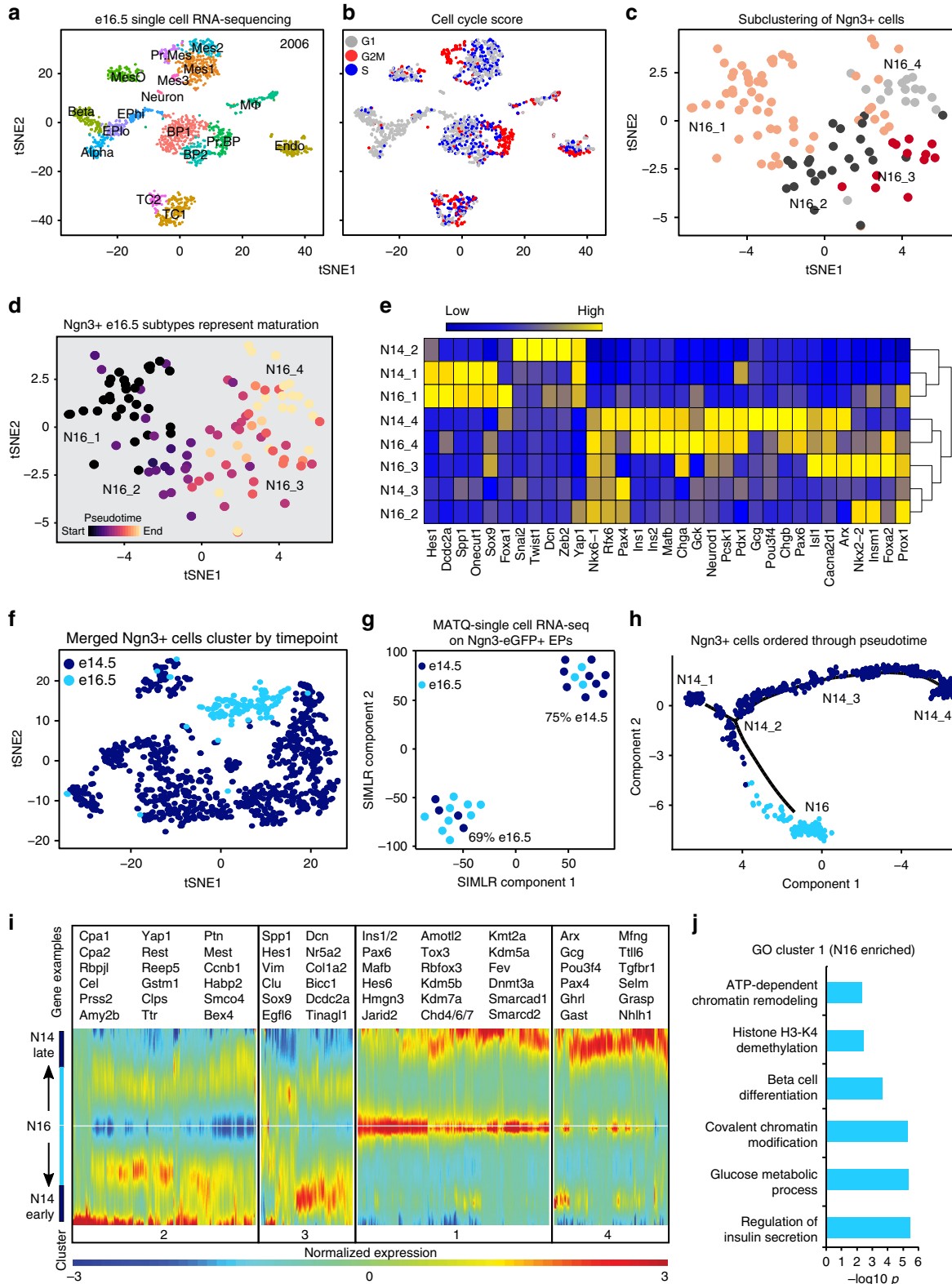

**EP chromatin accessibility shifts from e14.5 to e16.5.** The increase in expression of epigenetic regulators associated with the beta cell fate in e16.5 Ngn3+ cells concurrent with increased beta cell and decreased alpha cell genes led us to investigate the contribution of the chromatin state in diverting EPs to become alpha or beta cells. We found changes in epigenetic regulator expression in EPs, some with known functions in endocrine cell development, including *Dnmt3a, Ezh2*, and *Rest*[23,24,41], as well as many not yet described in endocrine differentiation (Supplementary Fig. 15a). Interestingly, the expression of known beta cell determinant *Dnmt3a*[24,42] was increased in e16.5 Ngn3+ cells, while the alpha cell regulator *Ezh2*[23,43] was enriched in e14.5 Ngn3+ cells. To investigate chromatin accessibility changes over time, we performed Assay for Transposase Accessible Chromatin

**Fig. 4** *Ngn3* + cells at e14.5 are distinct from *Ngn3* + cells at e16.5. **a** tSNE representation of 2006 single-cell transcriptomes from 21 e16.5 pancreata from two litters. Cell types are annotated from gene expression profile. Colors indicate cluster identity. Mes mesenchyme, MesO mesothelium, EPhi EPs with high *Ngn3*, BP bipotent progenitors, EPlo EPs with low *Ngn3*, MΦ macrophages, Endo endothelial cells. **b** Cell cycle phase analysis of single embryonic pancreatic cells. The approximate cell cycle phase was calculated by scoring individual cells on their expression for S-phase genes, G1 genes, and G2M transition genes as defined by Kowalczyk et al.[31]. **c** An enhanced view shows four subpopulations of *Ngn3* + cells at e16.5. Subclustering of *Ngn3* + cells visualized using tSNE. Colors indicate cluster identity. **d** *Ngn3* + cell subtypes belong to different temporal groups. Pseudotemporal ordering of single cells revealed the developmental trajectory as N16_1, N16_2, N16_3, then N16_4. **e** Log2 z-score normalized heatmap showing expression of key EP genes (x-axis) in EP subtypes (y-axis). Expression ranges from blue (low) to high (yellow). **f** Single *Ngn3*+cell transcriptomes merged for unsupervised clustering and visualization by tSNE. E14.5 *Ngn3* + cells (dark blue) cluster separately from e16.5 *Ngn3* + cells (light blue). **g** MATQ-seq shows the majority of e14.5 and e16.5 single cells cluster into temporally restricted groups. SIMLR component analysis of single Ngn3-eGFP EPs from e14.5 and e16.5 from two batches. E14.5 cells are shown in dark blue while e16.5 cells are shown in light blue. $p < 0.05$ by Fisher's exact test. **h** E14.5 and e16.5 *Ngn3* + cells are transcriptionally distinct. Ordering of merged e14.5 and e16.5 single *Ngn3* + cells shows a divergence of e16.5 *Ngn3* + cells from e14.5. **i** Branched expression analysis modeling showing transcriptional differences between e16.5 *Ngn3* + cells (N16) and N14_1 or N14_4. Examples of genes from each cluster are shown to the right, including enrichment of tip markers in N14_1, alpha cell genes in N14_4, and beta cell genes in N16. **j** Gene ontology analysis of genes enriched in e16.5 *Ngn3* + cells compared to e14.5 *Ngn3* + cells (kinetic cluster 1). See also Supplementary Figures 8, and 11-14

with high-throughput sequencing (ATAC-seq) on purified Ngn3-eGFP + EPs from e14.5 and e16.5 pancreas (Fig. 5a). Analysis of motifs enriched in the open chromatin regions showed those of known endocrine regulators like Rfx6 and Neurod1 (Fig. 5b)[44,45]. We also found 609 distinct e14.5 peaks and 749 e16.5 enriched ATAC peaks with differential chromatin accessibility analysis (adjusted p-value < 0.05; Fig. 5c and Supplementary Data 8-11). In total, we identified 56,640 ATAC peaks, which tended to be located in non-promoter regions of the genome (Supplementary Fig. 15b), consistent with other tissues that show temporal shifts in enhancer usage across ontogeny[46].

Chromatin accessibility of *Ngn3* and *Nr5a2* (tip cell marker) loci was enriched at e14.5 compared to e16.5, while accessibility across *Ins1* and potassium calcium activated channel *Kcnn2* loci was increased at e16.5 (Fig. 5d). To investigate if temporal changes in motif accessibility in EPs corresponded with endocrine fate determination, we performed motif enrichment analysis on the statistically differentially accessible ATAC-seq peaks (adjusted P-value < 0.05). Strikingly, we found that beta cell associated transcription factor motifs including Nkx6-1 and NeuroD1 were well represented at e16.5 (Fig. 5e). Indeed, Nkx6-1 is essential for maintaining beta cell identity and molecular physiology into adulthood[47–50]. scRNA-seq showed a robust increase in expression of Nkx6-1 in e16.5 *Ngn3* + cells compared to e14.5 *Ngn3* + cells (Supplementary Fig. 15d). Interestingly, analysis of Nkx6-1 ChIP-seq data[47] from adult pancreatic islets confirmed that Nkx6-1 was bound to many of the genomic regions we found to be more accessible in e16.5 EPs (Fig. 5f, g). Furthermore, as NeuroD1 motifs were enriched globally in both data sets as well as in e16.5-specific EP ATAC-seq peaks, we assessed NeuroD1 DNA-binding from e14.5 pancreas NeuroD1 ChIP-seq data[51] and found moderately enhanced occupancy across the e16.5 peak set (Fig. 5h). Thus, many of the accessible chromatin regions identified in embryonic EPs are occupied by key pancreatic transcription factors, including Nkx6-1 and NeuroD1.

As many of the differentially accessible regions were annotated to intronic and intergenic regions (92% for e14.5, and 94% for e16.5), we interrogated their epigenetic status. We integrated available histone mark ChIP-seq profiling data derived from murine e13.5 EPs (Ngn3-GFP + low), and e17.5 beta cells[51]. Indeed, the 609 e14.5 enriched ATAC-seq peaks showed high H3K4me1 signal in e13.5 Ngn3-GFP + low cells, consistent with their status as active enhancers with high chromatin accessibility (Fig. 5i). Conversely, e13.5 Ngn3-GFP + low H3K4me1 signal was not highly enriched across the 749 e16.5 accessible ATAC-seq peaks (Fig. 5i). This suggests that the 749 e16.5-specific peaks are not yet commissioned as active enhancer regions at e14.5. However, further histone mark profiling is required to

definitively determine the enhancer status of these regions at e16.5. Thus, the differentially accessible regions identified in EPs via ATAC-seq are dynamic at the epigenetic level. In e17.5 beta cells, we also noted increased signal for the active H3K27ac mark in the accessible peaks of e16.5 EPs compared to e14.5 EPs (Fig. 5j). Conversely, we observed decreased signal for inactive H3K27me3 marks in e17.5 beta cells across e16.5 EP peaks (Fig. 5k). Together this supports that the e16.5 peaks are also active genomic regulatory regions in e17.5 beta cells. In addition, gene ontology analysis of the accessible peaks in e14.5 and e16.5 EPs revealed enrichment of hallmarks of beta cells at e16.5 (Supplementary Fig. 15c). Overall, this suggests that changes in EP chromatin accessibility could influence EP potential, with chromatin accessibility remodeling between e14.5 and e16.5 leading to a more permissive environment for beta cell formation at e16.5.

**EPs exhibit temporal alpha versus beta lineage biases**. To determine if shifts in the chromatin landscape corresponded with changes in EP alpha or beta cell potential, we analyzed alpha and beta cell percentages in the e14.5 and e16.5 endocrine compartment and observed more beta cells present than alpha cells at e16.5 (Fig. 6a). We validated the findings from scRNA-seq by immunostaining and quantification, confirming that more alpha cells were present at e14.5 while more beta cells were present at e16.5 (Fig. 6b, c). We next compared non-hormonal late e14.5 (N14_4) and e16.5 EPs (N16_4) to the earliest hormonal expressing cells at e14.5 (*Ngn3* + /*Gcg* + and *Ngn3* + /*Ins2* + ). N14_4 cells expressed hallmark alpha cell genes similar to the earliest *Ngn3* + /*Gcg* + alpha cells, including *Irx1, Arx, Pou3f4, Irx2,* and *Gcg,* while N16_4 cells were closely related to the earliest born *Ngn3* + /*Ins2* + beta cells expressing *Ins1, Ins2,* and *Ociad2* (Fig. 6d). This suggests that the EP fate is primed depending on the embryonic stage, with a higher potential for alpha cells early during the secondary transition and beta cells later. Overall, these data suggest that the timing in which EPs differentiate and their epigenetic state is critical to fate determination. As we find that the chromatin of e16.5 EPs is prepatterned, cells are likely primed for the beta cell fate. These data potentially outline the ideal EP chromatin accessibility configuration to generate beta cells (Supplementary Data 10 and 11).

**Pancreatic epithelial cells change from e14.5 to e16.5**. We next asked what changes might be associated with distinct EPs at e14.5 and e16.5. Principal component analysis revealed that e14.5 and e16.5 EPs were already different upon induction, with the earliest N14_1 and N16_1 subtypes clustering apart, even more so than

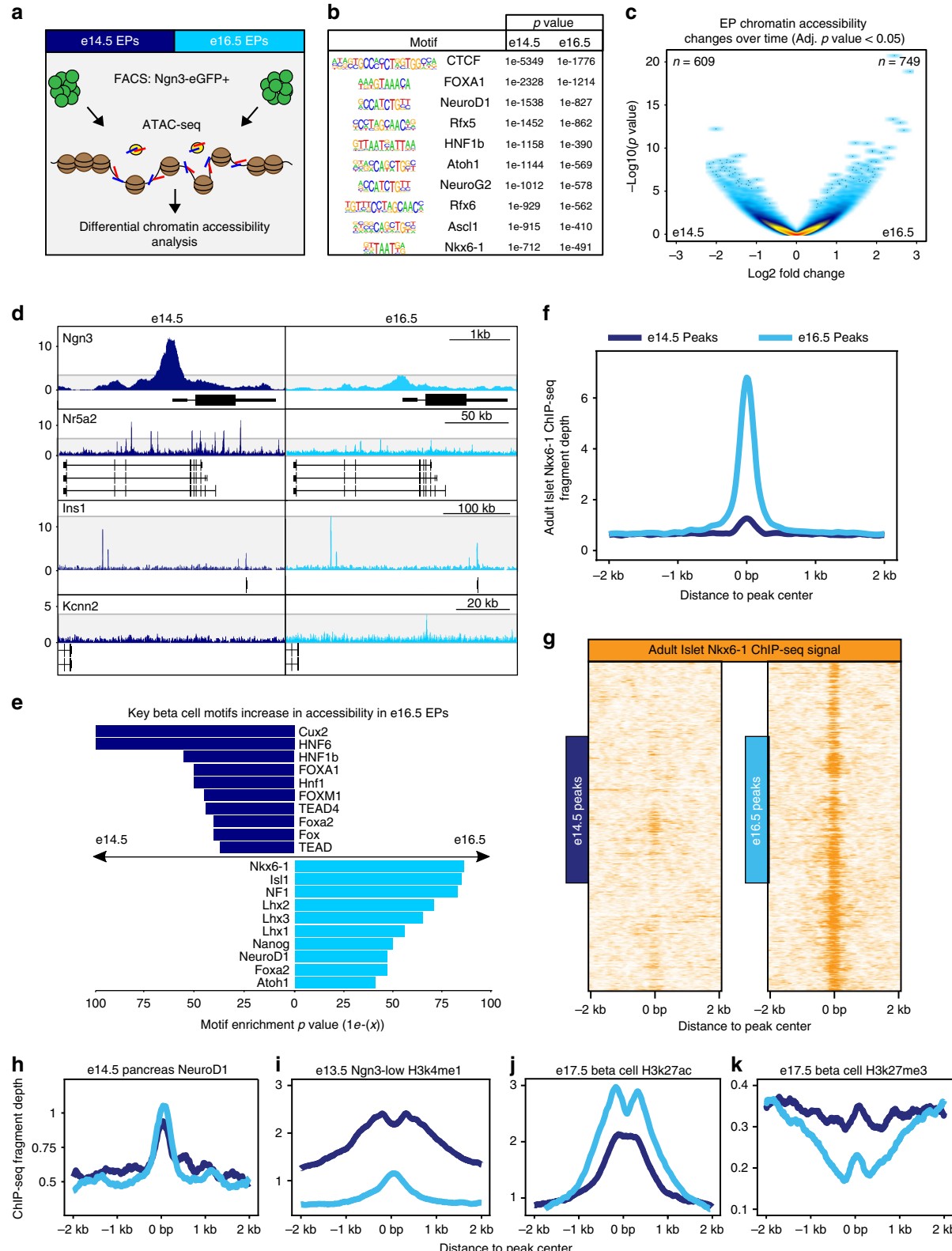

BPs (Fig. 7a; average cluster expression). Thus, we analyzed the epithelium that gives rise to EPs to see if changes were occurring prior to *Ngn3* expression, as the epithelium is remodeling from a plexus to an arborized ductal system throughout these stages (Fig. 7b). STAMP-ID tracing of the earliest e14.5 and e16.5 EP subtypes showed that both BPs and tip cells contribute to the EP pool. Interestingly, tip cells made up 16% of the earliest e14.5

*Ngn3* + cells (N14_1), while tip cells only made up 3% of the earliest e16.5 *Ngn3* + cells (N16_1; Fig. 7c).

As previously reported, we observe a high number of Ngn3-eGFP + cells forming in the proximal trunk regions of the e14.5 and e16.5 epithelium, as demarcated by Mucin1 staining (Supplementary Fig. 16a)[17]. At e14.5 we also found Ngn3-eGFP + cells appearing in distal regions of the epithelium,

**Fig. 5** Shifts in chromatin accessibility promote the potential of endocrine progenitors for the beta cell fate at later stages of embryonic development. **a** Scheme of ATAC-seq experiments. Pancreata from Ngn3-eGFP mice were dissected at e14.5 or e16.5 and Ngn3-eGFP + cells were isolated using FACS for ATAC-seq. $n = 3$ biological replicates. **b** Motifs globally enriched in accessible chromatin regions of e14.5 and e16.5 EPs. **c** The chromatin landscape changes from e14.5 to e16.5. Volcano plot showing chromatin accessibility. Adjusted $p$-value < 0.05. **d** ATAC-seq tracks highlighting the locus of *Ngn3, Nr5a2, Ins1*, and *Kcnn2* from e14.5 (shown in navy) and e16.5 (in light blue) EPs, after normalization and combination of all biological replicates. Genomic loci: *Ngn3*, chr10:61,593,975-61,597,673. *Nr5a2*, chr1:138,737,109-138,913,736. *Ins1*, chr19:52,066,906-52,389,227. *Kcnn2*, chr18:45,839,320-45,911,754. **e** Top differentially enriched motifs in accessible chromatin regions of e14.5 (in navy) and e16.5 EPs (in light blue). **f** Adult islet Nkx6-1 ChIP-seq fragment coverage histogram (per base pair per peak) counted across the e14.5 and e16.5 EP differentially accessible peaks from Fig. 5c. Data from GSM1006208. **g** Adult islet Nkx6-1 ChIP-seq signal (reads) heatmap across the e14.5 and e16.5 EP differentially accessible peaks, as show in Fig. 5f. Rows are peaks, and each peak is centered and signal visualized across a 4 kb window. Low ChIP-seq signal is colored as white, and high signal (reads) is colored orange. **h** The e14.5 whole pancreas NeuroD1 ChIP-seq (GSM2532979) fragment coverage histogram (per base pair per peak) counted across the e14.5- and e16.5-specific EP peaks. **i** The e13.5 Ngn3-GFP low H3K4me1 ChIP-seq fragment coverage histogram (per base pair per peak) counted across the e14.5- and e16.5-specific EP peaks. ChIP-seq data from GSE84324. **j** The e17.5 beta cell H3K27ac ChIP-seq fragment coverage histogram (per base pair per peak) counted across the e14.5- and e16.5-specific EP peaks. ChIP-seq data from GSE84324. **k** The e17.5 beta cell H3K27me3 ChIP-seq fragment coverage histogram (per base pair per peak) counted across the e14.5- and e16.5-specific EP peaks. ChIP-seq data from GSE84324. See also Supplementary Figure 15

---

however, none at e16.5 (Supplementary Fig. 16a). To verify co-expression of tip cell markers in Ngn3-eGFP + cells, we co-stained the e14.5 and e16.5 pancreas for Cpa1/Cpa2, confirming scRNA-seq data at protein level (Supplementary Fig. 16b). We next optically cleared whole e14.5 and e16.5 pancreas[52] and imaged in 3D to tile ~5000 confocal planes and reconstruct the organ (Fig. 7d and Supplementary Fig. 16c). Using this approach we confirmed quantitatively that Ngn3-eGFP + /Ptf1a + tip cells were present at e14.5 but very few at e16.5 (Supplementary Fig. 16d). We further used anti-Ngn3 antibody and confirmed Ngn3 protein presence in Ptf1a + tip cells (Supplementary Fig. 16e).

Despite the fact that EPs are continually born, we found that EPs were distinct upon formation, with N14_1 transcriptionally divergent from N16_1 cells. Moreover, we observed that tip cells contribute to the *Ngn3* + cell pool at e14.5 but significantly less at e16.5. While *Ngn3* + cell birthplace is likely an important contributor to EP heterogeneity and development, it may not be the sole determinant. We therefore hypothesized that if BPs were different from e14.5 to e16.5, then they could contribute to the temporal shift in EP formation and potential.

We merged e14.5 and e16.5 single cells from scRNA-seq and ordered BP, EPhi, EPlo, alpha, and beta cells through pseudotime to find how BPs contribute to endocrine fate decisions (Supplementary Fig. 17a and Fig. 8a). Consistent with our previous analysis, EPhi cells appear prior to EPlo before bifurcation into either alpha or beta cells, with e14.5 EPs clustering together into three clusters while e16.5 EPs clustered apart. Unexpectedly, we found that BPs also formed a branch, signifying a distinction between two BP populations. Tracing the identity of the BP branches showed that each branch consisted of either e14.5 or e16.5 BPs, denoting that BPs are changing over time (Fig. 8a).

BEAM was used to detect statistically branch-dependent genes changing as BPs make fate choices ($q$-value < 0.01; Supplementary Data 12). We assessed the gene expression patterns when e14.5 BPs diverge into either e16.5 BPs or EPs (Fig. 8b). Genes in cluster 1, including *Ngn3, Tox3, Btg2, Baiap3*, and *Rbfox3*, increased in expression as e14.5 BPs differentiated into EPs. Interestingly, the expression of the chromatin remodeler *Smarcd2* was restricted to the EPhi branch, supporting that changes in EP epigenome influence EP potential.

**Loss of AMOTL2 in BPs affects endocrine lineage decisions**. Using BEAM analysis we observed that expression of *Amotl2* in

BPs increased from e14.5 to e16.5 while *Tox3, Btg2*, and *Rbfox3* increased as e14.5 BPs differentiated into EPs. We hypothesized that if *Amotl2* loss-of-function influenced the alpha or beta cell fate determination, then EP fate could be primed prior to *Ngn3* expression, while if loss of *Tox3, Btg2*, or *Rbfox3* influenced the alpha or beta fate then the decision of EPs likely occurred after Ngn3 expression. Since function of these genes in pancreatic development is unknown, we used human pancreatic in vitro differentiation to address this hypothesis.

Firstly, we confirmed robust hESC pancreatic differentiation by staining for SOX17 (~95% positive cells) for definitive endoderm, PDX1 for PPs (~90% positive cells), and NGN3 for EPs (~50% positive cells; Supplementary Fig. 17b). We transduced hESC-derived PPs with lentivirus containing two different shRNAs against *Amotl2, Tox3, Btg2*, or *Rbfox3* and selected the most efficient shRNA for subsequent experiments (Fig. 8c and Supplementary Fig. 17c). After five days in basic media without additional growth factors to induce endocrine differentiation, *AMOTL2* knockdown led to a decrease in INS + cells and an increase in GCG + cells as was analyzed by qPCR and immunostaining (Fig. 8d, e; by immunostaining 2-fold decrease in INS + , 3.7-fold increase in GCG + cell number). The knockdown of *TOX3, BTG2*, and *RBFOX3* did not cause significant changes in alpha versus beta cell in vitro differentiation.

Amotl2 functions in endothelial cell migration, canonical Wnt signaling repression, polarity loss, and Hippo pathway activation[53-56]. *Amotl2*, which exhibited similar expression patterns to *Smarcd2* and *Ngn3*, showed enrichment of chromatin accessibility in e14.5 EPs compared to e16.5 EPs (Supplementary Fig. 17d-f). Further, as in murine pancreatic development, *AMOTL2* significantly increased in hESC-derived PPs and EPs compared to hESCs (Supplementary Fig. 17g). This shift in the number of GCG/INS cells after *AMOTL2 LOF* suggests that endocrine fate may be primed in earlier progenitors. In support of this, we found that e14.5 BPs expressed alpha cell markers while e16.5 BPs had increased expression of beta cell markers, mirroring the subsequent temporal lineage biases (Fig. 8f).

**Transcriptional maps as EPs become alpha or beta cells**. Finally, we sought to determine the transcript dynamics as EPs differentiate into alpha or beta cells. We addressed this by two independent methods of pseudotemporal ordering, Monocle2, which orders single cells along a trajectory before grouping transcripts based off of expression patterns over pseudotime, and Wishbone, which acts as a branch predictor to resolve bifurcating

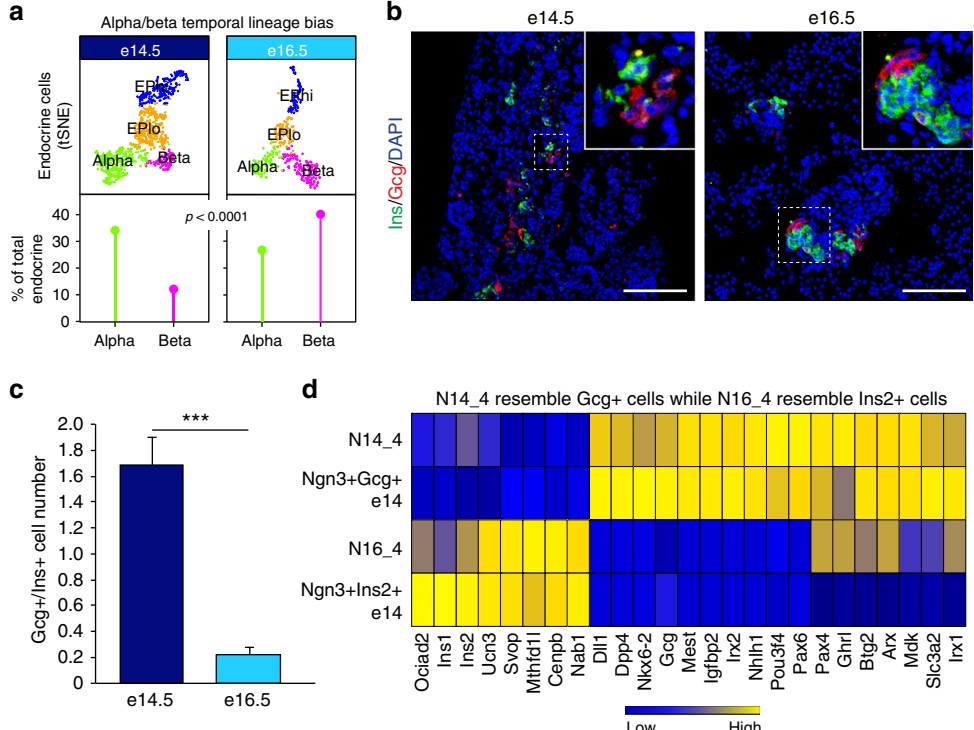

**Fig. 6** Temporal lineage bias of *Ngn3* + cells leads to more beta cells forming at e16.5. **a** Abundance of alpha and beta cells at e14.5 compared to e16.5. At e14.5, a higher percentage of alpha cells compared to beta cells were found compared to the overall number of endocrine cells in the pancreas. At e16.5, a higher percentage of beta cells compared to alpha cells were found. The top panel shows tSNE representation of endocrine populations with quantification below. *p* < 0.0001, Fisher's exact test. **b** Representative immunostaining of e14.5 and e16.5 pancreas for alpha (marked in red by Gcg) and beta (marked in green by Ins) cells. Nuclei are marked by DAPI in blue. Scale bar = 100 μm. **c** Quantification of Gcg + cell number over Ins + cell number from e14.5 and e16.5 pancreata. N = 4 e14.5 and N = 6 e16.5 biological replicates. *p* < 0.005 by two-sided Student's *t*-test. **d** Priming of e14.5 *Ngn3* + cells for the alpha cell fate and e16.5 *Ngn3* + cells for the beta cell fate. Average cluster log2 *z*-score normalized heatmap showing expression of select alpha and beta cell genes (*x*-axis) in late non-hormonal *Ngn3* + cells from e14.5 and e16.5 (N14_4 and N16_4) compared to the earliest born hormonal cells still expressing *Ngn3* (*y*-axis). Expression ranges from blue (low) to high (yellow)

trajectories. The use of both Monocle2 and Wishbone allowed the unbiased analysis of transcripts over pseudotime coupled with a secondary method to visualize transcript changes over time with higher branch fidelity. In Monocle2, we utilized a density peak clustering algorithm and differential expression analysis to determine the top 300 genes (lowest *q*-value) when EPlo cells bifurcate to alpha and beta cells and then analyzed these gene expression dynamics across differentiation (Fig. 9a). Genes that increase in expression as EPs differentiate into beta cells (in cluster 1) included *Gjd2, Mlxipl, Scg2, Sytl4*, and *Ociad2*, which have not yet been described in the embryonic pancreas (Fig. 9b).

Next, we employed the Wishbone algorithm to investigate the dynamics between developmental progression and alpha and beta cell markers. We first confirmed the accuracy of the Wishbone trajectory, showing expression of the early marker *Spp1* in cells ordered earliest, followed by *Ngn3* + cells, then bifurcating into either *Gcg* + or *Ins2* + cells (Fig. 9c). We found EP specific genes including *Ttll6*, which mirrored *Ngn3* expression, albeit its expression was prolonged in beta cells (Fig. 9d). Analysis of endocrine hormone gene expression patterns showed expected patterns for *Ins1, Ins2*, and *Gcg*, with specific expression in developing beta and alpha cells, respectively. Analysis of other hormone transcripts revealed dynamic expression patterns, with *Sst* peaking early in beta cells before complete inhibition, followed by another late peak in expression before final repression. In contrast, *Ghrl* showed an expression peak in early alpha cells before repression, while *Ppy* had early peaks in both alpha and beta cells. *Maob, Gjd2, Scg2, Sytl4, Mlxipl*, and *Papss2* increased in

beta cells as EPs differentiated, while *Isl1, Peg10, Gpx3*, and *Pyy* increased in developing alpha cells (Fig. 9d). RNA in situ hybridization validated the expression of beta cell-specific genes including *Gjd2, Mlxipl, Scg2*, and *Sytl4*, with *Ociad2* protein verified in the developing mouse and human pancreas (Fig. 9e, f). Alpha cell enriched genes, *Peg10* and *Gpx3*, were confirmed by RNA in situ hybridization on e14.5 and e15.5 pancreas, respectively (Fig. 9e). Further studies are needed to assess the function of these newly described pancreatic genes in endocrine development

From the branch point analysis of EP decisions between alpha or beta cells, we revealed potential regulators of this choice and created transcriptional maps of the directions progenitors take to become alpha or beta cells (Supplementary Data 13). Using our scRNA-seq data and available data sets, we compared hPSC-derived pancreatic cells at different stages to e14.5 and e16.5 single-cell clusters to determine the precise stage and transcriptional state of in vitro cells to in vivo cell subtypes (Supplementary Fig. 18a). We performed bulk RNA-sequencing of hPSC-derived cells pancreatic cells at early (12 h) and late EP stage (5 days). Further, we compared published data sets from hPSCs differentiated to the EP stage and beta cell stage from two groups. We found that our hPSC-derived EPs resembled EPs, yet branched alone, likely due to the fact that these cells were a mixed population. Analysis of hPSC-derived sorted Ngn3-GFP + EPs[57] showed that these cells closely related to late N16 subtypes, which exhibit a higher beta cell potential. Finally, by comparing hPSC-derived beta cells[58] we found that they resemble yet are not

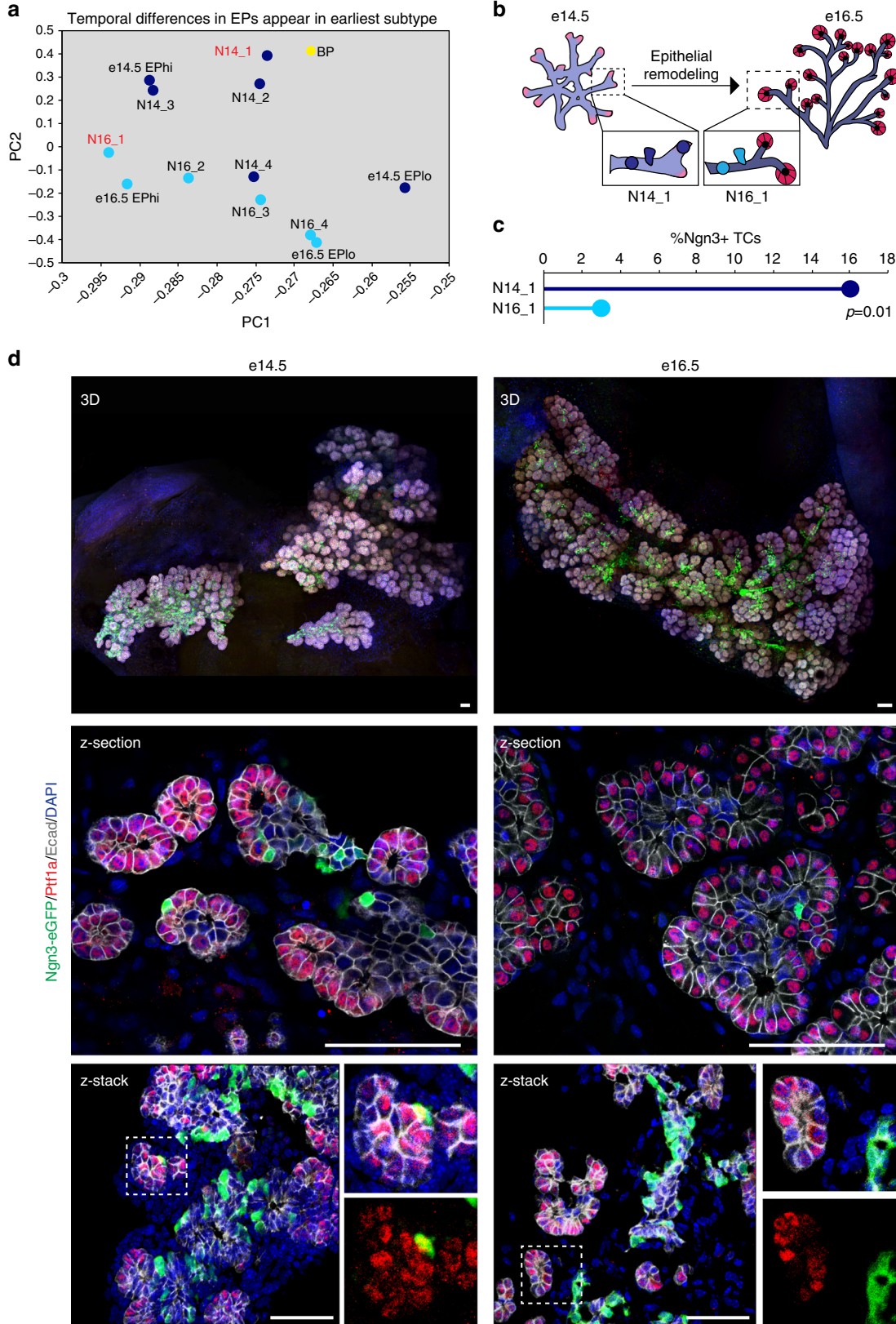

identical to e16.5 beta cells. In sum, these findings can be utilized to identify the precise route to mimic EP maturation from N16_1 to N16_4 before beta cell differentiation (Supplementary Fig. 18b).

## Discussion

In this study, we found four subtypes of *Ngn3* + cells at e14.5 and e16.5 during the peak of *Ngn3* + EP formation. Rather than reflecting different endocrine cell types, these subtypes reflect

**Fig. 7** Endocrine progenitors are induced from both epithelial tip and trunk domains at e14.5 but segregated to the trunk by e16.5. **a** Principle component analysis of e14.5 and e16.5 *Ngn3* + cell subtypes and original *Ngn3* + cell clusters. As a control, e14.5 BPs are shown in yellow. **b** Diagram showing changes in the structure of the pancreatic epithelium over time. Tip cells are marked by pink/red while trunk regions are marked by purple. **c** Barcode tracing of the earliest *Ngn3* + cell subtypes at e14.5 and e16.5 (N14_1 and N16_2) revealed 16% of e14.5 *Ngn3* + cells are induced from the tip compared to only 3% of e16.5 *Ngn3* + cells. *p* = 0.01, Fisher's exact test. **d** 3D imaging of whole pancreas reveals *Ngn3* + Tip cells at e14.5 but not e16.5. Ngn3-eGFP marks EPs (in green), Ptf1a marks Tip cells (in red), E-cadherin marks epithelium (E-cad in white), and nuclei are shown by DAPI in blue. Individual images were tiled together to produce a final image of the entire organ in 3D. Top, scale bar = 100 μm. Middle and bottom, scale bar = 50 μm. See also Supplementary Figure 16

maturation of *Ngn3* + cells from induction, delamination, and differentiation into islet cells. Delineating the genes enriched in different stages of EP maturation is of particular interest, as regulators of EP maturation are far from understood. Defining the stages of EP differentiation also gives a glimpse into the processes occurring in each stage and may lend to the discovery of factors or pathways that regulate these transitions. We also detected very few (<3%) *Ngn3* + hormonal cells in late EP clusters (N14_4 and N16_4). These cells likely represent the earliest born hormonal cells, which either transiently co-express *Ngn3* alongside hormones or post-transcriptionally repress *Ngn3* from being translated into protein, and these data may help to identify genes involved in initiation of hormone expression. Moreover, looking for genes with similar fluctuation patterns like *Ngn3* may help to identify factors contributing to their diversity or controlling EP maturation stages, some of which are detailed herein, with complete gene lists provided at depository GSE100622 and in Supplementary Data sets.

We found that *Ngn3* + cells are constantly born and while e14.5 and e16.5 *Ngn3* + cells shared select genes, these *Ngn3* + cells are overall transcriptionally distinct. As a consequence, e16.5 *Ngn3* + cells have a higher propensity to form beta cells. Thus *Ngn3* + cells cannot all be treated as the same. *Ngn3* by itself is sufficient to determine endocrine commitment, however it might not be sufficient to distinguish early stages (such as N16_1) from later stages that are more equipped for differentiation into hormonal cells (such as N16_4) or to discern between N14 (alpha cell prone) and N16 (beta cell prone) *Ngn3* + populations. This is of particular importance for in vitro directed differentiation efforts to produce specific pancreatic endocrine cells. For the beta cell, comparing in vitro derived EPs to the transcriptional profiles of e16.5 *Ngn3* + cells and against e14.5 *Ngn3* + cells would likely improve efficiency, and we find that some hPSC-derived Ngn3-eGFP + cells[57] resemble e16.5 *Ngn3* + cells. The temporal differences in *Ngn3* + cells are reminiscent of neural progenitors in the developing cortex, which also exhibit different competencies for different neural fates as development progresses[59]. Therefore, the heterogeneity between stages in *Ngn3* + cell identity may be a phenomenon not restricted to progenitors in the pancreas, but rather globally applicable to many different progenitors during development.

The question remains as to what mechanisms are controlling EP changes over time. Pancreatic epithelial cells remain in a plexus state until transformation between e14.5 and e16.5 into an arborized ductal system. The EP changes and shift in the formation of alpha cells to beta cells parallels the morphogenetic changes as the epithelium is remodeled, with the epithelial transition nearly complete by e16.5[17]. Further, timely induction of EPs is critical to prevent defects in pancreatic epithelial development[17,60]. Prior to remodeling, *Ngn3* + cells originate either from MPCs (around e9.0) or tip cells (prior to e14.5) while after the plexus-to-duct transition a shift in competency leaves trunk progenitors as a sole the source of *Ngn3* + cells. We observed that tip cells are still competent to give rise some *Ngn3* + at e14.5 but this ability is almost completely lost at e16.5. Very

few *Ngn3* + cells do not becoming endocrine cells[12], thus these *Ngn3* + tip cells likely contribute to the endocrine pool. The majority of *Ngn3* + cells, however, come from bipotent trunk epithelium, and we find differences between e14.5 and e16.5 BPs. It is plausible that the changing epithelium is tied to changes in the genome and epigenome and thus the formation of distinct EPs at each stage. While the epigenome clearly changes in EPs from e14.5 to e16.5, it will be interesting to determine how these changes are related to the morphogenetic shifts in the epithelium and whether these chromatin shifts are instructive to temporal changes in EPs or if these are merely consequences of the changing transcriptome.

Another potential mechanism that regulates the temporal changes in EPs is changes in the surrounding niche. This is an attractive hypothesis, as mesenchyme, neural, and endothelial cells play important roles in endocrine cell formation[61]. Our data show extensive heterogeneity of mesenchyme subtypes at both e14.5 and e16.5. Further studies on pancreatic niche heterogeneity will aid in understanding of paracrine signaling, providing a framework to coax stem cells efficiently to become beta cells. In addition, understanding these signals could shed light onto how EPs progress through each maturation stage, for example what factors promote N14_1 cells to delaminate from the epithelium and progress into N14_2. The resources provided herein can be used to dissect which pathways are upstream of these transitions regulating EP maturation. Previous studies have shown the import of FGF, Notch, Hedgehog, and TGFb signaling in EP differentiation, but whether these pathways promote specific subtypes of *Ngn3* + cells in unknown. Further, we observed by GO analysis of kinetic clusters enrichment of Slit/Robo, Met, PDGF, and PI3K/Akt signaling in early *Ngn3* + cells while later *Ngn3* + cells had enrichment of calcium, Mapk, cAMP, and adrenergic signaling. Finding how niche cell subtypes are spatially arranged in the developing pancreas and what signals they provide to different epithelial domains will offer clues into how epithelial patterning, compartmentalization, and morphogenesis occurs.

An important question in the pancreatic field is the timing in which endocrine cell fates are determined; whether the decision of EPs to become alpha or beta cells occurs after *Ngn3* expression or even earlier, before endocrine commitment[35]. Here, we present evidence that the choice might be made prior to *Ngn3* expression, although more investigation is required to conclusively determine this. We found that in the epithelial progenitor state, even the earliest *Ngn3* + cells were different (N14_1 and N16_1) and that the underlying epithelium (BPs) was also temporally divergent. Interestingly, by screening factors enriched in e16.5 BPs compared to e14.5 BPs, we found *AMOTL2*, and its knockdown in hESC-derived PPs led to increased *GCG* and decreased *INS* expression, suggesting reversion to e14.5 BPs. It has been shown that fate allocation for the three major pancreatic lineages—duct, acinar, and endocrine—may be primed in multipotent progenitor cells (MPCs), with heterogeneity of these cells and some exhibiting unipotency for the endocrine fate at e9.5[62]. The effect of *AMOTL2*

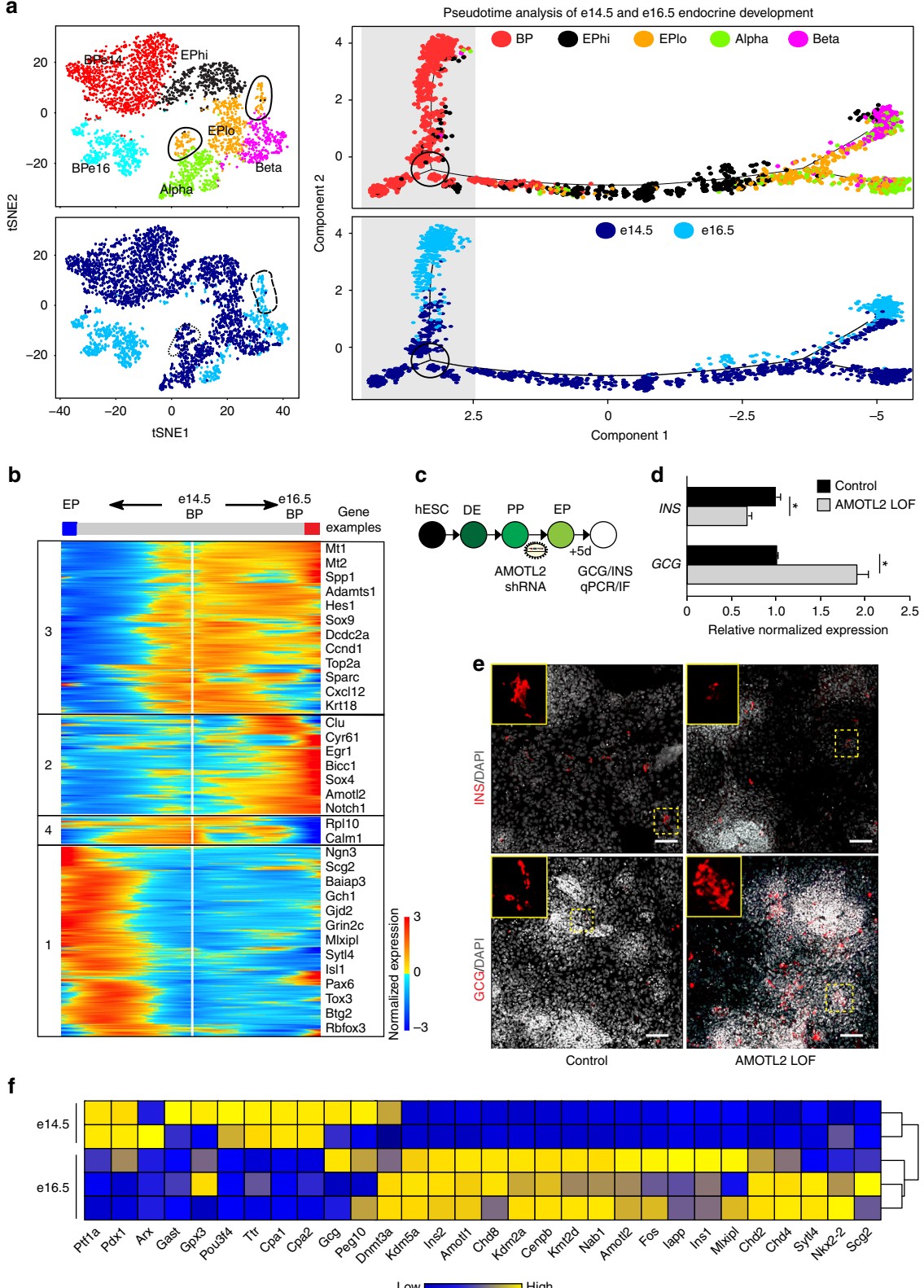

on endocrine lineage decisions indicates that alpha or beta cell priming might occur in BPs, and future studies to detail the clonality of BPs would shed light onto this process.

Using pseudotime analysis of progenitor fate decisions, we created transcriptional maps of the directions progenitors take to become alpha or beta cells and revealed potential regulators of

these choices. Our current data indicate that progenitor heterogeneity is conserved in humans. The fact that different subtypes of human EPs might exist has implications for ongoing efforts to efficiently generate functional beta cells in vitro. For example, identification of progenitor subtypes produced in vitro may lead to more efficient formation of desired endocrine cells, such as

**Fig. 8** Bipotent progenitors change with more alpha and tip cell genes expressed at e14.5 and more beta cell genes expressed at e16.5. **a** Pseudotemporal ordering of merged e14.5 and e16.5 single cells revealing the developmental trajectory of BPs as they differentiate. On left, tSNE plot of merged e14.5 and e16.5 single cells in clusters (top) and with original identities (bottom). On right, BPs branch depending on developmental stage of origin, with e14.5 BPs making a choice between maturation into e16.5 BPs or differentiation into endocrine cells (clusters on top; original identities on bottom). **b** Branched expression analysis modeling (BEAM) showing transcriptional kinetics as e14.5 BPs either mature to e16.5 BPs or differentiate into EP, alpha, or beta cells. Examples of genes from each cluster are shown to the right, including *Dcdc2a*, which is conserved in both e14.5 and e16.5 BPs, and *Amotl2*, which is increasing as e14.5 BPs mature into e16.5 BPs. **c** Schematic for *AMOTL2* knockdown in hESC-derived EPs. Lentivirus with pGIPZ *AMOTL2* shRNA was transduced into hESCs at day 0 of the EP stage and cells were collected for analysis after 5 days in basic DMEM/B27 media. **d** AMOTL2-LOF cells resemble the e14.5 phenotype, with more GCG and less INS. qPCRs showing *INS* and *GCG* expression 5 days after loss of *AMOTL2* in hESC-derived EPs. Samples are normalized to *TBP*. N = 3 biological replicates. Scale bars are SEM, *$p < 0.05$ by Student's *t*-test. **e** Immunostaining of INS, GCG, and CHGA in control and *AMOTL2*-LOF hESC-derived EPs. Scale bar = 100 μm. **f** Average cluster log2 *z*-score normalized heatmap of select beta, alpha, and tip cell markers in e14.5 and e16.5 BPs. See also Supplementary Figure 17

*Ngn3* + e16.5-like EPs which have a higher potential for the beta cell fate. Our datasets provide a framework for evaluating different stages of ESC-derived pancreatic differentiation for a variety of protocols. For instance, this data can be used to train machine-learning tools for prototype scoring of in vitro derived cells, comparing them to our in vivo cells throughout a differentiation trajectory[63]. scRNA-seq analysis of the developing human fetal pancreas would provide further insights into human endocrine differentiation.

As scRNA-seq studies have been performed in postnatal beta cells[64,65], our findings can be coupled with this to create a developmental atlas spanning a broad time spectrum. Adult mouse islets are composed of 60–80% beta cells and 15–20% alpha cells, while we find ~28% alpha cells and 41% beta cells out of all endocrine cells in the e16.5 pancreas. Thus, merging the embryonic and postnatal datasets would be appealing to determine if there are further shifts in fate as cells mature leading to the adult composition. Moreover, it would be interesting to identify sets of genes expressed when beta cells are differentiating and which are maintained in expression in mature beta cells, as many developmentally critical factors have been shown to regulate beta cell physiology and dysregulation of these factors can lead to diabetes.

## Methods

**Animals**. Animal studies were approved by the Baylor College of Medicine Institutional Animal Care and Use Committee. All mice were on mixed background and were housed at Baylor College of Medicine Animal Facility. Ngn3-eGFP knock-in mice were generously provided by Dr. Klaus Kaestner. Transgenic Sox9-eGFP mice were made by Dr. Haruhiko Akiyama. Sox9-Cre[ERT2] were originally made by Dr. Maike Sander and were obtained from The Jackson Laboratory (stock#018829), ROSAmTmG were originally made by Dr. Liqun Luo and obtained from The Jackson Laboratory (stock#007676). Wild-type mice were ICR. The mice were housed at 22–24 °C with a 12 h light/12 h dark cycle with standard chow (Lab Diet Pico Lab 5V5R, 14.7% calories from fat, 63.3% calories from carbohydrate, 22.0% calories from protein) and water provided ad libitum. Genotyping was done using the HotStart Mouse Genotyping Kit with its instructed PCR reaction set up (KAPA Biosystems). To genotype Ngn3-eGFP mice, three primers were used (F-ATACTCTGGTCCCCCGTG, R-TGTTTGCTGAGTGCCAACTC, Ngn3-GAACTTGTGGCCGTTTACGT) with 60 °C annealing temperature, yielding 300 bp wild type and 180 bp GFP band. To genotype Sox9-eGFP, two primers were used (F-CGACGGCAACTACAAGACCC and R-GGGTGCTCAGG-TAGTGGTTG) with the expected band size of 304 bp and no band present in wild-type animals.

To genotype Sox9-Cre[ERt2] mice, two primers were used (F-GAACGCACTGATTTCGACCA and R-AACCAGCTTTTCGTTCTGC) with 53 °C annealing temperature, yielding 200 bp GFP band.

To genotype ROSA-mTmG mice, three primers were used (common F-CTCTGCTGCCTCCTGGCTTCT, R- CGAGGCGGATCACAAGCAATA, and CAG R- TCAATGGGCGGGGGTCGTT) with 58 °C annealing temperature, yielding 320 bp wild type and 250 bp EGFP-L10a bands.

To pulse chase Sox9 + cells, Sox9-CRE[ERT2]; ROSAmTmG mice received single intraperitoneal injection of tamoxifen (4-OHTm, Sigma #T5648, dissolved in corn oil) at 6 mg/40 g of body weight, at e13.5 or e15.5. Pancreas was analyzed 48 h later.

For all experiments, both male and female embryos were analyzed.

**Droplet-based RNA-seq**. Timed pregnant ICR mice were dissected at 10:00 in the morning on day 14.5 and day 16.5. We performed Droplet-based RNA-seq in three batches from e14.5 ICR three litters for a total of 39 pancreata at an average depth of 2957.2 unique molecular identifiers (UMIs) per cell and in two batches from two e16.5 litters for a total of 21 pancreata. The pancreas was dissected and pooled together before dissociation in TrypLE (Invitrogen) at 37 °C for 30 minutes with agitation. Cells were quenched in FACS buffer (2% FBS, 10 mM EDTA in PBS) and filtered through a 40 μm cell strainer (BD Biosciences). Dissociated cells were diluted to a concentration of 200 cells per μL in PBS with 0.01% BSA. RNA-seq was then performed according to Macosko et al.[30]. Here cells were co-encapsulated into nano-liter sized droplets containing barcoded microparticles (ChemGenes, catalog number Macosko201110) and lysis buffer using a custom microfluidics device (FlowJEM, Toronto, Canada). After droplet breakage, reverse transcription, and exonuclease treatment all cDNA was PCR amplified, pooled, purified with Ampure XP beads (Beckman Coulter), and ran on a Fragment analyzer (Advanced Analytical Technologies, Inc.) for quality control, quantification, and size determination. Library preparation was performed with the Illumina Nextera XT kit. All libraries were sequenced on an Illumina NextSeq500 instrument.

**Droplet-based RNA-seq Data Analysis**. Sequencing data were handled as described in Shekhar et al.[66]. First, FastQ files were converted to BAMs with Picard tools (MergeSamFiles) and then used is input for STAR alignment, cell barcode correction, and digital gene expression (DGE) matrix generation via the Droplet-based RNA-seq tools software package (available at http://mccarrolllab.com/dropseq/). The minimum gene per cell threshold was set to 500 for inclusion into the final digital expression matrix. Subsequently, DGEs from each experiment were merged and then we imported the comprehensive DGE into Seurat (version 1.4.0.5) where normalization was performed according to package default settings. Batch effects were corrected for by regressing out the number of molecules per cell and the percentage of mapped mitochondrial reads with the RegressOut function (Seurat package). Next, principle components analysis (PCA) was performed and significant PCs were used as input for graph-based clustering and 2-dimensional visualization of the multi-dimensional data set was done with tSNE. Differential expression of the individual clusters was performed using the likelihood-ratio test for single-cell gene expression. To account for over-clustering, clusters that were not transcriptionally distinct were merged, including clusters 0 and 2 as well as clusters 19 and 25 (Fig. 1e). Multikernel-based clustering of e14.5 Droplet RNA-seq data was conducted with SIMLR (version 1.2.1). The approximate cell cycle phase of each cell was calculated using Seurat by scoring individual cells on their expression for S-phase, G1, and G2M genes as defined by Kowalczyk et al.[31]. Ngn3 + cell subclustering was performed using subset data functions in the Seurat package with a 0.1 accept.low cutoff. Gene ontology analysis was performed using DAVID Bioinformatics Resources 6.8[67,68] and Metascape (metascape.org)[69]. For pseudotemporal analysis, the normalized data from the indicated clusters calculated in Seurat was then passed directly into Monocle2. The Monocle2 BEAM statistical test was utilized to isolate the branch-specific gene expression patterns (256 genes with *q*-val < 0.01). For the visualization of bifurcating endocrine trajectories, Droplet RNA-seq DGEs containing the specified cells were transposed and used as input into Wishbone, where PCA was performed, followed by tSNE visualization, and finally Wishbone analysis (version 0.4).

**Immunostaining**. Whole pancreas was fixed in 4% paraformaldehyde for 2–16 h, PBS washed, incubated in 30% sucrose overnight at 4 °C, and embedded in O.C.T. compound for immunofluorescent staining. The cells were briefly washed with PBS and then fixed with 4% PFA in PBS for 20 min at RT followed by three washes with PBST (PBS + 0.1% Triton-X). The unspecific binding of antibodies was blocked by 30 min incubation with blocking solution (10% donkey serum in PBST) at RT. The primary antibodies (see origin and concentrations used in Supplementary Table 1) were in blocking solution for 16 h at 4 °C with shaking and then cells were washed three times with PBST for 10 min. The secondary antibodies were conjugated with appropriated Alexa Fluor Dye (Jackson ImmunoResearch Laboratories), diluted

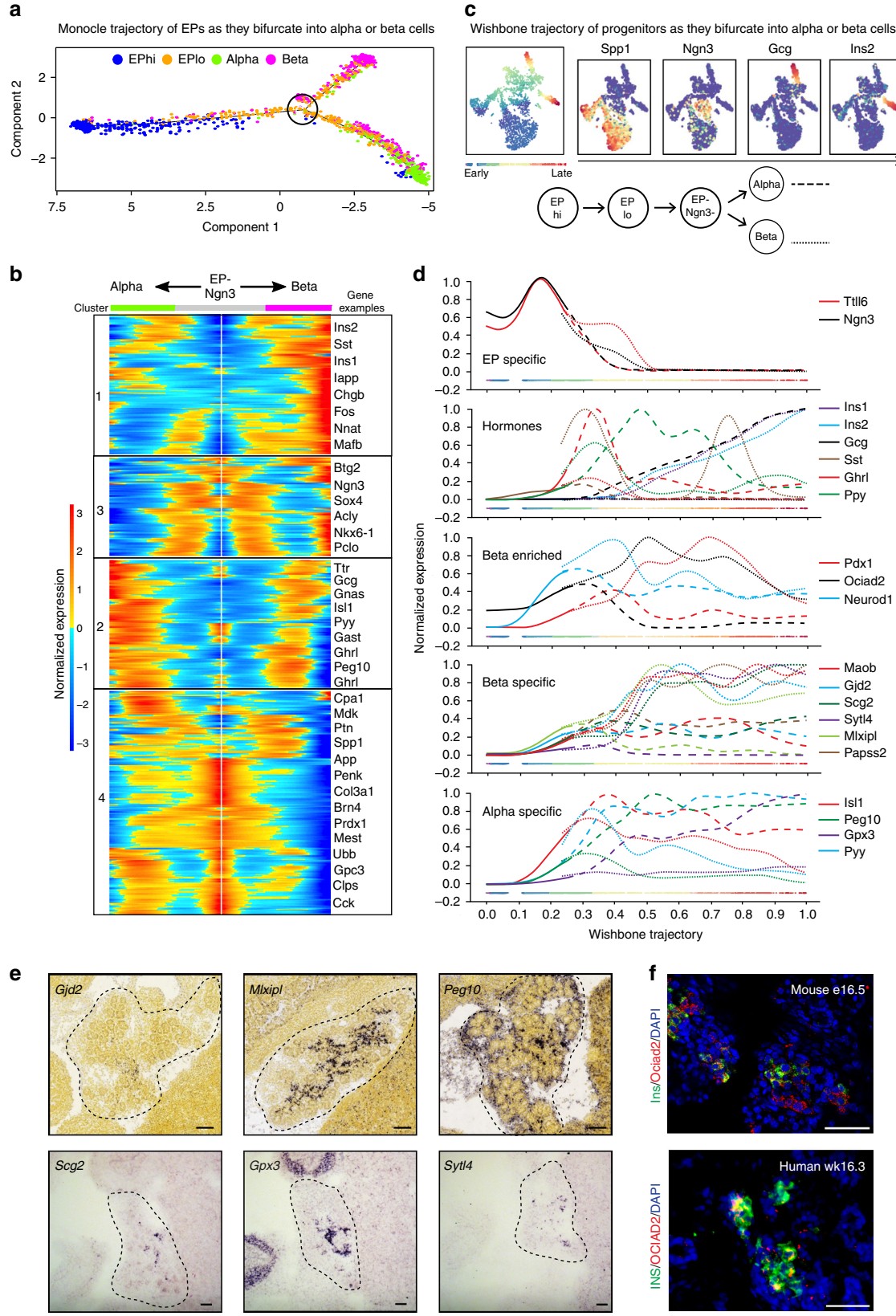

with blocking solution and incubated with specimens for 1 h at RT and then washed three times with PBST. Nuclei were stained with DAPI (Invitrogen). Slides were mounted in Fluoromount G (SouthernBiotech), covered with coverslips, and sealed with nail polish. All primary antibodies and dilutions are listed below. 6D6 antibody was deposited to the DSHB by Pringle, G.A. (DSHB Hybridoma Product

6D6). Imaging was performed on Leica CTR DM6000 FS, Zeiss Axioplan 2, and Zeiss 710 Confocal. For optical clearing, tissues were dissected before fixing and immunostaining in 3D, blocked overnight at 4 °C with primary incubated for 3 days at 4 °C followed by 2 days at 4 °C in secondary antibody. Following immunostaining, the tissues were put through a fructose gradient as described by

**Fig. 9** Transcriptional blueprints of endocrine progenitor differentiation into alpha or beta cells. **a** Pseudotemporal ordering of merged e14.5 and e16.5 single cells using Monocle2 revealed the developmental trajectory of EPs as they differentiate. Branch point shows EPs at the decision point for the alpha or beta lineages. **b** BEAM showing transcriptional kinetics as EPs differentiate into either alpha or beta cells. Examples of genes from each cluster are shown to the right. **c** Pseudotemporal ordering of single cells using Wishbone. Colors on far left panel indicating early (blue) to late (red) cells. Four right panels show examples of genes to verify the temporal associations, including early marker *Spp1*, followed by *Ngn3*, then bifurcation into either *Gcg* or *Ins2* expressing cells; red indicates high and purple low expression levels. **d** Projection of gene expression of select genes of interest. Dashed lines indicate alpha cells while dotes lines indicate beta cells. **e** RNA in situ of select alpha enriched genes (*Peg10*, *Gpx3*) and beta enriched genes (*Gjd2*, *Mlxipl*, *Scg2*, *Sytl4*). *Peg10*, *Gjd2*, and *Mlxipl* images are from e15.5 Allen Brain Atlas. Gpx3, Scg2, and Sytl4 are from e14.5 mouse. Scale bars = 198 µm. **f** Ociad2 (red) is co-expressed with Ins (green) in the developing mouse and human pancreas. Nuclei are marked by DAPI, in blue. Scale bars = 50 µm. See also Supplementary Figure 18

Ke, et al.[52]. until mounting in 3D for confocal images. Images were taken at ×20 magnification before tiling together to create the 3D image. Scale bars and quantifications were performed using ImageJ and Fiji.

**RNA in situ hybridization**. Non-radioactive in situ hybridization for *Isl1*, *Scg2*, *Sytl4*, and *Gpx3* mRNA was performed by fixing slides in 4% PFA, followed by PBS washes, proteinase K incubation and another 4% PFA incubation. After 5 min acetic anhydride incubation, the slides were incubated in hybridization buffer for 1 h at 65 °C, before overnight incubation with antisense RNA. Next day, slides were washed first in 2xSSC at 65 °C then in PBST. Tissue was blocked in PBST supplemented in sheep serum, then incubated with anti-Digoxigenin antibody coupled with alkaline phosphatase (Roche). After PBST washes, staining was developed in alkaline phosphatase buffer complemented with BCIP and NBT (Roche). Where indicated, RNA in situ was derived from the Allen Brain Atlas (http://www.brain-map.org/).

**Ngn3-eGFP + cell isolation**. Timed pregnant ICR mice were dissected in the morning on day 14.5 and day 16.5. The pancreas was dissected and dissociated in TrypLE (Invitrogen) at 37 °C for 30 min with agitation. Cells were quenched in FACS buffer (2% FBS, 10 mM EDTA in PBS) and filtered through a 40 µm cell strainer (BD Biosciences) before FACS using FACSAria (BD Biosciences). Gates were set with reference to negative controls.

**Flow cytometric analysis**. The pancreas was dissected before dissociation in TrypLE (Invitrogen) at 37 °C for 30 minutes with agitation. The cells were quenched in FACS buffer (2% FBS, 10 mM EDTA in PBS) and filtered through a 40 µm cell strainer (BD Biosciences). The cells were stained using transcription factor staining buffer kit (Invitrogen). The cells were incubated in fixation and permeabilization buffer on ice for 15 minutes, washed with wash buffer, and stained with primary antibodies for 1 h on ice in wash buffer. After another wash, the cells were stained with secondary antibodies for 1 h on ice in wash buffer. The cells were spun down, resuspended in FACS buffer (3% FBS in PBS), and analyzed on LSRFortessa II (BD Biosciences). Data were analyzed with FlowJo software (Tree Star Inc.).

**qPCR**. For qPCR, total RNA was isolated using Trizol (Invitrogen). The RNase-free DNAse treatment was used to remove any traces of genomic DNA according to the manufacturer's protocol (Qiagen). A volume of 1 µg of RNA was used for reverse transcription using iScript (Biorad). 1/20 of cDNA was used for PCR using SYBR Green (KAPA Biosystems) and a Connect CFX light cycler (Biorad) (≤40 cycles). Primers were designed to amplify across the exon junctions using qPrimerDepot and Primer3 software. PCR product specificity was verified by gel electrophoresis. Threshold data were analyzed in CFX Manager Software v3.1 (Applied Biosystems) using the Comparative Ct relative quantitation method, with beta-actin and TBP as the endogenous controls. All primers were purchased from Sigma Aldrich and sequences are shown in Supplementary Table 2.

**Fluidigm C1**. EP cells were collected from Ngn3-eGFP mice at e14.5 and e16.5 by FACS sorting into a 96-well plate into lysis buffer (9 µL Ambion single-cell buffer + 1 µL DNAse1). The cells were incubated at RT for 5 min. A volume of 1 µL of Ambion Stop reagent was added before incubation for two minutes at RT then transferring to ice. For reverse transcription, 4 µL of 5× VILO reaction mix was added with 2 µL 10× SuperScript to each well. Plate contents were mixed before proceeding with 10 min at 25 °C, 1 h at 42 °C, then 85 °C for 5 min before keeping on ice. For pre-amplification, 25 µL of 2× TaqMan Pre-amplification Master Mix was added with 12.5 µL of 0.2× pooled assay mix. After centrifugation, the pre-amplification reaction was run at 95 °C for 10 min, before 14 cycles of 95 °C for 15 s and 60 °C for 4 min. Next 2.5 µL of 20× TaqMan primers/probe assay (Invitrogen) and 2× Fluidigm assay loading reagent were combined. Then 3 µL of 2× TaqMan Advanced PCR Master Mix was added to 0.3 µL of 20× GE Sample Buffer and 1× STA cDNA. 3.3 µL of sample pre-mix was added to 2.7 µL of each pre-amplification sample. The chip was primed in IFC controller:script—Prime (113 × ) before 5 µL of assay mixes and 5 µL of sample mixes were added to corresponding inlets. Unused inlets were loaded with 3.3 µL of sample pre-mix and 2.7 µL of

nuclease-free water. The chip was run in IFC controller:script—Load Mix (113 × ). Finally, the chip was run via BioMark HD Data Collection software.

**MATQ-seq**. After flow sorting, Ngn3-eGFP + cells were collected in RPMI medium. We used mouth pipette to pick single cells into PCR tubes with MATQ-seq lysis buffer. MATQ-seq was then performed to amplify the whole transcriptome of single cells as published. Briefly, single cells are obtained and lysed individually before reverse transcription, followed by PCR amplification and finally barcoding and pooling of libraries. After second strand synthesis, the input and yield of each sample was quantified by qPCR. Highly degraded samples or samples with multiple cells were discarded. After PCR amplification, we performed library preparation and data analysis according to the previous publication. Duplex specific Nuclease was used to remove the majority of ribosomal cDNA. By MATQ-seq, we had an average of 1,612,788.9 e14.5 and 1,827,836.3 e16.5 uniquely mapped reads ( ± SEM 530,306 at e14.5 and 241,592 at e16.5).

**ATAC-seq**. Approximately 5000–10,000 FACS-sorted Ngn3-GFP + cells were used as input for Fast-ATAC. Fast-ATAC was performed according to Corces, et al.[70]. Briefly, sorted cells were spun down, FACS buffer was removed, and the pellet was then resuspended in a transposase-containing reaction mixture complete with 0.05% digitonin prior to tagmentation at 37 °C with 300 rpm agitation for 30 minutes. Next, transposed DNA was purified with a Qiagen MinElute reaction cleanup kit. Fast-ATAC libraries were generated according to, with a few small modifications. Namely, library DNA was purified with a 1.8X SPR purification using AMPure XP beads following PCR amplification. Paired-end sequencing was performed on an Illumina Nextseq500 instrument.

**ATAC-seq Analysis**. Reads were mapped to the mouse genome (mm9) using Bowtie2 with default paired-end settings. Next, all non-nuclear reads, and unmapped paired reads were discarded. Duplicated reads were removed with picard MarkDuplicates. Visualizations of ChIP-seq signals was done with HOMER (annotatePeaks.pl), and all signals are normalized by read count, where scores represent read count per bp per $1 \times 10e7$ reads. Peak calling was carried out with Homer (findPeaks -style factor) on the merged BAM file, consisting of all e14.5 and e16.5 replicates. Blacklisted regions from mm9 were removed from the comprehensive peak file. Reads were counted for each condition from the comprehensive peak file using bedtools (multicov module). Differential accessibility analysis was carried out using DESeq2 with the multicov file as input. Motif enrichment analysis was conducted with Homer (findMotifsGenome.pl).

**hESC pancreatic differentiation**. hESC H1 line was cultured feeder-free on hESC-qualified Matrigel (BD Biosciences) in E8 media (Stemcell Technologies) with 30% of irradiated mouse embryonic fibroblasts (iMEFs) conditional media. Cells were passaged every 3–5 days at 80% confluent with TrypLE Express (Invitrogen). After dissociation, the cells were plated in E8 media with 10 µM Y-27632, (StemGent) for 24 h. After 24 h media without Y-27632 was replenished daily. iMEF conditional media was prepared by incubating iMEFs with hESC media without bFGF for 24 h for 7 days. Collected media were filtered, flash frozen and stored at −80 °C. To initiate differentiation, the cells were dissociated using TrypLE Express to single cells and seeded at 150,000 cell/cm2 onto 1:30 dilution of growth factor reduced Matrigel (BD Biosciences) in DMEM/F12 in E8-MEF conditional media with 10 µM Y-27632. Two days following seeding the differentiation was started. Day 1 cells were exposed to RPMI + 3 µM CHIR-99021 (Stemgent) + 100 ng/ml rhActivinA (R&D Systems). Days 2–3: + 100 ng/ml rhActivinA + 0.2% FBS. Day 4–5: + 2% FBS + 50 ng/ml KGF (Peprotech). Days 6–9: DMEM/B27 + 50 ng/ml KGF + 2 µM RA (Sigma) + 0.25 µM SANT-1 (Sigma) + 100 ng/ml rhNoggin (R&D Systems). Days 10–14: DMEM/B27 + PdBU (1 µM) + Alk5i (1 µM) + 100 ng/ml rhNoggin (R&D Systems). PPs are defined as Day 9 and EPs as Day 14 of differentiation, respectively.

**Comparison of hPSC in vitro derived pancreatic cells to droplet-based RNA-seq data**. Groups of cells were compared by first cross-referencing shared genes. Next, genes that were not detected across all groups were removed before

constructing dendograms. Gene expression was normalized to the column expression level to prevent technical variations between sequencing and normalizing methods.

**Transduction**. AMOTL2 and scrambled pGIPZ shRNAs were obtained from the Cell-Based Assay Screening Service at Baylor College of Medicine. Two shRNAs per gene were tested in hESC-derived EPs to determine the shRNA with the most efficient knockdown. The scrambled shRNAs were cloned from pGIPZ vector into the doxycycline inducible pINDUCER10 backbone. pGIPZ Amotl2 and scrambled were packaged into lentivirus using the third generation system (PsPax2 and PMD2.G) in low passage (<10) 293 T cells. Lentivirus stock was tested in 293 T cells after concentrating virus. Differentiating hESCs were transduced with lentivirus at the EP stage in DMEM/B27 media for 24 h before changing to DMEM/B27 media alone for five days.

**Statistical analysis**. P-values were calculated as indicated in figure legends using two-sided Student's t-test or Fisher's exact test and noted in the Figure Legends. Data are presented as mean ± SEM and the following symbols are used to represent p-values, $*p < 0.05$, $**p < 0.01$, $***p < 0.005$, and $****p < 0.001$. N represents number of independent experiments.

**Data availability**. The authors declare that all data supporting the findings of this study are available within the article and its supplementary information files or from the corresponding author upon reasonable request.

Raw data have been deposited in the GEO database under accession code GSE100622. Fetal human data is derived from GSE96697. hESC-derived Ngn3-eGFP + cell data is from GSE54879. hESC-derived beta cell data is from GSE61714. hPSC-Ngn3-eGFP + EP stage data derived from GSE54879. Mixed hPSC-EP stage cell data is derived form GSE102877. Adult Islet Nkx6-1 ChIPseq from GSM1006208 was mapped to the mouse genome using Bowtie2 default parameters. Mapped reads were then converted into HOMER tag directories (makeTagDirectory). Neurod1 ChIPseq, e17.5 Beta cell H3K27ac and H3K27me3, and Ngn3-GFP low H3K4me1 ChIP-seqs BedGraphs were downloaded directly from the NCBI gene expression omnibus, gunzipped and then processed into HOMER tag directories at GSE84324.

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

## Acknowledgements

We thank Dr. Klaus Kaestner for Ngn3-eGFP mice and Dr. Richard Behringer and Dr. Haruhiko Akiyama for Sox9-eGFP mice. We would like to thank Dr. Vladislav Sharin and Katrina Wamble for expert technical support. We are grateful to Catherine Gillespie for valuable comments on the manuscript. We are thankful for shared reagents from Dr. Thomas Cooper (Rbfox2 antibody), Dr. Christopher Wright (Ptf1a antibody), Dr. Michael Wegener (Sox4 antibody), Dr Guoqiang Gu and Dr. Maike Sander (Ngn3 antibodies). We also thank Dr. Keith Chen for assistance with the Fluidigm C1 system. We thank Andrew Scavuzzo for invaluable support. This work was supported by the NIH (P30-DK079638 to M.B.; 5T32HL092332-13 to M.A.S. and M.B.; DE023177, HL127717, HL130804, and HL118761 to J.F.M.; and F31HL136065 to M.C.H. and J.F.M.), the McNair Medical Foundation (to M.B.), the Vivian L. Smith Foundation (to J.F.M.), the Transatlantic Network of Excellence Award LeDucq Foundation Transatlantic Networks of Excellence in Cardiovascular Research (14CVD01) "Defining the genomic topology of atrial fibrillation" (to J.F.M.), the confocal core at the BCM Intellectual and Developmental Disabilities Research Center (NIH U54 HD083092 from the Eunice Kennedy Shriver National Institute of Child Health and Human Development), the Cytometry and Cell Sorting Core at Baylor College of Medicine with funding from the NIH (P30-AI036211, P30-CA125123, and S10-RR024574) and the expert assistance of Joel M. Sederstrom, the RNA In Situ Hybridization Core at Baylor College of Medicine, which is, in part, supported by a Shared Instrumentation grant from the NIH (1S10OD016167), and the Cell-Based Assay Screening Service at Baylor College of Medicine (P30 Cancer Center Support Grant (NCI-CA125123).

## Author contributions

Conceptualization: M.A.S., M.C.H., J.F.M, and M.B. Methodology: M.A.S., M.C.H., J.C., D.Y., K.S., Y.K., C.Z., M.B., and J.T. Investigation: M.A.S., M.C.H., J.C., D.Y., J.T., M.B., and K.S. Resources: J.F.M. and M.B. Writing—Original Draft: M.A.S. Writing—Review & Editing: M.A.S., M.C.H., J.C., D.Y., C.Z., J.F.M., and M.B. Funding Acquisition: M.A.S., M.C.H., C.Z., J.F.M., and M.B.

## Additional information

**Competing interests:** The authors declare no competing interests.

