## [Peer Review File · Nature Communications]

Reviewers' Comments:

Reviewer #1:

Remarks to the Author:

The authors generated single-cell RNA-seq, bulk ATAC-seq, and other genomic data from e14.5 and e16.5 mouse pancreata to reveal the regulation of endocrine progenitors (EPs) cell fate. They discovered that e14.5 EPs have higher potential to develop into alpha cells while e16.5 EPs have higher potential to develop into beta cells. The genomic data they generated can be a useful resource for studying pancreatic development and cell fate determination.

Although this is a very interesting work, there are some issues in the data analyses. A major issue is the analyses in different sections of the manuscript are not consistent. The authors should address these issues to give the audience a clearer and more consistent picture of their findings.

Specific points:

1. There are three batches of e14.5 pancreatic cells and two batches of e16.5 pancreatic cells in the single-cell RNA-seq data. Although the authors mentioned they have corrected for batch effects, it will be more convincing to show plots similar to Fig. 1b and Fig. 4a but colored by batches to show that different cell types identified with t-SNE are not affected by batch effects. In Fig. 1b and Fig. 4a, the authors should clarify how the number of significant PCs is selected (e.g., top 22 PCs for e14.5 in Fig. 1b and top 18 PCs for e16.5 in Fig. 4a). Similarly, the authors should also clarify how the number of cell clusters is selected (e.g., 26 clusters in Fig. 1b and 17 clusters in Fig. 4a).

2. In the EP subtype analysis in e14.5 (Fig. 2a), the cells used in the analysis may contain non-EP cells since the authors only used the expression of Ngn3 to filter cells. This is not consistent with their previous analysis (Fig. 1b) where they claimed they identified two EP subtypes (EP_{hi} and EP_{lo}). Therefore, the four subtypes of EP cells may contain artifacts. If the authors want to perform the subtype analysis to EPs, the analysis should be restricted to the two EP cell clusters identified in the cell clustering based on t-SNE. If the authors want to perform subtype analysis to all Ngn3 expressed cells, they should change the statement about "EP subtypes" to "Ngn3+ pancreatic cell subtypes" since it may contain non-EP cells. The same issue applies to the EP subtype analysis in e16.5. The authors should also clarify how the number of cell clusters is selected (e.g., 4 for e14.5 and 4 for e16.5).

3. In the pseudotime analysis of the four EP14 subtypes, Monocle2 generated a very different cell clustering structure (Supplementary Fig. 4a) as compared to the previous analysis using Seurat based on t-SNE (Fig. 2a). For example, EP14_4 cells are located in at least two different branches which contradicts the cell clustering structure with t-SNE (similar cells should locate in a neighboring region). As a result, individual cells in the EP14_4 subtype have very different pseudotime and there are a number of EP14_3 cells located in between the EP14_4 cells. The authors should present a consistent analysis result either defining EP subtypes using the clustering structure in Monocle2 or use other pseudotime analysis algorithms which can construct pseudotime based on t-SNE from Seurat (e.g., the cell clustering structure in Fig. 2a). For example, in the second case (using cell clustering structure in Fig. 2a), the pseudotime could be EP14_2, EP14_1, EP14_3, to EP14_4 which may better represent the maturation process of EP cells since the proliferative capacity analysis (Fig. 3b) show that EP14_2 has higher proliferative capacity than EP14_1.

4. In the comparison of EP14 and EP16 subtypes, why EP14_2 is not included in the hierarchical clustering analysis (i.e., Fig. 4e)?

The authors claimed that EPs are different between e14.5 and e16.5 (Fig. 4f and 4h). They have to test whether the observation is caused by batch effects (e14.5 and e16.5 cells are from different mice and different sequencing runs). For instance, Fig 4f and 4h lack of a control cell type to show

the difference of e14.5 and e16.5 EPs is not caused by batch effects. Such a control cell type could be BPs as shown in Fig. 7a where BPs from e14.5 and e16.5 are partially overlapped. The authors can either mark the EPs in Fig. 7a according to their origins (e.g., from e14.5 or e16.5) or add BPs to Fig. 4f to see if EPs from the two developmental time points are very distinct from each other. Similarly, the authors should also show that the PCA plot in Fig. 6a is not affected by batch effects (e.g., including BPs in the plot may help to clarify). The authors should also clarify how Fig. 6a is generated (it seems to be generated by pooling cells together as pseudo bulk samples). For the gene clustering analysis in Fig. 4i, it seems that genes that are highly expressed in EP16 are also highly expressed in EP14 late. It will be more interesting to see a differential gene expression analysis by comparing EPs from e14.5 and e16.5 to identify differentially expressed genes. In addition, analyzing the differentially expressed genes between subtypes of EPs from e14.5 and e16.5 may also be interesting (e.g., EP14_1 vs. EP16_1, EP14_4 vs. EP16_4). The authors also applied single-cell qPCR to compare EP signature genes between e14.5 and e16.5 in Supplementary Fig. 7a. They concluded that the gene expression pattern is different between e14.5 and e16.5 by showing the heatmap of gene expression. It seems that some cells are actually similar between e14.5 and e16.5. A more convincing way is to show a clustering analysis of all cells from both e14.5 and e16.5 (e.g., hierarchical clustering of all the rows in Fig. 7a) to see if cells from different time points are in distinct clusters. In Supplementary Fig. 7b, the authors used MATQ-seq to show the gene expression of signature genes in EP14_4. However, the plot is not consistent with what they claimed. The plot showed that genes in EP14_3 are highly expressed.

5. It is unclear what cells are used in the analysis of differentiation from EPs to alpha and beta cells in Fig. 8a (e.g., e14.5? e16.5? or merged e14.5 and e16.5?). If merged e14.5 and e16.5 EPs are used in Fig. 8a, this contradicts the previous conclusion that EPs from e14.5 and e16.5 were very distinct.

The developmental trajectory in Fig. 8a may not be a good example since some alpha and beta cells appear to be in the same branch which is not consistent with the t-SNE analysis in Fig. 5g. The cell clustering based on t-SNE in Fig. 5g actually showed a very clear trajectory of cell development from EPhi, EPlo to beta or alpha cells for both e14.5 and e16.5 cells. It will be interesting to see a pseudotime analysis based on this cell clustering structure and identify genes that show different gene expression patterns along the pseudotime for e14.5 and e16.5 respectively.

6. Fold change, p-value, and FDR are not provided in the differential analysis of the ATAC-seq data (e.g., Supplementary Table 8). It is unclear that the ATAC-seq signals in Fig. 5e were normalized or not in the comparison of e14.5 and e16.5. The signals are not directly comparable if the library sizes from e14.5 and e16.5 are different. It will also be interesting to see functional annotation analysis of the differential ATAC-seq peaks between e14.5 and e16.5 (e.g., either use GREAT for the differential peaks or use GO for the neighboring genes of the differential peaks).

7. The Supplementary Tables are not easy to interpret. The authors should provide more information about the Supplementary Tables (e.g., what are the values in each column represent).

Reviewer #2:

Remarks to the Author:

Scavuzzo et al. performed single-cell RNA-seq on mouse pancreata at e14.5 (15,228 cells) and e16.5 (2,006 cells) in order to better characterize endocrine progenitors (EP). By subclustering Ngn3 positive cells, the authors determine that the EP populations are transcriptionally heterogeneous and identify 4 distinct subtypes present at both e14.5 and e16.5. Based on pseudotime ordering, they propose that these subtypes reflect different stages of EP maturation. Furthermore, the authors show that EP at e14.5 and e16.5 are transcriptionally different and

appear to originate from distinct progenitor pools, which underlies their bias towards alpha and beta cells, respectively. Based on these findings, the authors propose that the fate of distinct hormone-expressing lineages are determined before Ngn3 expression in epithelial progenitors (BPs or MPPs). The study is technically of high quality and the results are interesting, especially the finding that developmental timing (although already known, but mechanistically not understood) is important for EP specification into alpha or beta cells.

Major points:

- There is not enough experimental evidence to support some of the main conclusions made by the authors, e.g. the notion that the distinct endocrine lineages are specified prior to Ngn3 expression. Without providing solid evidence for some of the claims the authors should tune down some of the (over)statements.
- Another example is that the authors identify and characterize the putative maturation stages of EPs, but they do not directly demonstrate (other than by the pseudotime analysis) that EPs sequentially progress through these stages. It would be ideal to validate this finding by tracing EPs progression through the four stages, which is technically very challenging. Moreover, it would be interesting to know what factors/pathways are upstream of these putative transitions. This should be discussed.
- The authors analyzed 2006 cells from e16.5, with possibly only a very small fraction being EP (5% at e14.5). It is difficult to draw any conclusions from such a low number of cells (i.e. EP16_1 does not show a tight cluster in Fig 4c and the number of cells analyzed is not even mentioned). It is not obvious why the authors sampled 15.000 cells at e14.5 and only 2000 at e16.5 although material is even better accessible at e16.5?! This might bias the analysis/interpretation and is the only experimental/technical weakness of the study.
- Moreover, other potentially interesting findings, such as the differences in chromatin accessibility between EP at e14.5 and e16.5, are not followed up in order to reveal factors and pathways regulating endocrine specification. This is experimentally challenging, but maybe the authors can discuss some of these points. First, apart from the increased expression of epigenetic regulators, can one show increased activity of these complexes, for example enhanced H3K27me3 or H3K4 demethylation? Second, it would be good to determine whether the chromatin changes are necessary ('drivers') for endocrine specification or are only 'passengers', e.g. by inhibiting epigenetic modulating enzymes. Moreover, how can the authors explain the increase in expression of epigenetic regulators during beta cell specification but not during alpha cell specification?

Minor points:

- GO-Term analysis showed that processes associated with epigenetic regulation were enriched as well in EP14_3/4 (Fig. 3d). However, this does not fit with the heatmap in Supp. Fig. 7c. Also, later the authors somehow imply that this finding is specific to E16.5 EPs.
- An heatmap showing all marker genes and genes changing in pseudotime (and not only selected genes) would be ideal for validation of the clustering and a continuous pseudotemporal progression.
- The pseudotime analysis shows that EP14_1 progresses into two distinct branches: EP14_4 and EP16. Could the authors explain the transition from EP14_1 into EP16? Which maturation stage of EP16 do EP14_1 progress into?
- Fig. 5g: To confirm the single cell RNAseq data, alpha- and beta cells should be counted in sections.
- Supplementary Fig. 1a,b: If the authors want to make any conclusion about the abundance of Ngn3+ cells at different embryonic stages, they need to count the Ngn3+ cells since the qPCR analysis does not give any information about it. Also, the qPCR analysis does not reflect the immunostainings.
- Fig. 1b: Would be informative to include a figure with the actual number of captured EPlo, EPhi, alpha and beta cells (in addition to Supplementary Fig. 2a). The authors claim that EPlo cells bifurcated into alpha or beta cells. Not sure if one can draw such a conclusion from a tSNE plot.
- Fig. 1g: Immunostainings (and selected sections) are not very convincing. It would be good to show single channels. Also, Sox9-GFP seems to be cytoplasmic.
- The authors claim that the higher number of alpha cells captured signifies that at e14.5 alpha

cells preferentially differentiate. However, many alpha cells are formed during the primary transition which could also explain the higher number of alpha cells present at e14.5.

- Figure 3d: Can the authors explain why Ngn3 expression is peaking in late maturation, whereas according to the previous data EP16_4 consists mainly of Ngn3 low cells?
- Supplementary Fig. 2d + Fig. 2e: Immunostainings not convincing. Single channels required.
- Supplementary Fig. 3a,b: No information on flow cytometry analysis in experimental procedure part (staining protocol, antibodies, fluorophores...). In b) the labelling GFP-/GFP+ is missing.
- Supplementary Fig. 4a: Can the authors please explain the monocle ordering of single EPs in particular of the EP14_1 and EP14_2 cells. Why do the EP14_2 cells not map close to EP14_3 cells?
- Fig. 4a: Actual number of captured EPlo, EPhi, alpha and beta cells should be included.
- Fig. 4g: These are very few cells to draw conclusions on cell type distributions and that differences are not due to sampling biases.
- Fig. 4h: It would be interesting to show, where the different subtypes from e14.5 and e16 are in the overall pseudotime analysis. Does it agree with the individual analyses?
- Page 13, line 306 – Suppl. Fig 7b does not show qPCR analysis of Chgb, Chga and Pax6
- Fig.6a and 4e: I am a bit confused in Fig. 4e the authors show that EP14_1 and E16_1 cluster (hierarchical clustering) and in Fig. 6e EP14_1 and E16_1 cluster apart (PC analysis).
- Why is EP14_2 not included in the analysis in Figure 4e and f?
- In Figure 7a it is not clear to me why e14.5 BP branch into beta and alpha cells while the authors stated previously that e14.5 EP give rise predominantly to alpha cells. I don't understand the conclusion of the authors that the BP branches impact fate decisions as the branching occurs before the EPhi cells appear.
- Fig7e: The immunostaining inlet for Ins in the Amotl GOF does not really support the conclusion.
- GO enrichment analysis is not described, please add the type of tests and groups which were compared.
- Computational analyses use standard algorithms. However, quantitative metrics are missing. It would be nice to supplement the qualitative trend observations described in the main text with fold changes (e.g. initial to maximal) or p.values for differential expression
- The description/discussion of the results is very often difficult to follow. There are also many sentences in which words are missing e.g. page 5 line 108-111.

Reviewer #3:

Remarks to the Author:

Endocrine lineage biases arise in temporally distinct endocrine progenitors during pancreatic morphogenesis

In this work, the authors performed single-cell RNA sequencing of the pancreas at e14.5 and e16.5, when the majority of pancreatic EPs appear. They show that at each of these developmental stages four subtypes of EPs are identified, reflecting different maturation stages. New EPs are being born at both e14.5 and e16.5, and specific markers for the different subtypes maintain in both stages. Despite these commonalities, the EPs from e14.5 are inherently different than the EPs in e16.5 in regards to gene expression, and cluster separately. Importantly, the data suggests that EPs from e16.5 have higher potential for beta cell differentiation than EPs from e14.5. The authors also perform ATAC-seq on purified Ngn3 positive EPs from e14.5 and e16.5, and show different chromatin accessibility. This is not surprising considering their previous finding that these EPs are transcriptionally distinct. The e16.5 ATAC-seq maps reveal enrichment for beta cell motifs, also supporting their conclusions of a bias towards beta cell differentiation of these EPs. Over all, the authors present a large body of evidence to support their conclusions, which would be of great interest to the community, and may be a valuable resource for future studies.

Specific comments:

1. The abstract is not well written to reflect what is shown in the paper. It should better reflect what was done in this work, and not the possible interpretation of the results (similar to a discussion). For example: The first sentence is too general and not specific to this work. In the sentence: "Thus fate determination occurs before Neurogenin3 expression, affecting the chromatin state and transcriptome of EPs at each stage"- the causality of the change in fate determination that affects chromatin and transcription is not shown here- it is an interpretation of the results. Also claiming to present a transcriptional and epigenetic atlas of the pancreatic endocrine development is an over statement.
2. Page 6: the authors mention specific genes such as Btbd17, Grasp, Fev and St18 that were not previously associated with EPs. It would be informative to mention what are these genes (do they have known roles in other pathways?) and why these genes were specifically chosen here. Are these the most differentially expressed? Or is there a biological rationale? The same questions also apply for Dcdc2a- why did you choose to follow up specifically on this gene?
3. Fig 1G: it seems in the image that Dcd2a is expressed in more cells than Sox9, is that indeed the case? Doesn't it contradict the expression data in figure 1F?
4. Is it possible to do IF for some of the differentially expressed genes in EPhi and EPlo, in combination with Ngn3-GFP, in order to validate expression of these genes is indeed associated with low/high levels of Ngn3?
5. Figure 2: In order to validate their results of the different subtypes, it would be helpful to stain for two proteins in the same cells. For example, it would highly support their conclusions if they can show with IF that Hes1 is found in the same cells with Vim, but not in cells that express Neurod1. Also, it might indicate on the spatial distribution of the different subtypes. Adding some statistics (such as what percentage of cells express different markers according to IF) would also be helpful.
6. In the epigenetic analysis, the authors conclude that there is a chromatin accessibility remodeling between e14.5 and e16.5 that leads to a more permissive environment for beta cell formation. They connect it to the changes seen in the expression of chromatin remodelers (Fig 3). However, the causality is not clear here, and I would refrain from over interpretation of the results (especially without any functional studies to support it, such as perturbations to chromatin regulators, which I believe is out of scope for this work). As the EPs from e14.5 and e16.5 were already shown to be transcriptionally distinct, it makes sense that ATAC-seq maps are different. The authors suggest that changes in chromatin lead to changes in expression, but it is not necessarily the case (it is very possible that changes in chromatin accessibility merely represent the changes in gene expression). In that regard, it would be interesting to further analyze some of the beta-specific TFs that are indicated in figure 5d. For example- we see enrichment for Nkx6.1 motifs in the e16.5 EPs. Are there changes in the levels of Nkx6.1 itself between e14.5 and e16.5? Is it possible to do ChIP-seq for this TF to see where it is bound in the two stages?

Minor comments:

1. Page 3 line 61: "only" appears twice.
2. Page 8 line 179: Vim is expressed also in EP14_2 and not just EP14_1.
3. Page 16 line 393: delete "observed".

Reviewer #1 (Remarks to the Author):

The authors generated single-cell RNA-seq, bulk ATAC-seq, and other genomic data from e14.5 and e16.5 mouse pancreata to reveal the regulation of endocrine progenitors (EPs) cell fate. They discovered that e14.5 EPs have higher potential to develop into alpha cells while e16.5 EPs have higher potential to develop into beta cells. The genomic data they generated can be a useful resource for studying pancreatic development and cell fate determination.

Although this is a very interesting work, there are some issues in the data analyses. A major issue is the analyses in different sections of the manuscript are not consistent. The authors should address these issues to give the audience a clearer and more consistent picture of their findings.

Thank you for your careful analysis of the manuscript. We agree that the datasets included can be a useful resource to others in the field. We have followed your suggestions below to maintain consistency throughout the manuscript. We believe these changes have made the findings more clear to future readers and have overall made the manuscript much stronger.

Specific points:

1. There are three batches of e14.5 pancreatic cells and two batches of e16.5 pancreatic cells in the single-cell RNA-seq data. Although the authors mentioned they have corrected for batch effects, it will be more convincing to show plots similar to Fig. 1b and Fig. 4a but colored by batches to show that different cell types identified with t-SNE are not affected by batch effects.

We previously had provided batch effects data for *Ngn3+* cell subtypes at e14.5 (now in Supplementary Fig. 4b), however we agree it is important to show the global batch effects for the plots in Fig. 1b and 4a. Therefore we have included this data in Supplementary Fig. 2c and Supplementary Fig. 11a. While batch effects do not affect *Ngn3+* and endocrine cell types, we do find increased presence of mesenchyme in the first batch. We also note that most of the few hepatocytes were also derived from batch 1, thus these mesenchyme subtypes are likely derived from including more of the surrounding tissue in the first dissection that was removed in the second and third batches. Therefore we have added the following sentence in the manuscript,

“We found equal representation of cells from all three batches in every cluster, with the exception of three mesenchyme clusters and a cluster of hepatocytes composed mostly of batch 1 cells (Mes2 cluster 1; Pr. Mes2 cluster 6; Mes3 cluster 8; Hepato cluster 27), likely due to increased inclusion of surrounding tissue during the first dissection (Supplementary Fig. 2d).”

In Fig. 1b and Fig. 4a, the authors should clarify how the number of significant PCs is selected (e.g., top 22 PCs for e14.5 in Fig. 1b and top 18 PCs for e16.5 in Fig. 4a). Similarly, the authors should also clarify how the number of cell clusters is selected (e.g., 26 clusters in Fig. 1b and 17 clusters in Fig. 4a).

To select meaningful PCs for inclusion into scRNA graph-based clustering we looked at the standard deviation of each principle component in Seurat (PCElbowPlot), and then determined a cutoff based on this graph (point at which elbow is clear, see http://satijalab.org/seurat/pbmc3k_tutorial.html). Next, PCs near cutoff were inspected to determine if meaningful genes were present (non-mitochondrial and non-ribosomal). Finally, if

the PC at the cutoff had enrichment for relevant markers, it was used as the cutoff for graph-based clustering and tSNE visualization.

Graph-based clustering was performed with Seurat (FindClusters function). To identify all distinct clusters we used a high-resolution parameter (resolution = 1-2) in order to over-cluster the data. Next, we performed differential expression analysis on all spatially proximal clusters. Clusters without significant transcriptional distinctions were merged together and then classified based on the expression of known markers. Clusters that were transcriptionally distinct were kept intact. We have ensured this is noted in the methods section:

“Next, principle components analysis (PCA) was performed and significant PCs were used as input for graph-based clustering and 2-dimensional visualization of the multi-dimensional data set was done with tSNE. Differential expression of the individual clusters was performed using the likelihood-ratio test for single cell gene expression. To account for over-clustering, clusters that were not transcriptionally distinct were merged, including clusters 0 and 2 as well as clusters 19 and 25 (Fig. 1e).”

2. In the EP subtype analysis in e14.5 (Fig. 2a), the cells used in the analysis may contain non-EP cells since the authors only used the expression of *Ngn3* to filter cells. This is not consistent with their previous analysis (Fig. 1b) where they claimed they identified two EP subtypes (EPhi and EPlo). Therefore, the four subtypes of EP cells may contain artifacts. If the authors want to perform the subtype analysis to EPs, the analysis should be restricted to the two EP cell clusters identified in the cell clustering based on t-SNE. If the authors want to perform subtype analysis to all *Ngn3* expressed cells, they should change the statement about “EP subtypes” to “*Ngn3*+ pancreatic cell subtypes” since it may contain non-EP cells. The same issue applies to the EP subtype analysis in e16.5. The authors should also clarify how the number of cell clusters is selected (e.g., 4 for e14.5 and 4 for e16.5).

We understand this terminology can be confusing, therefore we have changed the *Ngn3*+ cell subtypes from EP14_# to N14_# (i.e. EP14_1 is now N14_1). We have also changed the e16.5 *Ngn3*+ cell subtypes to this terminology to avoid confusion and refer in the text to these only as *Ngn3*+ cell subtypes.

In the analysis of *Ngn3*+ cells in Fig. 2 and Fig. 4, we used a threshold of *Ngn3* expression to select cells for subclustering. We found this method powerful as we then include cells that are highly expressing markers related to other cell types that cluster apart from the EPhi and EPlo cells in Fig. 1b. For example, the N14_2 subtype has high expression of EMT markers as these *Ngn3*+ cells delaminate from the epithelial layer into the surrounding niche, and therefore in Fig. 1b clustered with Mes subtypes that also have high expression of EMT markers. This was also the case with many cells expressing *Ngn3* above threshold that remained in the epithelium- high expression of epithelial markers such as *Spp1* and *Sox9* led to these cells clustering together with BPs in Fig. 1b. Further, by selecting using *Ngn3* expression rather than EP clusters, we are able to capture cells that cluster in alpha and beta cell groups that still express *Ngn3* above threshold, representing the earliest born hormonal cells that have not yet repressed *Ngn3*. We hope the change from “EP subtypes” to “*Ngn3*+ cell subtypes” is clearer for readers and appreciate this suggestion.

3. In the pseudotime analysis of the four EP14 subtypes, Monocle2 generated a very different cell clustering structure (Supplementary Fig. 4a) as compared to the previous analysis using Seurat based on t-SNE (Fig. 2a). For example, EP14_4 cells are located in at least two different branches which contradicts the cell clustering structure with t-SNE (similar cells should locate in a neighboring region). As a result, individual cells in the EP14_4 subtype have very different pseudotime and there are a number of EP14_3 cells located in between the EP14_4 cells. The

authors should present a consistent analysis result either defining EP subtypes using the clustering structure in Monocle2 or use other pseudotime analysis algorithms which can construct pseudotime based on t-SNE from Seurat (e.g., the cell clustering structure in Fig. 2a). For example, in the second case (using cell clustering structure in Fig. 2a), the pseudotime could be EP14_2, EP14_1, EP14_3, to EP14_4 which may better represent the maturation process of EP cells since the proliferative capacity analysis (Fig. 3b) show that EP14_2 has higher proliferative capacity than EP14_1.

We have repeated the Monocle2 analysis with the density peak clustering method (Monocle2 dpFeature procedure), which better represents the trajectory and is consistent with the analysis performed in Fig. 9. The data in Fig. 3a, Fig. 3e, and Supplementary Fig. 9 are now showing this new Monocle2 analysis.

To do this, we carried out density peak clustering (Monocle2 dpFeature procedure) to order N14 cells based on the genes differentially expressed between clusters, with the thresholds for density clustering set to 2 and 4 for rho and delta, respectively. The top 1000 significant genes (ordered by qvalue) were used for ordering e14.5 *Ngn3+* cells in pseudotime. The results are consistent with the expression-based tSNE (Seurat), which highlights an e14.5 *Ngn3+* cell trajectory progressing sequentially from N14_1 and N14_2 to N14_3, and finally N14_4 (Supplementary Fig. 9 and Fig. 3). Next, we examined the genes changing as a function of pseudotime or *Ngn3+* cell maturation. We performed hierarchical clustering on the 928 significantly differentially expressed genes ($qval < 0.01$), and have added in characterization of each of the 4 individual kinetic clusters (Supplementary Fig. 9e). We have also added a new figure panel to represent this pseudotemporal ordering in an additional way (Supplementary Fig. 9b). In addition, the ordering of N14_1 before N14_2 is expected based on previous studies, in that *Ngn3+* cells in the epithelium are present in N14_1 prior to delamination and migration out into the surrounding niche with high EMT markers in N14_2. The new Monocle2 ordering results are consistent with the Seurat tSNE.

4. In the comparison of EP14 and EP16 subtypes, why EP14_2 is not included in the hierarchical clustering analysis (i.e., Fig. 4e)?

We have updated the hierarchical clustering analysis to include N14_2 in Fig. 4e.

The authors claimed that EPs are different between e14.5 and e16.5 (Fig. 4f and 4h). They have to test whether the observation is caused by batch effects (e14.5 and e16.5 cells are from different mice and different sequencing runs). For instance, Fig 4f and 4h lack of a control cell type to show the difference of e14.5 and e16.5 EPs is not caused by batch effects. Such a control cell type could be BPs as shown in Fig. 7a where BPs from e14.5 and e16.5 are partially overlapped. The authors can either mark the EPs in Fig. 7a according to their origins (e.g., from e14.5 or e16.5) or add BPs to Fig. 4f to see if EPs from the two developmental time points are very distinct from each other. Similarly, the authors should also show that the PCA plot in Fig. 6a is not affected by batch effects (e.g., including BPs in the plot may help to clarify). The authors should also clarify how Fig. 6a is generated (it seems to be generated by pooling cells together as pseudo bulk samples).

While we previously showed origin of each batch in the *Ngn3+* cells (now Supplementary Fig. 4b), we agree that adding in other cell types to the tSNE and PCA analysis such as BPs or endocrine cells would be a strong control and we are thankful for this suggestion.

We have provided this new data in Supplementary Fig. 13d, showing tSNE plots of e14.5 and e16.5 *Ngn3+* cells with e14.5 beta cells as well as with e14.5 BPs, alpha, and beta cells. We believe that showing e16.5 *Ngn3+* cells clustering apart even with these distinct cell types

added in strengthens the conclusion that e14.5 and e16.5 *Ngn3*⁺ cells are divergent. We have added the following sentence to describe this:

“To ensure that the differences between the e14.5 and e16.5 *Ngn3*⁺ cells was not an artifact of batch effects, we subclustered *Ngn3*⁺ cells with e14.5 beta cells as well as with e14.5 beta cells, alpha cells, and BPs, and in all cases founded the e16.5 *Ngn3*⁺ cells clustered separately from e14.5 *Ngn3*⁺ cells (Supplementary Fig. 13d).”

Further, we have added BPs in the PCA plot in Fig. 7a (previously Fig. 6a), which substantiates that N14_1 and N16_1 are divergent. This has been added in the text as follows:

“Principal component analysis revealed that e14.5 and e16.5 EPs were already different upon induction, with the earliest N14_1 and N16_1 subtypes clustering apart, even more so than BPs (Fig. 7a; average cluster expression).”

To clarify how Fig. 7a is generated, as well as the data in Supplementary Fig. 14c, Supplementary Fig. 15a, Fig. 6d, and Fig. 8f, we have added the following sentences:

Supplementary Fig. 14c:

“Direct comparison of average cluster expression *Ngn3*⁺ cell subtypes between e14.5 and e16.5 revealed cohorts of differentially expressed genes, with the top 50 most significant changes illustrated in Supplementary Fig. 14c.”

Supplementary Fig. 15a:

“We found changes in the expression of epigenetic regulators in EPs, some with known functions in endocrine cell fate determination, including *Dnmt3a*, *Ezh2*, and *Rest*^{27,28,42}, as well as many not yet described in endocrine differentiation (average cluster expression data in Supplementary Fig. 15a).”

Fig. 6d legend:

“Priming of e14.5 *Ngn3*⁺ cells for the alpha cell fate and e16.5 *Ngn3*⁺ cells for the beta cell fate. Average cluster log₂ z-score normalized heatmap showing expression of select alpha and beta cell genes (x-axis) in late non-hormonal *Ngn3*⁺ cells from e14.5 and e16.5 (N14_4 and N16_4) compared to the earliest born hormonal cells still expressing *Ngn3* (y-axis). Expression ranges from blue (low) to high (yellow).”

Fig. 8f legend:

“Average cluster log₂ z-score normalized heatmap of select beta, alpha, and tip cell markers in e14.5 and e16.5 BPs.”

For the gene clustering analysis in Fig. 4i, it seems that genes that are highly expressed in EP16 are also highly expressed in EP14 late. It will be more interesting to see a differential gene expression analysis by comparing EPs from e14.5 and e16.5 to identify differentially expressed genes. In addition, analyzing the differentially expressed genes between subtypes of EPs from e14.5 and e16.5 may also be interesting (e.g., EP14_1 vs. EP16_1, EP14_4 vs. EP16_4).

We understand the utility of this data, therefore we have performed the analysis you suggested and show now in Supplementary Fig. 14c the top 50 most significant differentially expressed genes between *Ngn3*⁺ cell subtypes. We have added the following description in the results section:

“Direct comparison of average cluster expression from *Ngn3*+ cell subtypes between e14.5 and e16.5 revealed cohorts of differentially expressed genes, with the top 50 most significant changes illustrated in Supplementary Fig. 14c.”

The authors also applied single-cell qPCR to compare EP signature genes between e14.5 and e16.5 in Supplementary Fig. 7a. They concluded that the gene expression pattern is different between e14.5 and e16.5 by showing the heatmap of gene expression. It seems that some cells are actually similar between e14.5 and e16.5. A more convincing way is to show a clustering analysis of all cells from both e14.5 and e16.5 (e.g., hierarchical clustering of all the rows in Fig. 7a) to see if cells from different time points are in distinct clusters.

Thank you for this suggestion. We have merged the e14.5 and e16.5 *Ngn3*-eGFP+ single cells together and re-ran the analysis to perform clustering analysis, and indeed this convincingly shows that most of the e14.5 cells and e16.5 cells cluster separately, and is much more convincing and clear for readers to see immediately. In fact, in the two major branches we observe one composed of 92.9% e16.5 EPs while the other was 81.6% e14.5 EPs (see Rebuttal Figure 1 below).

We have edited this figure to show this data (Supplementary Fig. 14a) and have added the following in the text:

“After isolating e14.5 and e16.5 *Ngn3*-eGFP+ cells, we observed differential expression of established EP genes in single cells over time using Fluidigm C1 analysis, with clustering analysis showing one branch composed of 92.9% e16.5 single EPs while the other was 81.6% e14.5 single EPs (Supplementary Fig. 14a).”

In Supplementary Fig. 7b, the authors used MATQ-seq to show the gene expression of signature genes in EP14_4. However, the plot is not consistent with what they claimed. The plot showed that genes in EP14_3 are highly expressed.

Thank you for your keen eye; this was an error in labeling in the key that we have amended. The gene expression of *Ngn3*-eGFP+ cells from MATQ-seq are indeed consistent with N14_4.

5. It is unclear what cells are used in the analysis of differentiation from EPs to alpha and beta cells in Fig. 8a (e.g., e14.5? e16.5? or merged e14.5 and e16.5?). If merged e14.5 and e16.5 EPs are used in Fig. 8a, this contradicts the previous conclusion that EPs from e14.5 and e16.5 were very distinct.

In Figure 8a, we included both e14.5 and e16.5 data sets. Overall, with this analysis we aimed to determine how BPs contribute to endocrine cell fate at e14.5 and e16.5. And to accomplish this goal we broadly categorize cells as BPs, EPhi, EPlo, alpha, and beta cells (Rebuttal Figure 2B). Thus, for this analysis we merged, or classified, the distinct e14.5 and e16.5 EP clusters

together prior to input into Monocle2 (Rebuttal Figure 2A-D). Inspection of the graph-based clustering results (Seurat) prior to input into Monocle2, shows clear differences in the clusters (Rebuttal Figure 2A-D) distinguishing e14.5 from e16.5 EPs. This is consistent with our previous analysis defining the distinctions between e14.5 and 16.5 Ngn3 expressing cells. Close inspection of the Monocle2 trajectory colored by cluster (same as Rebuttal Figure 2A), shows that e14.5 and e16.5 EPs do cluster apart (Rebuttal Figure 2E-F). It should be noted that all of the expression analysis done with the Monocle2 algorithm and then used for determining the genes that were input for the ordering of all cells along the shown developmental trajectories would be averaged between the two time points for the EPs. Thus, the distinctions as shown across the Monocle2 minimum spanning tree (MST) appear minimal for this group of cells given their smaller overall contribution to the shown dataset as compared to BPs (Rebuttal Figure 2E-F). Moreover, their distinctions may also appear less obvious because of their position in the BP-EP-alpha/beta pseudotime trajectory are very close together, which matches their apparent position embedded across the Seurat tSNE (Rebuttal Figure 2G-H). So, although e14.5 and e16.5 EPs have clear transcriptional differences, when merged they cluster together, which makes sense given that both e14.5 and e16.5 EPs should occupy a place in pseudotime prior to alpha and/or beta cells. Indeed, EPe14 cells have a mean pseudotime score of 10.2 (standard deviation = 1.8) and EPe16 have a pseudotime score of 8.5 (standard deviation = 2.6), thus they are similar but distinct. Below is the Rebuttal Figure 2 addressing these points.

A) tSNE of showing endocrine cell clusters. **B)** tSNE showing endocrine cell clusters. Same data set as A, and manuscript Fig. 8a. **C)** tSNE showing what embryonic timepoint each individual transcriptome is derived from. Dark blue is e14.5, and lighter blue is e16.5. **D)** Feature plots showing expression for Ngn3 (Neurog3), Ins2, Gcg, and Spp1. Low or no expression is yellow, and high expression is shown in red. **E)** Embryonic endocrine cell differentiation trajectory. Colored by cell cluster identity from A. **F)** Monocle2 endocrine cell MST colored by timepoint. **G)** Monocle2 pseudotime score for individual barcoded single cell transcriptomes and embedded across the Seurat tSNE. Darker colors highlight beginning of pseudotime, and lighter colors indicate the end (more differentiated) of pseudotime. Position of EPe14 (dotted line) and EPe16 (dashed line) along pseudotime trajectory (arrow) highlighted in yellow box. **H)** Violin plot of pseudotime scores for all cells within indicated clusters. Dot indicates mean pseudotime score, and black lines are standard deviation. Colored by cluster identity.

We have added the following to the text and revised Figure 8a in main text so that this is clear to the reader:

“Consistent with our previous analysis, EPhi cells appear prior to EPlo before bifurcation into either alpha or beta cells, with e14.5 EPs clustering together into three clusters while e16.5 EPs clustered apart (EPe14 cells have a mean pseudotime score of 10.2, standard deviation = 1.8; EPe16 have a pseudotime score of 8.5, standard deviation = 2.6).”

The developmental trajectory in Fig. 8a may not be a good example since some alpha and beta cells appear to be in the same branch which is not consistent with the t-SNE analysis in Fig. 5g. The cell clustering based on t-SNE in Fig. 5g actually showed a very clear trajectory of cell development from EPhi, EPlo to beta or alpha cells for both e14.5 and e16.5 cells. It will be interesting to see a pseudotime analysis based on this cell clustering structure and identify genes that show different gene expression patterns along the pseudotime for e14.5 and e16.5 respectively.

We agree that the Monocle2 ordering of single cells in this trajectory does not discretely resolve the alpha vs. beta fate choice. We performed a number of tests to resolve this branch point, however, this was the best result after carrying out density peak clustering (Monocle2 dpFeature procedure) and differential expression analysis to determine the top 300 genes (lowest q-value) for use in ordering single cells. However, there are benefits to using this Monocle2 method as it uses pseudotime interference and provides lists of genes associated with specific kinetic clusters (or gene expression trends). This gives an unbiased list of gene expression changes, which can then be validated using secondary methods.

For this reason, we also used the Wishbone algorithm as this secondary method to order single cells along a trajectory, which very clearly separated the Gcg+ and Ins2+ cells into discrete branches. Wishbone, unlike Monocle2, is an algorithm that is specifically suited to positioning cells along bifurcating ontological trajectories. Importantly, a critical assumption made by the Wishbone algorithm that makes it so robust for ordering bifurcating trajectories is the following; the trajectory of any single cell bifurcates to one of two cell fates. And Wishbone has been shown to outperform other pseudotemporal ordering algorithms, like Monocle (version 1) (Setty et. al. 2016). Thus, we see the two algorithms as complementary in that we can identify developmental gene trends with Monocle2 and then visualize the bifurcating genes for alpha and beta cells with Wishbone. Wishbone visualization of specific kinetically expressed genes uncovered from Monocle2 allows us to determine if they follow the same trajectory at this alpha/beta branch point when better resolved, for example Isl1, Gpx3, and Pyy are found by both Wishbone and Monocle2 to increase in alpha cells. We have added more information

regarding this into the text to clarify why we used two algorithms and the limitations and benefits of each:

“We addressed this by two independent methods of pseudotemporal ordering, Monocle2 which orders single cells throughout pseudotime before grouping transcripts based off of expression patterns over pseudotime, and Wishbone which acts as a branch predictor to resolve the bifurcating trajectories. The use of both Monocle2 and Wishbone allowed the unbiased analysis of gene expression trends over pseudotime coupled with a secondary method to visualize transcript changes over time with higher branch fidelity.”

6. Fold change, p-value, and FDR are not provided in the differential analysis of the ATAC-seq data (e.g., Supplementary Table 8). It is unclear that the ATAC-seq signals in Fig. 5e were normalized or not in the comparison of e14.5 and e16.5. The signals are not directly comparable if the library sizes from e14.5 and e16.5 are different. It will also be interesting to see functional annotation analysis of the differential ATAC-seq peaks between e14.5 and e16.5 (e.g., either use GREAT for the differential peaks or use GO for the neighboring genes of the differential peaks).

We have added in Fig. 5c (previously Fig. 5e) that the significantly differentially accessible ATAC-seq peaks were determined based off of an adjusted p-value <0.05, thus the FDR cutoff is 0.05. We have added peak locations, statistics from the DEseq2 output, and annotations to Supplementary Table 8. The UCSC browser tracks shown are normalized by read count (default HOMER makeUCSC), which we have added into the methods section:

“Visualizations of ChIP-seq signals was done with HOMER (annotatePeaks.pl), and all signals are normalized by read count, where scores represent read count per bp per 1×10^7 reads.”

We agree the functional annotation analysis of the differential ATAC-seq peaks between e14.5 and e16.5 would be interesting, therefore we have performed this analysis and added it into Supplementary Fig. 15c. This analysis was very exciting to us, as the results supported our findings with hallmarks of pancreatic beta cells enriched in e16.5 peaks, and we appreciate this suggestion from you. We have added the following to the text to describe this analysis and results:

“In addition, gene ontology analysis of the accessible peaks in e14.5 and e16.5 EPs revealed enrichment of hallmarks of pancreatic beta cells at e16.5 (Supplementary Fig. 15c).”

7. The Supplementary Tables are not easy to interpret. The authors should provide more information about the Supplementary Tables (e.g., what are the values in each column represent).

We apologize this was not clear before, and have spent time to clarify each table and what each column represents.

--

Reviewer #2 (Remarks to the Author):

Scavuzzo et al. performed single-cell RNA-seq on mouse pancreata at e14.5 (15,228 cells) and e16.5 (2,006 cells) in order to better characterize endocrine progenitors (EP). By subclustering Ngn3 positive cells, the authors determine that the EP populations are transcriptionally heterogeneous and identify 4 distinct subtypes present at both e14.5 and e16.5. Based on

pseudotime ordering, they propose that these subtypes reflect different stages of EP maturation. Furthermore, the authors show that EP at e14.5 and e16.5 are transcriptionally different and appear to originate from distinct progenitor pools, which underlies their bias towards alpha and beta cells, respectively. Based on these findings, the authors propose that the fate of distinct hormone-expressing lineages are determined before Ngn3 expression in epithelial progenitors (BPs or MPPs). The study is technically of high quality and the results are interesting, especially the finding that developmental timing (although already known, but mechanistically not understood) is important for EP specification into alpha or beta cells.

Thank you for your kind comments. We were also very interested to find the importance of developmental timing in EP fate potential. We have followed your suggestions below to strengthen the findings in the manuscript and make the results clearer to the reader.

Major points:

- There is not enough experimental evidence to support some of the main conclusions made by the authors, e.g. the notion that the distinct endocrine lineages are specified prior to Ngn3 expression. Without providing solid evidence for some of the claims the authors should tune down some of the (over)statements.

We have revised our manuscript and made sure to not overinterpret results as they are described, but rather to only speculate on their implications in the discussion section.

- Another example is that the authors identify and characterize the putative maturation stages of EPs, but they do not directly demonstrate (other than by the pseudotime analysis) that EPs sequentially progress through these stages. It would be ideal to validate this finding by tracing EPs progression through the four stages, which is technically very challenging. Moreover, it would be interesting to know what factors/pathways are upstream of these putative transitions. This should be discussed.

We agree that this will be very interesting routes to pursue in detail in the future. Here we show that *Ngn3*⁺ cells can be divided depending on their development, starting from the initial pool of *Ngn3*⁺ cells that still remain in the epithelium, through markers of delamination and migration, and by the up-regulation of mature endocrine markers. Although it has never before been described at this resolution, this finding reflects *Ngn3*⁺ cell differentiation supported by other studies, which we have made sure to cite to convey that these stages are based off of known transcripts.

For example, Shih et al. 2012 showed that the earliest EPs express both Ngn3 and Sox9, as Sox9 is required for the expression of Ngn3. These cells that are induced and still in the epithelium are known to then delaminate and migrate out into the surrounding mesenchyme (Gouzi et al. 2011; Metzger et al. 2012). Finally, these cells begin to express markers of endocrine maturation before finally having some *Ngn3*⁺ cells co-expressing hormones in the earliest differentiating EPs into mature endocrine cells. We expand upon the known process of *Ngn3*⁺ cell development by ordering single *Ngn3*⁺ cells on a continuous spectrum to outline all of the transcripts globally changing over time including the expression and trends of novel genes. We have added into the discussion section discourse regarding upstream factors/pathways, and have also added more citations and stated that these key, stage-specific markers were previously established in the results section:

“Supporting the *Ngn3*⁺ cell subtype developmental order, we observed enriched expression of known early EP markers in N14_1, including *Sox9*¹⁶, *Spp1*³⁴, *Onecut1*³⁵, and *Hnf1b*³⁶, known delamination and migration markers in N14_2 including *Twist1*, *Snai1*, and *Snai2*^{37,38}, and

known committed EP markers in N14_3 and N14_4 including *Chga*³⁹, *Arx*²³, *Irx1*²⁴, *Irx2*, *Neurod1*⁴⁰, *Pax4*⁴¹, and *Pax6*⁴² (Fig. 3c and Supplementary Fig. 9d). ”

Further, using Sox9cre lineage tracing experiment we now show that e13.5 Sox9+ cells express markers of different EP subtypes, indicating that the identified here EP subpopulations reflect EP maturation. Specifically we observed co-expression of Sox9-GFP with E-cadherin (for N14_1), *Zeb2* (for N14_2), *Insm1*, *Nkx6-1*, and *Acly* (for N14_3) and *Chga*, *Ghrl* and *Sst* (for N14_4). These data is shown in the new Supplementary Figure 10.

We have also added the following to the text:

“To support that N14_1 cells mature into other *Ngn3*+ cell subtypes, we crossed Sox9-Cre^{ERI2} mice⁴⁸ with the ROSA-mTmG line⁴⁹. As Sox9 is expressed in BPs and is essential for the expression of *Ngn3*¹⁶, after Cre recombination GFP marks BPs as well as their progeny: EPs and endocrine cells. We pulsed these mice with tamoxifen at e13.5 to label N14_1 cells with GFP before collecting tissue at e15.5 to analyze markers of *Ngn3*+ cell maturation (Supplementary Fig. 10a and 10b). We observed GFP+ cells expressing markers of all four stages of *Ngn3*+ cells supporting that N14_1 cells expressing Sox9 and E-cadherin develop into N14_2 (expressing *Zeb2*), N14_3 (expressing *Insm1*, *Nkx6-1*, and *Acly*), and N14_4 (expressing *Insm1*, *Nkx6-1*, *Acly*, *Chga*, *Ghrl*, and *Sst*; Supplementary Fig. 10c).”

- The authors analyzed 2006 cells from e16.5, with possibly only a very small fraction being EP (5% at e14.5). It is difficult to draw any conclusions from such a low number of cells (i.e. EP16_1 does not show a tight cluster in Fig 4c and the number of cells analyzed is not even mentioned). It is not obvious why the authors sampled 15.000 cells at e14.5 and only 2000 at e16.5 although material is even better accessible at e16.5?! This might bias the analysis/interpretation and is the only experimental/technical weakness of the study.

We were able to sequence a higher number of single cells from approximately the same number of embryos at e14.5 compared to e16.5, which we believe is due to high RNase activity in the e16.5 pancreas affecting library composition, as our STAMP number dropped to about 20% the number at e14.5. As we also analyze EPs using ATAC-seq, Fluidigm C1 qPCR, MATQ-seq, qPCRs, immunostaining, and RNA *in situs*, we believe that our results are convincing despite the relatively low cell number in the droplet based single-cell sequencing methods. In these other methods, including Fluidigm C1 and MATQ-seq, we used FACS to sort for *Ngn3*-eGFP+ cells, which ensured that live cells are captured followed by several washes before lysing cells. We have also FACS sorted for *Ngn3*-eGFP+ cells to perform Drop-seq, however the combination of Drop-seq low efficiency (12% according to Macosko et al. 2016), the low numbers of *Ngn3*+ cells per litter and loss during sorting made this unobtainable to obtain high quality data. To increase confidence in our findings regarding e16.5 EPs, we have performed additional experiments to immunostain and validate EP heterogeneity, providing a new figure (Supplementary Fig. 12). We have also added the following to the text:

“Next, we validated e16.5 *Ngn3*+ cell heterogeneity by immunostaining and flow cytometry. Immunostaining confirmed e16.5 *Ngn3*+ cell subtypes, showing heterogeneity of *Ngn3*-eGFP+ cells with *Acly*, *Sox4*, *Yap1*, E-cadherin, *Chga*, and *Nkx6-1* (Supplementary Fig. 12). Flow analysis showed subsets of *Ngn3*+ cells co-expressing *Mucin1* (enriched in N16_1), *Rbfox2* (N16_2 and N16_3) and *Nkx6-1* (Supplementary Fig. 8b).”

- Moreover, other potentially interesting findings, such as the differences in chromatin accessibility between EP at e14.5 and e16.5, are not followed up in order to reveal factors and

pathways regulating endocrine specification. This is experimentally challenging, but maybe the authors can discuss some of these points. First, apart from the increased expression of epigenetic regulators, can one show increased activity of these complexes, for example enhanced H3K27me3 or H3K4 demethylation? Second, it would be good to determine whether the chromatin changes are necessary ('drivers') for endocrine specification or are only 'passengers', e.g. by inhibiting epigenetic modulating enzymes. Moreover, how can the authors explain the increase in expression of epigenetic regulators during beta cell specification but not during alpha cell specification?

We agree that this is an interesting direction to follow and hope that by providing our lists of datasets, others or we can use this information to pursue new studies.

We were able to analyze available datasets and cross compare to our ATAC-seq accessibility peaks, including ChIP-seq data from two key transcription factors increased at e16.5, Nkx6-1 and NeuroD1, both of which are enriched in binding at e16.5 peak regions. The overlay of the 16.5 ATAC-seq peaks with this independent dataset showing Nkx6-1 ChIP-seq is in very good agreement with our previous analysis. We have provided below an example of these tracks showing an enriched e14.5 peak next to an enriched e16.5 peak (in novel EP gene, *Tox3*), with high enrichment of Nkx6-1 in the e16.5 peak but not the e14.5 peak. See below Rebuttal Figure 3.

Further, we analyzed H3K27ac and H3K27me3 marks from e17.5 beta cells and compared to our accessible peaks, and again find more H3K27ac around e16.5 peaks and less H3K27me3. Finally, we analyzed the active enhancer mark H3K4me1 in e13.5 Ngn3-low cells and see high enrichment of this mark across our 609 enriched e14.5 EP peaks. Also, we see a very low e13.5 Ngn3-low H3K4me1 ChIP-seq signal across our 749 e16.5 ATAC-seq peaks. While this data doesn't directly support a model where increased demethylation occurs, it is well known that chromatin accessibility correlates strongly with active enhancer markers (e.g. H3k27ac and H3K4me1; Yu et al. 2018), thus it could very well be that the e14.5 EP accessible regions are demethylated by e16.5. However, we acknowledge that this point is not addressable without e16.5 EP specific H3K4me1 ChIP-seq experiments. Overall, these data support our observation that increased chromatin accessibility at specific loci is concomitant with the induction of beta cells. We have added this data into Fig. 5, with Fig. 5f-k showing this, and the following to the results section:

“Chromatin accessibility of *Ngn3* and *Nr5a2* (a tip cell marker) loci were enriched at e14.5 compared to e16.5, while accessibility across *Ins1* and potassium calcium activated channel *Kcnn2* loci were increased at e16.5 (Fig. 5d). To investigate if temporal changes in motif accessibility in EPs corresponded with endocrine fate determination, we performed motif enrichment analysis on the statistically differentially accessible ATAC-seq peaks (adjusted P-value <0.05). Strikingly, we found that beta cell associated transcription factor motifs including Nkx6-1, NeuroD1, and Isl1 were well represented at e16.5 (Fig. 5e). Indeed, Nkx6-1 is known to be part of a gene regulatory network essential for maintaining beta-cell identity and molecular physiology into adulthood⁶⁰⁻⁶³. From scRNA-seq, we observed a robust increase in expression of Nkx6-1 in e16.5 *Ngn3*+ cells compared to e14.5 *Ngn3*+ cells (Supplementary Fig. 15d). Importantly, analysis of Nkx6-1 ChIP-seq data from adult pancreatic islets confirmed that Nkx6-1 was bound to many of the genomic regions we found to be more accessible in e16.5 EPs (Fig. 5f and 5g). Furthermore, as NeuroD1 motifs were enriched globally in both data sets as well as in e16.5-specific EP ATAC-seq peaks, we assessed NeuroD1 DNA-binding from e14.5 pancreas NeuroD1 ChIP-seq data and found moderately enhanced occupancy across the e16.5 peak set (Fig. 5h). Thus, many of the accessible chromatin regions identified in embryonic EPs are occupied by key pancreatic transcription factors, including Nkx6-1 and NeuroD1.

Given that many of the differentially accessible regions were annotated to intronic and intergenic regions (92% for e14.5, and 94% for e16.5), we sought to interrogate their epigenetic status. We integrated available histone mark ChIP-seq profiling data derived from murine e13.5 embryonic EPs (*Ngn3*-GFP+ low), and e17.5 Beta-cells⁶⁴. Indeed, the 609 e14.5 enriched ATAC-seq peaks showed high H3K4me1 signal in e13.5 *Ngn3*-GFP low cells, consistent with their status as active enhancers with high chromatin accessibility (Fig. 5i). Conversely, e13.5 *Ngn3*-GFP low H3K4me1 signal was not highly enriched across the 749 e16.5 accessible ATAC-seq peaks (Fig. 5i). This suggests that the 749 e16.5 specific peaks are not yet commissioned as active enhancer regions at e14.5. However, further histone mark profiling is required to determine the enhancer status of these regions at e16.5 definitively. Thus, the differentially accessible regions identified in EPs via ATAC-seq are dynamic at the epigenetic level. In e17.5 beta cells, we also noted increased signal for the active H3K27ac mark in the accessible peaks of e16.5 EPs compared to e14.5 EPs (Fig. 5j). Conversely, we observed decreased signal for inactive H3K27me3 marks in e17.5 beta cells across e16.5 EP peaks (Fig. 5k). Together, this supports that the e16.5 peaks are also active genomic regulatory regions in e17.5 beta cells. In addition, gene ontology analysis of the accessible peaks in e14.5 and e16.5 EPs revealed enrichment of hallmarks of pancreatic beta cells at e16.5 (Supplementary Fig. 15c). Overall, this suggests that changes in EP chromatin accessibility could influence EP potential, with chromatin accessibility remodeling between e14.5 and e16.5 leading to a more permissive environment for beta cell formation at e16.5.”

Minor points:

- GO-Term analysis showed that processes associated with epigenetic regulation were enriched as well in EP14_3/4 (Fig. 3d). However, this does not fit with the heatmap in Supp. Fig. 7c. Also, later the authors somehow imply that this finding is specific to E16.5 EPs.

The GO analysis showed that compared to other e14.5 *Ngn3*+ cell subtypes (Fig. 3d), the later subtypes (N14_3 and N14_4) had enriched epigenetic regulation, however when we compare not only within e14.5 EP subtypes but across timepoints, we see enrichment of specific epigenetic regulators in e16.5 EPs as well. However, while we do see ATP-dependent chromatin remodeling and nucleosome disassembly highly enriched in N14_3, our N14_4 had more enriched processes including peptide hormone processing, endocrine pancreatic

development, and second-messenger mediated signaling. Therefore we have changed the figure and text to present these more highly significant processes for this subtype:

“N14_4 exhibited enriched genes associated with endocrine pancreatic development, peptide hormone processing, and second-messenger mediated signaling.”

We agree that there are increased epigenetic regulators in both e14.5 and e16.5 EPs, and have revised our text to ensure that we do not convey that this is specific to e16.5 EPs. What we find, however, is that at e16.5 we find increased expression of *Dnmt3a*, which is known to cooperate with *Nkx2-2* to bind and repress expression of the alpha cell regulator *Arx*, thus potentiating the beta cell fate. We have edited the following in the results section to clarify:

“The increase in expression of epigenetic regulators associated with the beta cell fate in e16.5 *Ngn3+* cells concurrent with increased beta cell genes and decreased alpha cell genes led us to investigate the contribution of the chromatin state in diverting EPs towards the alpha or beta cell fate. We found changes in the expression of epigenetic regulators in EPs, some with known functions in endocrine cell fate determination, including *Dnmt3a*, *Ezh2*, and *Res*^{27,28,50}, as well as many not yet described in endocrine differentiation (average cluster expression data in Supplementary Fig. 15a). Interestingly, the known beta cell determinant *Dnmt3a*^{28,51} was increased in expression in e16.5 *Ngn3+* cells, while the alpha cell regulator *Ezh2*^{27,52} was enriched in e14.5 *Ngn3+* cells.”

- An heatmap showing all marker genes and genes changing in pseudotime (and not only selected genes) would be ideal for validation of the clustering and a continuous pseudotemporal progression.

We have added this heatmap to the figures in Supplementary Fig. 9e.

- The pseudotime analysis shows that EP14_1 progresses into two distinct branches: EP14_4 and EP16. Could the authors explain the transition from EP14_1 into EP16? Which maturation stage of EP16 do EP14_1 progress into?

The pseudotemporal ordering of single cells places cells with similar gene expression levels and patterns together across a trajectory. We used this method to show that e16.5 EPs are divergent from e14.5 and to uncover the differences in gene expression, however the e14.5 EPs do not progress into e16.5 EPs. Actually, it is quite interesting that the EPs analyzed at e14.5 should no longer be present at e16.5 (the duration of *Ngn3+* cell period is about 11.3 hours according to Bankaitis et al. 2015, therefore the 48 hour timeframe should not pick up any cells still in the *Ngn3+* phase from e14.5). With this knowledge, we are confident that the *Ngn3+* cells assayed at e16.5 are not more mature cells from e14.5 but rather newly formed EPs.

- Fig. 5g: To confirm the single cell RNAseq data, alpha- and beta cells should be counted in sections.

We have provided a new figure with immunostainings and quantification of *Gcg+* and *Ins+* cells at e14.5 and e16.5, now in Fig. 6b and 6c. Below are the new figure panels:

We have added this description in the text of the new figure panels:

“We validated the findings from scRNA-seq by immunostaining and quantification, confirming that more alpha cells were present at e14.5 while more beta cells were present at e16.5 (Fig. 6b and 6c). We next compared non-hormonal late e14.5 (N14_4) and e16.5 EPs (N16_4), to the earliest hormonal expressing cells at e14.5 EPs (*Ngn3+*/*Gcg+* and *Ngn3+*/*Ins2+*). N14_4 cells expressed hallmark alpha cell genes similar to the earliest born *Ngn3+*/*Gcg+* alpha cells, including *Irx1*, *Arx*, *Pou3f4*, *Irx2*, and *Gcg*, while N16_4 cells were closely related to the earliest born *Ngn3+*/*Ins2+* beta cells expressing *Ins1*, *Ins2*, and *Ociad2* (Fig. 6d). This suggests that the EP fate is primed depending on the embryonic stage, with a higher potential for alpha cells early during the secondary transition and beta cells later.”

- Supplementary Fig. 1a,b: If the authors want to make any conclusion about the abundance of *Ngn3+* cells at different embryonic stages, they need to count the *Ngn3+* cells since the qPCR analysis does not give any information about it. Also, the qPCR analysis does not reflect the immunostainings.

We have added new data showing the quantification of *Ngn3*-eGFP+ cells in Supplementary Fig. 1b. The discrepancy between the immunostainings and the qPCR could be due to a decrease in the number of mesenchymal cells as the pancreas develops (early it is heavily enveloped by mesenchyme), leading to a higher transcript when normalized to *Gapdh* in later stages. It is also possible that there is higher transcript per cell rather than a higher number of *Ngn3+* cells, since the quantification supports that the e14.5 pancreas has the highest number of *Ngn3+* EPs.

- Fig. 1b: Would be informative to include a figure with the actual number of captured EPlo, EP_{hi}, alpha and beta cells (in addition to Supplementary Fig. 2a). The authors claim that EPlo cells bifurcated into alpha or beta cells. Not sure if one can draw such a conclusion from a tSNE plot.

We have added the numbers in a table in Supplementary Fig. 3a. We also agree that the relationships in space in tSNE cannot be used to draw such conclusions, therefore we have changed the way we describe this data:

“However, graph-based clustering more faithfully captured the developmental relationships of cells than kernel-based clustering when visualized by dimensionality reduction, for example the close proximity of EPs into alpha or beta cells (Supplementary Fig. 2g). Thus, we used graph-based clustering for subsequent droplet-based scRNA-seq analyses.

We uncovered two EP subtypes, with EPhi (high levels of *Ngn3*) clustering further from alpha and beta cells while EPhi (low levels of *Ngn3*) were in closer proximity to alpha or beta cells (Fig. 1b).”

- Fig. 1g: Immunostainings (and selected sections) are not very convincing. It would be good to show single channels. Also, Sox9-GFP seems to be cytoplasmic.

We have added the single channel images in Supplementary Fig. 1d. The Sox9-eGFP mouse line we utilized is not a fusion but rather a knock-in mouse line; therefore GFP can be cytoplasmic while the Sox9 protein is nuclear. The mice were obtained from Haruhiko Akiyama and Richard Behringer and have been previously characterized (Nel-Themaat et al. 2010). Below is an example of how the Sox9-eGFP should look from their Developmental Dynamics publication:

We have added a sentence to clarify this mouse line in the figure legend so it is not confusing to readers:

“On left, immunofluorescent staining of Dcdc2a (green), knock-in Sox9-eGFP (blue), with Nkx6-1 (red) and nuclei (DAPI, gray).”

- The authors claim that the higher number of alpha cells captured signifies that at e14.5 alpha cells preferentially differentiate. However, many alpha cells are formed during the primary transition which could also explain the higher number of alpha cells present at e14.5.

Yes we agree with this point: alpha cells are also born during primary transition, mostly in dorsal bud. The molecular events associated with the differentiation of these early alpha cells are not well understood, but some differences have been suggested between alpha cells formation during primary and secondary transition. Importantly very few alpha cells are born during the primary transition. While many of the first-born alpha cells are lost as development proceeded, some might persist into adult islets (Gu G. et al. 2002). However, we do understand that the existence of earlier formed alpha cells can not be conclusively ruled out with scRNA-seq, therefore we performed Sox9 lineage tracing during secondary transition (at e13.5 and e15.5) and quantified alpha and beta cell ratio at during two time points during secondary transition. The results are shown below in Rebuttal Fig. 4:

While the vast majority of EPs form during the secondary transition (e12.5-e17.5), few endocrine cells form during the primary transition (e9.5-e12.0) and have been shown to predominantly give rise to alpha cells (Gu et al 2002). Therefore to further substantiate the alpha vs. beta lineage bias of progenitors during secondary transition, we labeled alpha and beta cell born after e13.5. To this end, we crossed Sox9-CreERT2 mouse with ROSA-mTmG line to collect tissue during different developmental windows and analyze the ratio of Gcg and Ins cells co-expressing GFP (Rebuttal Fig. 4a). Using this strategy, we pulsed mice with tamoxifen to sparsely label Sox9+ cells and their progeny and 48 hours later observed a higher ratio of GFP+ cells co-expressing Gcg than Ins during the e13.5-e15.5 window compared to e15.5-e17.5 (Rebuttal Fig. 4b and 4c). These results are consistent with our observations of a shift in alpha vs. beta cell potential in progenitors during secondary transition.

- Figure 3d: Can the authors explain why Ngn3 expression is peaking in late maturation, whereas according to the previous data EP16_4 consists mainly of Ngn3 low cells?

In the first figure, the EPs grouped into two groups, EPhi and EPlo, named based off of Ngn3 expression levels. In Fig. 3, we selected single cells by a threshold of Ngn3 expression for subclustering, which allowed us to include early, first born Ngn3+ cells that are still in the epithelium and thus cluster in Fig. 1 with BPs, as well as Ngn3+ cells (in Fig. 3, cluster 2) that highly express EMT markers and therefore in Fig. 1 cluster with mesenchymal cells. When we subclustered these and include the early cells that are still in the epithelium and delaminating/migrating, we find that Ngn3 expression peaks in the later cluster 3. To avoid confusion with the terminology, we have changed the names throughout the manuscript from EP subtypes to Ngn3+ cell subtypes, making more clear that these cells are not only derived from the EPhi and EPlo in Fig. 1 but include all Ngn3+ cells over threshold of expression. For example, "EP14_1" has now become "N14_1." We hope this is clearer now.

- Supplementary Fig. 2d + Fig. 2e: Immunostainings not convincing. Single channels required.

Single channels have been added for both of these, in Supplementary Fig. 2d and for Fig. 2e in Supplementary Fig. 5 with added quantification.

- Supplementary Fig. 3a,b: No information on flow cytometry analysis in experimental procedure part (staining protocol, antibodies, fluorophores...). In b) the labelling GFP-/GFP+ is missing.

Thank you for letting us know that this was missing. We have added the below information to the methods section and have also ensured the labeling in 3b includes GFP- and GFP+.

“Flow cytometric analysis

The pancreas was dissected before dissociation in TrypLE (Invitrogen) at 37°C for 30 minutes with agitation. Cells were quenched in FACS buffer (2% FBS, 10mM EDTA in PBS) and filtered through a 40µm cell strainer (BD Biosciences). Cells were stained using transcription factor staining buffer kit (Invitrogen). Cells were incubated in fixation and permeabilization buffer on ice for 15 minutes, washed with wash buffer, and stained with primary antibodies for 1 hour on ice in wash buffer. After another wash, cells were stained with secondary antibodies for 1 hour on ice in wash buffer. Cells were spun down, resuspended in FACS buffer (3% FBS in PBS) and analyzed on LSRFortessa II (BD Biosciences). Data were analyzed with FlowJo software (Tree Star Inc.).”

- Supplementary Fig. 4a: Can the authors please explain the monocle ordering of single EPs in particular of the EP14_1 and EP14_2 cells. Why do the EP14_2 cells not map close to EP14_3 cells?

We have repeated the Monocle2 analysis with the density peak clustering method (Monocle2 dpFeature procedure), which better represents the trajectory. The data in Fig. 3a, Fig. 3e, and Supplementary Fig. 9 are now showing this new Monocle2 analysis.

For this, we carried out density peak clustering (Monocle2 dpFeature procedure) to order N14 cells based on the genes differentially expressed between clusters, with the thresholds for density clustering set to 2 and 4 for rho and delta, respectively. The top 1000 significant genes (ordered by qvalue) were used for ordering e14.5 Ngn3+ cells in pseudotime. The results are consistent with the expression-based tSNE (Seurat), which highlights an e14.5 Ngn3+ cell trajectory progressing sequentially from N14_1 and N14_2 to N14_3, and finally N14_4 (Supplementary Fig. 9 and Fig. 3). We have also added a new figure panel to represent this pseudotemporal ordering in an additional way to support that N14_2 is between N14_1 and N14_3 (Supplementary Fig. 9b). The ordering of N14_1 before N14_2 is logical, in that Ngn3+ cells in the epithelium are present in N14_1 prior to delamination and migration out into the surrounding niche with high EMT markers in N14_2. The new Monocle2 ordering results are consistent with the Seurat tSNE.

- Fig. 4a: Actual number of captured EPlo, EPhi, alpha and beta cells should be included.

We have added the numbers in a table in Supplementary Fig. 11b.

- Fig. 4g: These are very few cells to draw conclusions on cell type distributions and that differences are not due to sampling biases.

Despite the power of droplet-based RNA-seq to survey a large number of cells, on average only around 2000 genes are detected per cell. This is enough to separate different cell types in a large population of cells. However, to look at individual single cells and compare the gene expression differences at a single gene level, we sequenced deeper with a more sensitive assay. MATQ-seq permits sequencing a cell in depth with high sensitivity and accuracy. We obtained more than 5 million reads per cell. This data provides a more detailed picture of EPs, in addition to the results obtained from droplet-based RNA-seq, ATAC-seq, and Fluidigm C1 qPCR. As we already identified e14.5 vs e16.5 *Ngn3*⁺ cell clustering through droplet-based RNA-seq, we used MATQ-seq to characterize these cells further. With the sensitivity and accuracy provided by MATQ-seq combined with the high-throughput nature of droplet-based RNA-seq, we feel confident in the results, however we also can appreciate that with this low cell number the sampling of our data can not definitively show that e14.5 and e16.5 EPs are different and rather just support this, therefore we have amended our text to represent this more accurately:

- Fig. 4h: It would be interesting to show, where the different subtypes from e14.5 and e16 are in the overall pseudotime analysis. Does it agree with the individual analyses?

We have added the subtype information for e14.5 *Ngn3*⁺ cells (N14_1, N14_2, N14_3, and N14_4) in Fig. 4h as well as in Supplementary Fig. 13f, however the e16.5 *Ngn3*⁺ cell subtypes cluster in very close proximity, therefore we have labeled this as N16.

- Page 13, line 306 – Suppl. Fig 7b does not show qPCR analysis of *Chgb*, *Chga* and *Pax6*.

This has been amended to refer to the correct analysis, MATQ-seq.

- Fig.6a and 4e: I am a bit confused in Fig. 4e the authors show that EP14_1 and E16_1 cluster (hierarchical clustering) and in Fig. 6e EP14_1 and E16_1 cluster apart (PC analysis).

In Fig. 4e we select key markers of different EP maturation stages to compare e14.5 and e16.5 *Ngn3*⁺ cells based off of known transcripts (for example, *Sox9* is known to be necessary for *Ngn3* expression, therefore is in the earliest *Ngn3*⁺ cell subtype). However in Fig. 6e, the PCA is comparing gene expression on a global scale, not limited to these key markers. Therefore overall *Ngn3*⁺ cells are divergent between e14.5 and e16.5, but they still express key markers of EPs. To clarify this in the text, we have changed the following:

“We next wanted to know how similar the e16.5 *Ngn3*⁺ cell subtypes were to the e14.5 *Ngn3*⁺ cell subtypes by analyzing select key markers of EPs. Hierarchical clustering of e14.5 *Ngn3*⁺ cell subtypes markers in e16.5 *Ngn3*⁺ cells showed that, based on these key markers, N14_1 and N16_1 were related while N14_3 and N16_2 clustered together and N14_4 clustered with N16_3 and N16_4 (Fig. 4e). These results demonstrate that EPs are heterogeneous at e14.5 and e16.5 reflecting developmental progression, with canonical markers for each subtype maintained between both timepoints.”

- Why is EP14_2 not included in the analysis in Figure 4e and f?

We have updated the hierarchical clustering analysis to include N14_2 in Fig. 4e.

- In Figure 7a it is not clear to me why e14.5 BP branch into beta and alpha cells while the authors stated previously that e14.5 EP give rise predominantly to alpha cells. I don't

understand the conclusion of the authors that the BP branches impact fate decisions as the branching occurs before the EPhi cells appear.

While e14.5 EPs preferentially give rise to alpha cells, these EPs are still competent and able to become beta cells and beta cells do still differentiate at this earlier stage but in low numbers. What we note is shifts in potential between e14.5 and e16.5 EPs, which we believe impacts the formation of alpha and beta cells. As we saw that BPs, which are progenitors for EPs, were also divergent between e14.5 and e16.5, we sought to determine if any factors enriched in e16.5 BPs could impact the alpha/beta fate decision. Our findings that loss of Amotl2 (enriched in e16.5 BPs, when more beta cells form) in hESC-derived pancreatic progenitors leads to a decrease in beta cell formation suggests that endocrine fate determination can be influenced in even earlier stages. However, we recognize that we don't focus on this in the manuscript and therefore we have revised the manuscript to make sure we do not overstate these results, as they would require more experiments beyond the scope of this work to convincingly substantiate this finding.

- Fig7e: The immunostaining inlet for Ins in the Amotl GOF does not really support the conclusion.

We have changed this inlet to a more appropriate inlet that reflects our cell fate phenotype.

- GO enrichment analysis is not described, please add the type of tests and groups which were compared.

We have added this information throughout the results section as well as in the methods section, as shown below:

“Gene ontology analysis was performed using DAVID Bioinformatics Resources 6.8^{81,82} and Metascape [metascape.org]⁸³.”

- Computational analyses use standard algorithms. However, quantitative metrics are missing. It would be nice to supplement the qualitative trend observations described in the main text with fold changes (e.g. initial to maximal) or p.values for differential expression.

We have now revised the manuscript to include quantitative metrics where appropriate.

- The description/discussion of the results is very often difficult to follow. There are also many sentences in which words are missing e.g. page 5 line 108-111.

We have edited the manuscript to ensure there are no typos, and revised the discussion section to make it flow better.

--

Reviewer #3 (Remarks to the Author):

Endocrine lineage biases arise in temporally distinct endocrine progenitors during pancreatic morphogenesis

In this work, the authors performed single-cell RNA sequencing of the pancreas at e14.5 and e16.5, when the majority of pancreatic EPs appear. They show that at each of these developmental stages four subtypes of EPs are identified, reflecting different maturation stages.

New EPs are being born at both e14.5 and e16.5, and specific markers for the different subtypes maintain in both stages. Despite these commonalities, the EPs from e14.5 are inherently different than the EPs in e16.5 in regards to gene expression, and cluster separately. Importantly, the data suggests that EPs from e16.5 have higher potential for beta cell differentiation than EPs from e14.5. The authors also perform ATAC-seq on purified Ngn3 positive EPs from e14.5 and e16.5, and show different chromatin accessibility. This is not surprising considering their previous finding that these EPs are transcriptionally distinct. The e16.5 ATAC-seq maps reveal enrichment for beta cell motifs, also supporting their conclusions of a bias towards beta cell differentiation of these EPs. Over all, the authors present a large body of evidence to support their conclusions, which would be of great interest to the community, and may be a valuable resource for future studies.

Thank you for your careful reading of the manuscript and kind comments. We also believe this data could be a valuable resource for the pancreatic community. We appreciate your comments and have edited our manuscript, figures, and added new experimental data to address your comments and our responses are below.

Specific comments:

1. The abstract is not well written to reflect what is shown in the paper. It should better reflect what was done in this work, and not the possible interpretation of the results (similar to a discussion). For example: The first sentence is too general and not specific to this work. In the sentence: "Thus fate determination occurs before Neurogenin3 expression, affecting the chromatin state and transcriptome of EPs at each stage"- the causality of the change in fate determination that affects chromatin and transcription is not shown here- it is an interpretation of the results. Also claiming to present a transcriptional and epigenetic atlas of the pancreatic endocrine development is an over statement.

Thank you for letting us know that the abstract was not clear. We have re-written the abstract to better convey our findings. We also have read through the manuscript to ensure that we do not in the results section interpret the implications of the results and only made these inferences in the discussion section.

2. Page 6: the authors mention specific genes such as *Btbd17*, *Grasp*, *Fev* and *St18* that were not previously associated with EPs. It would be informative to mention what are these genes (do they have known roles in other pathways?) and why these genes were specifically chosen here. Are these the most differentially expressed? Or is there a biological rationale? The same questions also apply for *Dcdc2a*- why did you choose to follow up specifically on this gene?

These genes were selected because they were among the most differentially expressed without known functions in EPs. We have added a brief description of what is known about these genes to the text, and written in the text why they were chosen:

"We also found transcripts significantly enriched in EPhi and EPlo not previously associated with EPs, such as *Btbd17* and *Grasp*, expressed exclusively in EPhi (Fig. 1f). *Btbd17* has been shown to be epigenetically modified in hepatocellular carcinoma but has no other known function³³, while *Grasp* is known in neurons to regulate intracellular trafficking mGluR organization³⁴. In contrast, EPlo uniquely and differentially expressed *Fev*³⁵ and *St18*, a paralog of *Myt1*³⁶. We also found novel transcripts enriched in BPs, including highly enriched expression of *Dcdc2a*, which plays a role in inhibition of the canonical Wnt signaling pathway in renal tissue³⁷. We confirmed by immunostaining in e14.5 BPs and in the human wk10.6 pancreas

(Fig. 1g, 1h, and Supplementary Fig. 1d; 77% of Sox9+ cells were Dcdc2a+ by immunostaining with SEM 4.8).”

3. Fig 1G: it seems in the image that Dcd2a is expressed in more cells than Sox9, is that indeed the case? Doesn't it contradict the expression data in figure 1F?

To confirm that a subset of Sox9+ cells were Dcdc2a+, we quantified our immunostainings and found that 77.0% of Sox9+ cells were Dcdc2a+, with an SEM of 4.8%. By single cell RNA-sequencing, 31% of Sox9+ cells were Dcdc2a+. The difference in Dcdc2a+ cell percentage might be due to post-transcriptional regulation, as one technique illustrates protein while the other RNA, or it may at least partially result from poor quality of Dcdc2a antibodies in immunostaining the embryonic pancreas. We have added single channel images to clarify the expression of these genes and added the quantification data to the text in the manuscript, as shown below:

“We also found novel transcripts enriched in BPs, including expression of *Dcdc2a*, which we confirmed by immunostaining in e14.5 BPs and in the human wk10.6 pancreas (Fig. 1g, 1h, and Supplementary Fig. 1d; 77% of Sox9+ cells were Dcdc2a+ by immunostaining with SEM 4.8).”

4. Is it possible to do IF for some of the differentially expressed genes in EPhi and EPlo, in combination with Ngn3-GFP, in order to validate expression of these genes is indeed associated with low/high levels of Ngn3?

We have immunostained Ngn3-eGFP tissue for genes that were significantly different between EPhi and EPlo and provided this data in a new supplementary figure, Supplementary Fig. 3. We have edited the text to refer to this figure:

“We validated EPhi and EPlo cells by immunostaining Ngn3-eGFP pancreata for differentially expressed genes (Nkx6-1 and Sox4 in EPhi; Chga, NeuroD1, and E-cadherin in EPlo; Supplementary Fig. 3b and 3c).”

5. Figure 2: In order to validate their results of the different subtypes, it would be helpful to stain for two proteins in the same cells. For example, it would highly support their conclusions if they can show with IF that Hes1 is found in the same cells with Vim, but not in cells that express Neurod1. Also, it might indicate on the spatial distribution of the different subtypes. Adding some statistics (such as what percentage of cells express different markers according to IF) would also be helpful.

Following your suggestion, we have immunostained for combinations of markers in Ngn3-eGFP pancreatic sections, providing 45 new images from 9 different combinations and 9 different markers, resulting in two new figures (Supplementary Fig. 6 and 7). We have described this in the text as follows:

“We further validated the different subtypes by immunostaining for a combination of markers to delineate subtype identity (Supplementary Fig. 6 and 7). For example, we observed N14_1 cells that co-expressed Ngn3-eGFP with E-cadherin but not Chga, as well as Ngn3-eGFP+ cells that co-express Chga but not E-cadherin for N14_4 (Supplementary Fig. 6).”

We have added the percentage of Ngn3-eGFP+ cells from immunostaining side by side with the percentage determined through single-cell RNA-seq in Supplementary Fig. 5b, and added a sentence in the text to describe this:

“By quantification, we validated the proportion of *Ngn3*⁺ cells co-expressing these markers, showing strong association with the proportion by scRNA-seq (Supplementary Fig. 5b).”

6. In the epigenetic analysis, the authors conclude that there is a chromatin accessibility remodeling between e14.5 and e16.5 that leads to a more permissive environment for beta cell formation. They connect it to the changes seen in the expression of chromatin remodelers (Fig 3). However, the causality is not clear here, and I would refrain from over interpretation of the results (especially without any functional studies to support it, such as perturbations to chromatin regulators, which I believe is out of scope for this work).

As the EPs from e14.5 and e16.5 were already shown to be transcriptionally distinct, it makes sense that ATAC-seq maps are different. The authors suggest that changes in chromatin lead to changes in expression, but it is not necessarily the case (it is very possible that changes in chromatin accessibility merely represent the changes in gene expression). In that regard, it would be interesting to further analyze some of the beta-specific TFs that are indicated in figure 5d. For example- we see enrichment for *Nkx6.1* motifs in the e16.5 EPs. Are there changes in the levels of *Nkx6.1* itself between e14.5 and e16.5? Is it possible to do ChIP-seq for this TF to see where it is bound in the two stages?

We agree that the changes in chromatin accessibility may not be causal for the shifts in EP potential, and we have revised our results to more appropriately describe our findings without overstating their potential implications. As discussed for Reviewer 2, we have expanded our ATAC-seq analysis to include ChIP-seq data sets.

Per your suggestion, we interrogated the top enriched motif from the 749 differentially accessible (adjusted P-value < 0.05) e16.5 peaks, which was *Nkx6-1* (Motif enrichment p-value = 1e-86). We mapped and analyzed publically available adult *Nkx6-1* ChIP-seq data (from 4 week old murine islets), and found a striking enrichment for *Nkx6-1* binding at these e16.5 regions. The 609 e14.5 EP enriched ATAC-seq peaks showed very low *Nkx6-1* ChIP-seq enrichment (new Figure 5f-g), which is in agreement with our motif enrichment analysis (Figure 5e). We found that *Nkx6-1* expression levels dramatically increase in e16.5 *Ngn3*⁺ cells compared to e14.5 *Ngn3*⁺ cells (added to Supplement Fig. 14d and shown below). *Nkx6-1* is a critical regulator of beta cell identity and development. And this data suggests that at least a portion of the mature postnatal beta cell gene regulatory network and *Nkx6-1* cistrome is setup as early as e16.5. While we weren't able to perform ChIP-seqs for *Nkx6-1* at these two stages given the amount of tissue and time required, we believe that the strong correlation of our ATAC-seq data with the *Nkx6-1*, *NeuroD1*, and histone mark ChIP-seqs is sufficient to help bolster the validity of our ATAC-seq data. See below Rebuttal Figure 5.

“Chromatin accessibility of *Ngn3* and *Nr5a2* (a tip cell marker) loci were enriched at e14.5 compared to e16.5, while accessibility across *Ins1* and potassium calcium activated channel *Kcnn2* loci were increased at e16.5 (Fig. 5d). To investigate if temporal changes in motif accessibility in EPs corresponded with endocrine fate determination, we performed motif enrichment analysis on the statistically differentially accessible ATAC-seq peaks (adjusted P-value <0.05). Strikingly, we found that beta cell associated transcription factor motifs including Nkx6-1, NeuroD1, and Isl1 were well represented at e16.5 (Fig. 5e). Indeed, Nkx6-1 is known to be part of a gene regulatory network essential for maintaining beta-cell identity and molecular physiology into adulthood⁶⁰⁻⁶³. From scRNA-seq, we observed a robust increase in expression of Nkx6-1 in e16.5 *Ngn3*+ cells compared to e14.5 *Ngn3*+ cells (Supplementary Fig. 15d). Importantly, analysis of Nkx6-1 ChIP-seq data⁶² from adult pancreatic islets confirmed that Nkx6-1 was bound to many of the genomic regions we found to be more accessible in e16.5 EPs (Fig. 5f and 5g). Furthermore, as NeuroD1 motifs were enriched globally in both data sets as well as in e16.5-specific EP ATAC-seq peaks, we assessed NeuroD1 DNA-binding from e14.5 pancreas NeuroD1 ChIP-seq data⁶⁶ and found moderately enhanced occupancy across the e16.5 peak set (Fig. 5h). Thus, many of the accessible chromatin regions identified in embryonic EPs are occupied by key pancreatic transcription factors, including Nkx6-1 and NeuroD1.

Given that many of the differentially accessible regions were annotated to intronic and intergenic regions (92% for e14.5, and 94% for e16.5), we sought to interrogate their epigenetic status. We integrated publically available histone mark ChIP-seq profiling data derived from murine e13.5 embryonic EPs (*Ngn3*-GFP+ low), and e17.5 beta-cells⁶⁴. Indeed, the 609 e14.5 enriched ATAC-seq peaks showed high H3K4me1 signal in e13.5 *Ngn3*-GFP low cells, consistent with their status as active enhancers with high chromatin accessibility (Fig. 5i). Conversely, e13.5 *Ngn3*-GFP low H3K4me1 signal was not highly enriched across the 749 e16.5 accessible ATAC-seq peaks. This suggests that the 749 e16.5 specific peaks are not yet commissioned as active enhancer regions at e14.5. However, further histone mark profiling is required to determine the enhancer status of these regions at e16.5 definitively. Thus, the differentially accessible regions identified in EPs via ATAC-seq are dynamic at the epigenetic level. In e17.5 beta cells, we also noted increased signal for the active H3K27ac mark in the accessible peaks of e16.5 EPs compared to e14.5 EPs (Fig. 5j). Conversely, we observed decreased signal for inactive H3K27me3 marks in e17.5 beta cells across e16.5 EP peaks (Fig. 5k). Together this supports that the e16.5 peaks are also active genomic regulatory regions in e17.5 beta cells. In addition, gene ontology analysis of the accessible peaks in e14.5 and e16.5 EPs revealed enrichment of hallmarks of pancreatic beta cells at e16.5 (Supplementary Fig. 15c). Overall, this suggests that changes in EP chromatin accessibility could influence EP potential, with chromatin accessibility remodeling between e14.5 and e16.5 leading to a more permissive environment for beta cell formation at e16.5.”

Minor comments:

1. Page 3 line 61: “only” appears twice.
2. Page 8 line 179: Vim is expressed also in EP14_2 and not just EP14_1.
3. Page 16 line 393: delete “observed”.

Thank you for your keen eye, we have amended these errors.

Reviewers' Comments:

Reviewer #1:

Remarks to the Author:

The authors have addressed most of the comments. The manuscript is significantly improved after revision. However, there are still some issues which need to be resolved.

Specific points:

Previous comments are marked with ""

Authors' response to comments are marked with []

1. In the EP subtype analysis, the authors have not addressed the previous comment:

"2. ...The authors should also clarify how the number of cell clusters is selected (e.g., 4 for e14.5 and 4 for e16.5)."

The authors should clarify how the number of subtypes (i.e., 4) is selected for both e14.5 and e16.5.

2. In the EP subtype analysis in e14.5, subtype N14_2 seems to be very different from N14_3 and N14_4. Supplementary Fig. 4b shows that N14_2 contains more cells from batch 1 than the other two batches. The authors also mentioned in the revised manuscript that [We found equal representation of cells from all three batches in every cluster, with the exception of three mesenchyme clusters and a cluster of hepatocytes composed mostly of batch 1 cells (Mes2 cluster 1; Pr. Mes2 cluster 6; Mes3 cluster 8; Hepato cluster 27), likely due to increased inclusion of surrounding tissue during the first dissection (Supplementary Fig. 2d)]. The authors should check whether N14_2 contains cells from the surrounding tissue in batch 1. If so, these cells should be excluded from N14_2 since they may present artifacts and are not reproducible by the other two batches.

In addition, N14_2 does not seem to be a maturation stage between N14_1 and N14_3. According to the pseudotime analysis in Supplementary Fig. 9a, N14_1 and N14_2 shared very similar pseudotime and N14_1 cells are closer to N14_3 cells. From the EP subtype analysis in e16.5, N16_1 also contains EMT cells as showed in Supplementary Fig. 11d. The authors should show why N14_2 (mostly EMT cells) is treated as a separate cluster in e14.5. Based on the current analyses, it seems better to group N14_2 together with N14_1 as one cluster.

3. In response to the previous comment:

"4. ...The authors claimed that EPs are different between e14.5 and e16.5 (Fig. 4f and 4h). They have to test whether the observation is caused by batch effects (e14.5 and e16.5 cells are from different mice and different sequencing runs). For instance, Fig 4f and 4h lack of a control cell type to show the difference of e14.5 and e16.5 EPs is not caused by batch effects. Such a control cell type could be BPs as shown in Fig. 7a where BPs from e14.5 and e16.5 are partially overlapped. The authors can either mark the EPs in Fig. 7a according to their origins (e.g., from e14.5 or e16.5) or add BPs to Fig. 4f to see if EPs from the two developmental time points are very distinct from each other. Similarly, the authors should also show that the PCA plot in Fig. 6a is not affected by batch effects (e.g., including BPs in the plot may help to clarify). The authors should also clarify how Fig. 6a is generated (it seems to be generated by pooling cells together as pseudo bulk samples)."

[While we previously showed origin of each batch in the Ngn3⁺ cells (now Supplementary Fig. 4b), we agree that adding in other cell types to the tSNE and PCA analysis such as BPs or endocrine cells would be a strong control and we are thankful for this suggestion.

We have provided this new data in Supplementary Fig. 13d, showing tSNE plots of e14.5 and e16.5 Ngn3⁺ cells with e14.5 beta cells as well as with e14.5 BPs, alpha, and beta cells. We believe that showing e16.5 Ngn3⁺ cells clustering apart even with these distinct cell types added in strengthens the conclusion that e14.5 and e16.5 Ngn3⁺ cells are divergent. We have added the following sentence to describe this:

“To ensure that the differences between the e14.5 and e16.5 Ngn3+ cells was not an artifact of batch effects, we subclustered Ngn3+ cells with e14.5 beta cells as well as with e14.5 beta cells, alpha cells, and BPs, and in all cases found the e16.5 Ngn3+ cells clustered separately from e14.5 Ngn3+ cells (Supplementary Fig. 13d).”]

Here, the control cell type should include cells from both time points (e14.5 and e16.5). For example, the authors should show beta cells from both e14.5 and e16.5, or beta cells, alpha cells, and BPs from both e14.5 and e16.5 in Supplementary Fig. 13d. The idea is to show that the control cell type is similar between e14.5 and e16.5 while EPs from e14.5 and e16.5 are different. Actually, rebuttal Figure 2B and C are good examples for addressing this point.

[Further, we have added BPs in the PCA plot in Fig. 7a (previously Fig. 6a), which substantiates that N14_1 and N16_1 are divergent. This has been added in the text as follows:

“Principal component analysis revealed that e14.5 and e16.5 EPs were already different upon induction, with the earliest N14_1 and N16_1 subtypes clustering apart, even more so than BPs (Fig. 7a; average cluster expression).”]

Similarly, BPs from e14.5 and e16.5 should be showed separately in Fig. 7a. For instance, show BPs from e14.5 as a dot and BPs from e16.5 as another dot in the figure. The idea is to show BPs from e14.5 and e16.5 are similar while EPs are different.

4. In response to the previous comment:

“5. It is unclear what cells are used in the analysis of differentiation from EPs to alpha and beta cells in Fig. 8a (e.g., e14.5? e16.5? or merged e14.5 and e16.5?). If merged e14.5 and e16.5 EPs are used in Fig. 8a, this contradicts the previous conclusion that EPs from e14.5 and e16.5 were very distinct.”

Rebuttal Figure 2B and C are the best plots to address this point. The annotation of “Epe14” and “Epe16” in Rebuttal Figure 2A (i.e., Fig. 8a) is misleading. It would be better to show rebuttal Figure 2B and C instead of the current two plots (i.e., Rebuttal Figure 2A and B) in Fig. 8a.

[Indeed, Epe14 cells have a mean pseudotime score of 10.2 (standard deviation = 1.8) and Epe16 have a pseudotime score of 8.5 (standard deviation = 2.6), thus they are similar but distinct. Below is the Rebuttal Figure 2 addressing these points.

We have added the following to the text and revised Figure 8a in main text so that this is clear to the reader:

“Consistent with our previous analysis, EPhi cells appear prior to EPl0 before bifurcation into either alpha or beta cells, with e14.5 EPs clustering together into three clusters while e16.5 EPs clustered apart (Epe14 cells have a mean pseudotime score of 10.2, standard deviation = 1.8; Epe16 have a pseudotime score of 8.5, standard deviation = 2.6).”]

Rebuttal Figure 2F should be sufficient to address the point that e14.5 and e16.5 cells are distinct in the Monocle2 plot. The difference of mean and variance between the pseudotime of e14.5 and e16.5 cells is not a good argument since the number of cells from e14.5 and e16.5 are quite different. It would be better not to use this as an argument for the difference between e14.5 and e16.5 cells in the manuscript.

Reviewer #2:

Remarks to the Author:

Almost all of my comments have been addressed except that the authors have not increased the numbers of cells analyzed at E16.5 using scRNAseq. I still think that the number of cells is relative low to the high number of cells analyzed at E14.5.

Minor comments:

- Labels are incomplete in Fig. 1f, i.e. 17...Rik???, Grasp (1 or 2?). Please provide unique identifiers in the Figure.
- Line 153-154: immonstaining with SEM 4.8." What does this mean?

Reviewer #3:

Remarks to the Author:

The authors have adequately addressed my comments, and I recommend accepting the paper for publication. They have added data and clarifications in the text and the paper is significantly improved.

REVIEWERS' COMMENTS:

Reviewer #1 (Remarks to the Author):

The authors have addressed most of the comments. The manuscript is significantly improved after revision. However, there are still some issues which need to be resolved.

Specific points:

Previous comments are marked with ""

Authors' response to comments are marked with []

1. In the EP subtype analysis, the authors have not addressed the previous comment: "2. ...The authors should also clarify how the number of cell clusters is selected (e.g., 4 for e14.5 and 4 for e16.5)."

We did not select the number of clusters, but rather the four subtypes were derived from a series of unsupervised analyses to determine similarities and differences in expression between single cells. Thus, these clusters were produced in the same manner as our whole single cell RNA-seq clustering.

We first selected for cells expressing *Ngn3* (Seurat, `SubsetData` function) before identifying variable genes (Seurat, `FindVariableGenes`) and running PC analysis (Seurat, `RunPCA`). As with our whole pancreas scRNA-seq clustering, to select meaningful PCs for inclusion into the clustering we looked at the standard deviation of each principle component (Seurat, `PCElbowPlot`), and then determined a cutoff based on this graph (point at which elbow is clear, see http://satijalab.org/seurat/pbmc3k_tutorial.html). Next, PCs near cutoff were inspected to determine if meaningful genes were present. Finally, if the PC at the cutoff had enrichment for relevant markers, it was used as the cutoff for graph-based clustering (Seurat, `FindClusters`) and tSNE visualization (Seurat, `RunTSNE`). To ensure that the clusters obtained from this series of statistical tests were indeed distinct, we next performed differential expression analysis on all clusters. This was projected onto the heatmap shown in Fig. 2b.

The authors should clarify how the number of subtypes (i.e., 4) is selected for both e14.5 and e16.5.

Originally, as a default of the Seurat package, the number of each subtype reflected the size of the cluster, or how many cells were present in each. However after noting that the clusters reflected developmental progression, and ordered in a linear manner through pseudotime, we renamed each cluster according to their developmental order. Thus cluster 1 (previously #1) represents the cells in the first pseudotime group, followed by cluster 2 (previously #3), then cluster 3 (previously #0), and finally cluster 4 (previously #2). We have added this information to the manuscript in the figure legend of Fig. 2a as follows:

"Four subpopulations of *Ngn3*+ cells exist at e14.5. Subclustering of *Ngn3*+ cells visualized using tSNE. Colors indicate cluster identity. Clusters were numbered by developmental trajectory (see Fig. 3)."

2. In the EP subtype analysis in e14.5, subtype N14_2 seems to be very different from N14_3 and N14_4. Supplementary Fig. 4b shows that N14_2 contains more cells from batch 1 than the other two batches. The authors also mentioned in the revised manuscript that [We found equal representation of cells from all three batches in every cluster, with the exception of three mesenchyme clusters and a cluster of hepatocytes composed mostly of batch 1 cells (Mes2 cluster 1; Pr. Mes2 cluster 6; Mes3 cluster 8; Hepato cluster 27), likely due to increased inclusion of surrounding tissue during the first dissection (Supplementary Fig. 2d)]. The authors should check whether N14_2 contains cells from the surrounding tissue in batch 1. If so, these cells should be excluded from N14_2 since they may present artifacts and are not reproducible by the other two batches.

In addition, N14_2 does not seem to be a maturation stage between N14_1 and N14_3. According to the pseudotime analysis in Supplementary Fig. 9a, N14_1 and N14_2 shared very similar pseudotime and N14_1 cells are closer to N14_3 cells. From the EP subtype analysis in e16.5, N16_1 also contains EMT cells as showed in Supplementary Fig. 11d. The authors should show why N14_2 (mostly EMT cells) is treated as a separate cluster in e14.5. Based on the current analyses, it seems better to group N14_2 together with N14_1 as one cluster.

N14_2 cells represent the small population of cells undergoing widespread changes to undergo EMT; delaminating from the epithelium and migrating out into the surrounding niche. As you pointed out, we find many of these genes expressed in N16_1, which group together into one group likely due to a lower sampling size (837 cells at e14.5 vs. 110 at e16.5). However, the existence of these cells that co-express EMT genes with Ngn3 in both e14.5 and e16.5 supports that this is a distinct developmental stage of Ngn3+ cells. Further, we observe these group 2 Ngn3+ cells in secondary analyses including immunostainings (Fig. 2 and Supplemental Fig. 5: Vim, Dcn, Zeb2; Supplemental Fig. 6: Col4a1; Supplemental Fig. 7: Nfia), flow cytometry (Supplemental Fig. 8: Dcn) and lineage tracing (Supplemental Fig. 10: Zeb2). However, we have added the following sentence to the manuscript to acknowledge this:

“While we found N14_2 as a distinct subtype of Ngn3+ cells highly expressing EMT genes, enrichment of these transcripts were observed in N16_1. This is possibly due to a smaller cell number sampled at e16.5, with this cluster similar to both N14_1 and N16_1.”

3. In response to the previous comment:

“4. ...The authors claimed that EPs are different between e14.5 and e16.5 (Fig. 4f and 4h). They have to test whether the observation is caused by batch effects (e14.5 and e16.5 cells are from different mice and different sequencing runs). For instance, Fig 4f and 4h lack of a control cell type to show the difference of e14.5 and e16.5 EPs is not caused by batch effects. Such a control cell type could be BPs as shown in Fig. 7a where BPs from e14.5 and e16.5 are partially overlapped. The authors can either mark the EPs in Fig. 7a according to their origins (e.g., from e14.5 or e16.5) or add BPs to Fig. 4f to see if EPs from the two developmental time points are very distinct from each other. Similarly, the authors should also show that the PCA plot in Fig. 6a is not affected by batch effects (e.g., including BPs in the plot may help to clarify). The authors should also clarify how Fig. 6a is generated (it seems to be generated by pooling cells together as pseudo bulk samples).”

[While we previously showed origin of each batch in the Ngn3+ cells (now Supplementary Fig. 4b), we agree that adding in other cell types to the tSNE and PCA analysis such as BPs or endocrine cells would be a strong control and we are thankful for this suggestion. We have provided this new data in Supplementary Fig. 13d, showing tSNE plots of e14.5 and e16.5 Ngn3+ cells with e14.5 beta cells as well as with e14.5 BPs, alpha, and beta cells. We believe that showing e16.5 Ngn3+ cells clustering apart even with these distinct cell types added in strengthens the conclusion that e14.5 and e16.5 Ngn3+ cells are divergent. We have added the following sentence to describe this:

“To ensure that the differences between the e14.5 and e16.5 Ngn3+ cells was not an artifact of batch effects, we subclustered Ngn3+ cells with e14.5 beta cells as well as with e14.5 beta cells, alpha cells, and BPs, and in all cases founded the e16.5 Ngn3+ cells clustered separately from e14.5 Ngn3+ cells (Supplementary Fig. 13d).”]

Here, the control cell type should include cells from both time points (e14.5 and e16.5). For example, the authors should show beta cells from both e14.5 and e16.5, or beta cells, alpha cells, and BPs from both e14.5 and e16.5 in Supplementary Fig. 13d. The idea is to show that the control cell type is similar between e14.5 and e16.5 while EPs from e14.5 and e16.5 are different. Actually, rebuttal Figure 2B and C are good examples for addressing this point.

[Further, we have added BPs in the PCA plot in Fig. 7a (previously Fig. 6a), which substantiates

that N14_1 and N16_1 are divergent. This has been added in the text as follows:

“Principal component analysis revealed that e14.5 and e16.5 EPs were already different upon induction, with the earliest N14_1 and N16_1 subtypes clustering apart, even more so than BPs (Fig. 7a; average cluster expression).”]

Similarly, BPs from e14.5 and e16.5 should be showed separately in Fig. 7a. For instance, show BPs from e14.5 as a dot and BPs from e16.5 as another dot in the figure. The idea is to show BPs from e14.5 and e16.5 are similar while EPs are different.

4. In response to the previous comment:

“5. It is unclear what cells are used in the analysis of differentiation from EPs to alpha and beta cells in Fig. 8a (e.g., e14.5? e16.5? or merged e14.5 and e16.5?). If merged e14.5 and e16.5 EPs are used in Fig. 8a, this contradicts the previous conclusion that EPs from e14.5 and e16.5 were very distinct.”

Rebuttal Figure 2B and C are the best plots to address this point. The annotation of “EPe14” and “EPe16” in Rebuttal Figure 2A (i.e., Fig. 8a) is misleading. It would be better to show rebuttal Figure 2B and C instead of the current two plots (i.e., Rebuttal Figure 2A and B) in Fig. 8a. [Indeed, EPe14 cells have a mean pseudotime score of 10.2 (standard deviation = 1.8) and EPe16 have a pseudotime score of 8.5 (standard deviation = 2.6), thus they are similar but distinct. Below is the Rebuttal Figure 2 addressing these points.

We have added the following to the text and revised Figure 8a in main text so that this is clear to the reader:

“Consistent with our previous analysis, EPhi cells appear prior to EPlo before bifurcation into either alpha or beta cells, with e14.5 EPs clustering together into three clusters while e16.5 EPs clustered apart (EPe14 cells have a mean pseudotime score of 10.2, standard deviation = 1.8; EPe16 have a pseudotime score of 8.5, standard deviation = 2.6).”]

Rebuttal Figure 2F should be sufficient to address the point that e14.5 and e16.5 cells are distinct in the Monocle2 plot. The difference of mean and variance between the pseudotime of e14.5 and e16.5 cells is not a good argument since the number of cells from e14.5 and e16.5 are quite different. It would be better not to use this as an argument for the difference between e14.5 and e16.5 cells in the manuscript.

Thank you for these suggestions. We have changed in Fig. 8a the plots to Rebuttal Fig. 2B and 2C rather than 2A and 2B to better convey the differences between EPs at e14.5 and e16.5. Further, we have added Rebuttal Fig. 2F to Fig. 8a as well and removed the mean and variance from the manuscript.

Reviewer #2 (Remarks to the Author):

Almost all of my comments have been addressed except that the authors have not increased the numbers of cells analyzed at E16.5 using scRNAseq. I still think that the number of cells is relative low to the high number of cells analyzed at E14.5.

At e14.5 we sequenced 15,228 single cells while at e16.5 we sequenced 2,006 single cells. While there are fewer cells sequenced at e16.5, the 2,006 single cells are a high number and allow visualization of many subtypes of pancreatic cells. Further, the supplement of the droplet based scRNA-seq with other methods at e14.5 and e16.5 including MATQ-seq, Fluidigm C1, and immunostainings are supportive that the sample size is sufficient. However, we did note that at e16.5 the small cluster expressing EMT genes did not cluster alone, therefore we have added the following sentence to the manuscript to acknowledge this:

“While we found N14_2 as a distinct subtype of Ngn3+ cells highly expressing EMT genes, enrichment of these transcripts were observed in N16_1. This is possibly due to a smaller cell number sampled at e16.5, with this cluster similar to both N14_1 and N16_1.”

Minor comments:

- Labels are incomplete in Fig. 1f, i.e. 17...Rik???, Grasp (1 or 2?). Please provide unique identifiers in the Figure.
- Line 153-154: immunostaining with SEM 4.8.” What does this mean?

We have added the full gene name for 17...Rik into the figure legend, as the full name does not fit in the figure without reducing the text size. There is only one Grasp gene in the mouse genome, noted as Grasp on NCBI (ID: 56149). We have also amended line 153-154, changing to the following:

“We confirmed expression of Dcdc2a by immunostaining in e14.5 BPs and in the human wk10.6 pancreas (Fig. 1g, 1h, and Supplementary Fig. 1d; 77% \pm 4.8 of Sox9+ cells were Dcdc2a).”

--

Reviewer #3 (Remarks to the Author):

The authors have adequately addressed my comments, and I recommend accepting the paper for publication. They have added data and clarifications in the text and the paper is significantly improved.